# Single-cell transcriptome maps of myeloid blood cell lineages in *Drosophila*

Bumsik Cho[1,7], Sang-Ho Yoon [1,7], Daewon Lee[1], Ferdinand Koranteng[1], Sudhir Gopal Tattikota [2], Nuri Cha[1], Mingyu Shin[1], Hobin Do [1], Yanhui Hu[2], Sue Young Oh[3], Daehan Lee[1], A. Vipin Menon [1], Seok Jun Moon [3], Norbert Perrimon[2,4], Jin-Wu Nam [1,5,6 ✉] & Jiwon Shim [1,5,6 ✉]

The *Drosophila* lymph gland, the larval hematopoietic organ comprised of prohemocytes and mature hemocytes, has been a valuable model for understanding mechanisms underlying hematopoiesis and immunity. Three types of mature hemocytes have been characterized in the lymph gland: plasmatocytes, lamellocytes, and crystal cells, which are analogous to vertebrate myeloid cells, yet molecular underpinnings of the lymph gland hemocytes have been less investigated. Here, we use single-cell RNA sequencing to comprehensively analyze heterogeneity of developing hemocytes in the lymph gland, and discover previously unde-scribed hemocyte types including adipohemocytes, stem-like prohemocytes, and inter-mediate prohemocytes. Additionally, we identify the developmental trajectory of hemocytes during normal development as well as the emergence of the lamellocyte lineage following active cellular immunity caused by wasp infestation. Finally, we establish similarities and differences between embryonically derived- and larval lymph gland hemocytes. Altogether, our study provides detailed insights into the hemocyte development and cellular immune responses at single-cell resolution.

[1] Department of Life Sciences, College of Natural Science, Hanyang University, Seoul 04736, Republic of Korea. [2] Department of Genetics, Harvard Medical School, Boston, MA 02115, USA. [3] Department of Oral Biology, Yonsei University, College of Dentistry, Seoul 03722, Republic of Korea. [4] Howard Hughes Medical Institute, Boston, MA 02115, USA. [5] Research Institute for Natural Sciences, Hanyang University, Seoul 04736, Republic of Korea. [6] Research Institute for Convergence of Basic Sciences, Hanyang University, Seoul 04736, Republic of Korea. [7] These authors contributed equally: Bumsik Cho, Sang-Ho Yoon. ✉email: jwnam@hanyang.ac.kr; jshim@hanyang.ac.kr

Blood cells are highly specialized cells that play crucial roles in the elimination of foreign threats during immune responses and in various forms of stress responses and development[1]. Blood cells in *Drosophila*, collectively called hemocytes, are reminiscent of myeloid-lineage blood cells in vertebrates[1–3], and are represented by at least three morphologically distinct hemocyte populations: plasmatocytes (PM), crystal cells (CC), and lamellocytes (LM). Plasmatocytes, which comprise ~95% of the hemocytes, play a role in phagocytosis, tissue remodeling, and cellular immune responses—much like macrophages, their vertebrate counterparts[4,5]. Crystal cells account for ~5% of the blood population and are characterized by crystalline inclusions made up of prophenoloxidase (ProPO), an enzyme responsible for activating the melanization cascade[6–8]. Finally, lamellocytes, which are seldom seen in healthy animals grown at normal conditions, mostly differentiate upon parasitic wasp infestation or environmental challenges[9–12].

Blood development in vertebrates involves the primitive and definitive waves of hematopoiesis[13]. Reminiscent of vertebrate hematopoiesis, two hematopoietic waves have been described during *Drosophila* development, embryonic and larval lymph gland hematopoiesis[14]. Hematopoiesis in the lymph gland is initiated from hemangioblast-like cells in the embryonic cardiogenic mesoderm, which give rise to the primary lobe of the larval lymph gland[15]. Medially located prohemocytes, which sustain the developmental potential to generate all three mature hemocyte types, constitute the medullary zone (MZ) and continue to proliferate until the early third instar[16]. Mature hemocytes emerge at the distal edge of the lymph gland from mid-second instar and comprise the cortical zone (CZ)[4]. Located between the undifferentiated medullary zone and the differentiated cortical zone, is the intermediate zone (IZ) that contains a group of differentiating cells expressing markers for both the medullary zone and the cortical zone[17]. The posterior signaling center (PSC), a small group of cells that secrete various ligands, is located at the medioposterior side of the lymph gland and regulates proper growth of the rest of the lymph gland[18–20] (Fig. 1a). Lymph glands from healthy larvae reared under normal lab conditions generally follow fixed developmental states until late third instar. Remarkably, following the onset of pupariation, the lymph gland disintegrates, allowing hemocytes to disperse into circulation[21].

Female wasps, including those of the genus *Leptopilina*[22], attack second-instar larvae via a sharp needle-like ovipositor that efficiently delivers their eggs[23]. Wasp eggs trigger cellular immune responses that accompany lamellocyte differentiation from both embryonically-derived and lymph gland-derived hemocytes[24]. Lamellocytes are seen in circulation by 24 h post-infestation; yet, lamellocytes generated in the lymph gland remain in their original location. Within 48 h after infestation, a massive differentiation of lamellocytes takes place followed by disruption of the lymph gland[25]. Hemocytes in the lymph gland eventually dissociate into circulation, and mature lamellocytes derived from the lymph gland and hematopoietic pockets encapsulate and neutralize wasp eggs[26].

The *Drosophila* lymph gland has been largely characterized based on genetic markers and cellular morphology. However, the molecular underpinnings of hematopoietic cells such as different states and the gene regulatory network of each cell type have been less investigated. In addition, questions as to how prohemocytes and mature hemocytes differentiate into lamellocytes upon active immunity, and to what extent hemocytes derived from the embryonic and the lymph gland hematopoiesis differ have been unanswered[1].

Here, we build a census of myeloid-like *Drosophila* hemocytes by taking advantage of single-cell RNA sequencing (scRNA-seq) technology and establish a detailed map for larval hemocytes in the developing lymph gland. We identify classes of hemocytes and their differentiation trajectories and describe molecular and cellular changes of myeloid-like hemocytes upon immune challenges. Furthermore, we identify both distinct and common characteristics of hemocytes originating from embryonic and larval lymph gland lineages. Altogether, our work will stimulate future studies on the development and diverse functions of the myeloid-like blood cell lineage.

## Results

**Single-cell transcriptomic profiling of developing hemocytes.** To understand the cellular diversity of developing hemocytes in *Drosophila* lymph glands at a single-cell level, we dissected and dissociated lymph glands at three developmental timepoints, 72, 96, and 120 h AEL, and applied single cells to Drop-seq[27] (Fig. 1a). Fourteen independent sequencing libraries, representing 5 each for 72 and 96 h AEL and 4 for 120 h AEL, were prepared for scRNA-seq. We integrated the sequencing libraries after correcting for batch effects within and between timepoints using Seurat3[28]. Our quality-control pipeline eliminated cells with outlier unique molecular identifier (UMI) counts, low gene numbers, high mitochondrial gene contents, as well as doublets predicted by Scrublet (see "Methods" section). As a result, a total of 22,645 cells were retained for subsequent analyses (Supplementary Fig. 1a). The number of cells yielded 5.5 X, 6.8 X, and 2.4 X cell coverage of one lymph gland lobe at 72, 96, and 120 h AEL, respectively (Fig. 1b). The 22,645 high-quality cells exhibited a median of 6361 transcripts (UMIs) and 1477 genes per cell (Supplementary Tables 1–3; Supplementary Fig. 1b-c). In addition, the scRNA-seq libraries of individual timepoints corresponded well with genes detected in bulk RNA-seq, while undetected remainders displayed low expression levels (Supplementary Fig. 1d). Furthermore, we validated that the scRNA-seq libraries from 72, 96, or 120 h AEL align well with the pseudotime array of each library (Supplementary Fig. 1e). Altogether, our libraries appear sufficiently complex to reflect the whole transcriptome of developing hemocytes.

**Major cell types of *Drosophila* lymph gland hematopoiesis.** After validating the quality of the single cell libraries, we mapped the cells to the major zones of the lymph gland[16]. To explore the major cell types in the developing lymph gland, we aligned cells from the three developmental timepoints and clustered using the Louvain algorithm[29]. We aggregated cell clusters according to the expression of previously published marker genes by manual curation and identified seven distinctive groups of isolated hemocytes (Fig. 1c; Supplementary Fig. 1f; Supplementary Table 4). These clusters include prohemocytes, plasmatocytes, crystal cells, lamellocytes, PSC, and two additional clusters with uncharacterized genetic features. One cluster is enriched with glutathione-S-transferase transcripts that we named GST-rich (1.0%). The other cluster exhibits high expression levels of phagocytosis receptor (*crq, eater*), lipid metabolism-related (*LpR2, Lsd-2*), and starvation-induced (*Sirup*) genes. We referred to this cluster as adipohemocyte (1.4%) based on the name of similar hemocytes in other insects[30,31]. We verified the presence of GST-rich and adipohemocyte cell populations in wild-type lymph glands by confirming the expression of signature genes for these clusters in matched bulk RNA-seq (Supplementary Fig. 1g). In addition, we identified cells of the dorsal vessel as an extra non-hemocyte cell type based on previously identified marker genes for this tissue (Fig. 1c; Supplementary Fig. 1f).

Separation of clustered cells by developmental timepoints revealed that the relative population size of cell clusters changes as the lymph gland matures. Hemocytes in the lymph gland at 72

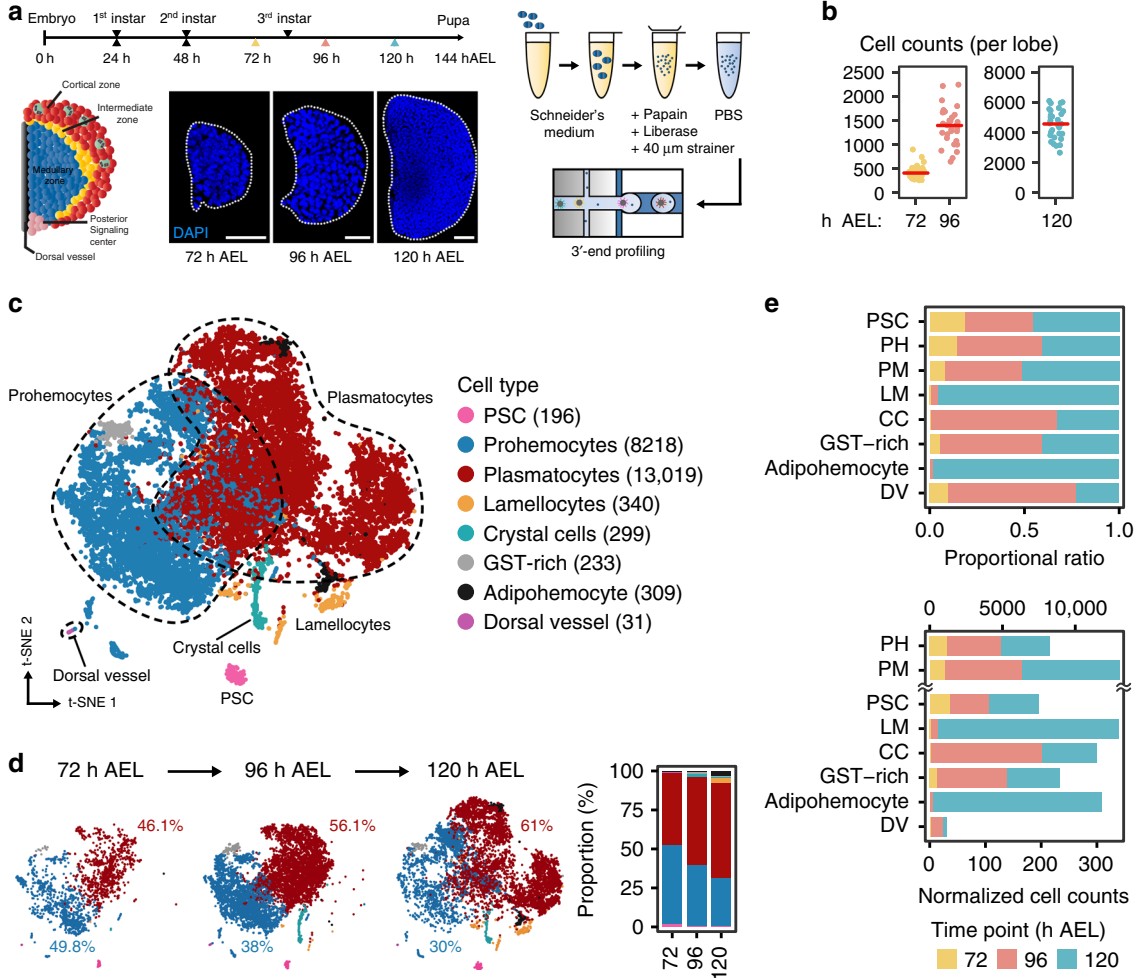

**Fig. 1 Major cell types identified in developing *Drosophila* lymph glands. a** A schematic diagram of the *Drosophila* lymph gland (left). Prohemocytes comprise the medullary zone at the inner core (blue) and give rise to mature hemocyte at the outermost area, called the cortical zone (red). Differentiating hemocytes in between the medullary zone and the cortical zone are termed the intermediate zone (yellow). The posterior signaling center (PSC) positions at the posterior end of the lymph gland (pink). *Drosophila* lymph glands (blue, DAPI) at three timepoints (72, 96, and 120 h AEL; After Egg Laying) (middle). Schematic workflow of sample preparation for scRNA-seq using Drop-seq (right). Scale bar, 30 μm. Lymph glands are demarcated by white dotted lines. **b** DAPI-positive cell counts of a single lymph gland lobe (n = 30 each for three timepoints). Red horizontal lines show median counts (397, 1392, and 4557 for 72, 96, and 120 h AEL, respectively). **c** A *t*-SNE plot showing the two-dimensional projection of eight major cell types identified in the scRNA-seq dataset (n = 22,645 in total; 72 h AEL, 2321; 96 h AEL, 9400; 120 h AEL, 10,924 cells). The count of each cell type is indicated in parentheses. Clusters are defined by following markers: Prohemocytes (PH: *Tep4, Ance*, 36.2%), Plasmatocytes (PM: *Hml, Pxn, NimC1*; 57.6%), Crystal cells (CC: *lz, PPO1, PPO2*; 1.3%), Lamellocytes (LM: *atilla*; 1.5%), the PSC (*Antp, col*; 0.9%), and Dorsal Vessel (DV: *Mlc, Hand*; 0.1%). Colors denote cell types. Dotted lines demarcate prohemocytes (blue) and plasmatocytes (red). **d** Two-dimensional projections of major cell types along developmental timepoints (left) and proportion of the cell types at each timepoint (right). Proportions of prohemocytes (blue) and plasmatocytes (red) are indicated. **e** Relative proportion (indicated as proportional ratio; top) and normalized cell counts (bottom) of each major cell type. Colors represent sampling timepoints.

h AEL are subdivided into two major groups—prohemocytes and plasmatocytes, with a virtually identical ratio of 49.8% and 46.1%, respectively (Fig. 1d, e). As the lymph gland matures, the proportion of plasmatocytes exceeds that of prohemocytes, and only 30% of the hemocytes retain the prohemocyte signature at 120 h AEL (Fig. 1d, e). This result is consistent with proportional changes of prohemocytes or plasmatocytes visualized by marker gene expression during development in vivo (Supplementary Fig. 1h). Different from plasmatocytes emerging from 72 h AEL, crystal cells and GST-rich cells first appear at 96 h AEL, and lamellocytes and adipohemocytes appear later at 120 h AEL (Fig. 1d, e). The PSC maintains constant cell numbers and relative ratios across lymph gland development (Fig. 1e). Due to temporal discrepancies in the emergence of mature hemocytes, we observed that most cells at 72 h AEL overlap well with cells in subsequent

timepoints, while cells at 96 and 120 h AEL segregate from those of preceding points (Supplementary Fig. 1i-j, see "Methods" section).

To better characterize the major cell types and transitions in gene regulatory networks during lymph gland development, we applied SCENIC[32]. We verified previously recognized transcriptional regulators in the lymph gland (Supplementary Fig. 1k)[19,24,33,34]. Moreover, we characterized transcription factors in well-known complexes or pathways in each cell type. In prohemocytes, we detected transcription factors of DREAM[35], including *E2F2* and *Dp*, and the Dpp pathway transcription factor *Mad* at 96 to 120 h AEL (Supplementary Fig. 1k). Plasmatocytes, on the other hand, utilize distinctive transcriptional regulators of the ecdysone pathway highlighted by *br, EcR, usp, Eip74EF*, and *Hr4*, and stress responsive genes

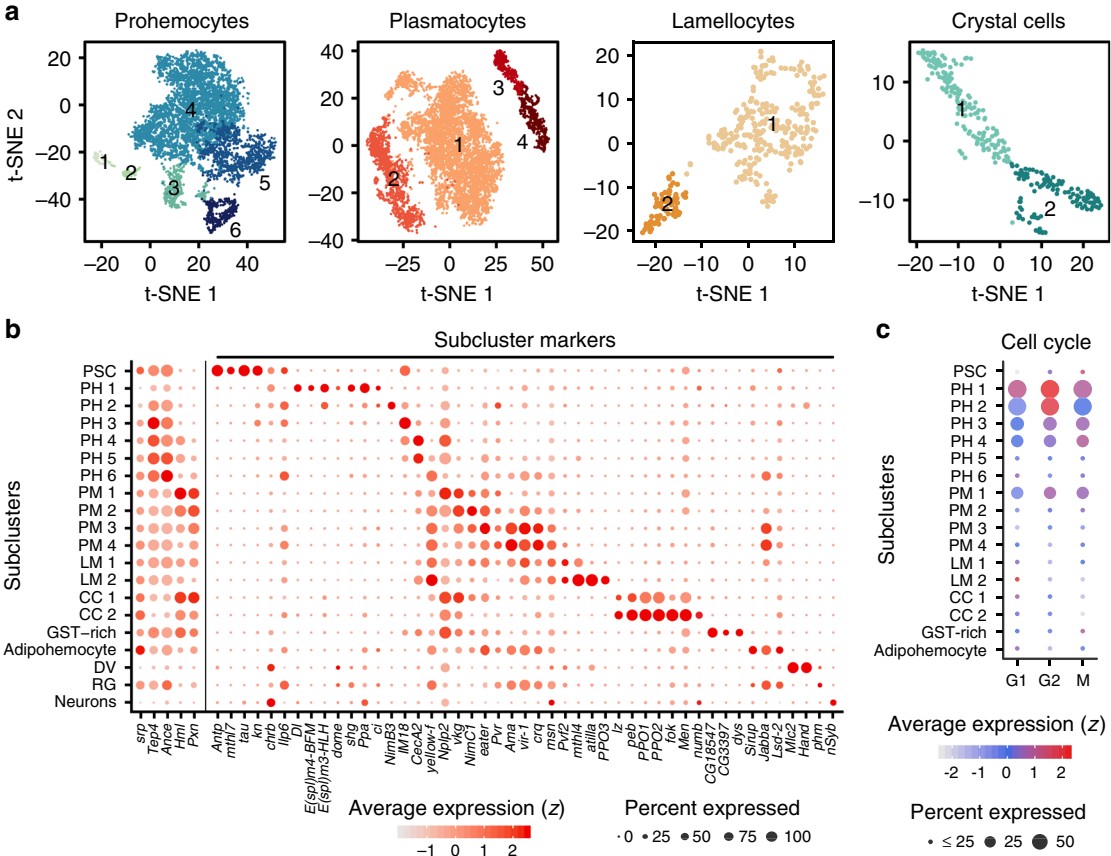

**Fig. 2 Subclustering analysis of hemocytes in the lymph gland. a** Subclusters of hemocytes—prohemocytes, plasmatocytes, lamellocyte, and crystal cells—are projected onto two-dimensional *t*-SNE plots. The numbers in the plots represent the subcluster number. **b** Dot plot presentation of significant gene sets in the 17 subclusters using Wilcoxon Rank-Sum test. 5 representative markers, *srp*, *Tep4*, *Ance*, *Hml*, and *Pxn* are indicated to the left column. Signature genes identified in this study are marked with subcluster markers. Dot color shows levels of average expression, and dot size represents the percentage of cells expressing the corresponding marker genes in each subcluster. Non-hematopoietic cell types are indicated (DV, dorsal vessel; RG, ring gland; Neurons). Cell count for each subcluster is listed in Supplementary Table 4. **c** Expressions of cell-cycle regulating genes in 17 subclusters. Dot color indicates average expression levels and dot size displays the percentage of cells with cell cycle controlling genes (*Cdk1*, *CycD*, and *CycE* for G1; *stg*, *CycA*, and *CycB* for G2; *polo*, *aurB*, and *Det* for M) in each subcluster.

such as *foxo* and *dl* (Supplementary Fig. 1k). Overall, our single-cell dataset of the entire lymph gland reliably reveals seven major types of hemocytes. Also, SCENIC analysis delineates transcriptional dynamics of the hemocytes and their regulators at the single-cell level.

**Heterogeneous populations of lymph gland hemocytes.** Our scRNA-seq data prompted us to further catalog the heterogeneity of primary cell types by performing unsupervised clustering. With subclustering analysis, we identified six subclusters of prohemocytes, four subclusters of plasmatocytes, and two subclusters each for lamellocytes and crystal cells (Fig. 2a; Supplementary Fig. 2a). We ensured that each subcluster contains cells from all libraries except stage-specific subsets (Supplementary Fig. 2b). When lymph gland cells at each developmental stage are individually clustered, the stage-specific subclustering and the combined subclustering show no differences (Supplementary Fig. 2c-d). In addition, we excluded non-hemocyte subclusters enriched with ring gland-specific or neuron-specific genes (Supplementary Fig. 2e-f). Interestingly, the PSC, adipohemocytes, and GST-rich clusters did not split into subclusters (Supplementary Fig. 2a-b).

The majority of prohemocyte subclusters is evenly represented at all timepoints and maintains high levels of *Tep4* and *Ance* throughout (Fig. 2b). Apart from their constant marker gene expression, we identified discrete fluctuations of cell-cycle

regulating genes within prohemocytes, as well as in PM1. Based on averaged levels of these genes, PH1 is likely to be in G1 or G2/ M; PH2 in G2; PH3 and PH4 in M; and PM1 in G2/M phase (Fig. 2c). Similar to prohemocytes, plasmatocytes exhibit gradual changes in plasmatocyte markers (Fig. 2b). PM1 maintains high *Hml* and *Pxn* with elevated levels of mature plasmatocyte markers, *vkg* and *NimC1* (Fig. 2b). PM3 and PM4 express characteristic signature genes such as *Ama*, *vir-1*, and *crq*, only detected at 120 h AEL (Fig. 2b; Supplementary Fig. 2b). Crystal cells are divided into two groups, CC1 and CC2 (Fig. 2a). CC1 expresses low levels of *lz* along with the expression of medullary zone and cortical zone markers. However, CC2 is devoid of the medullary zone or cortical zone markers and only displays high levels of *PPO1* and *PPO2*, suggesting that CC1 and CC2 correspond to early and mature crystal cells, respectively (Fig. 2a, b). Similarly, lamellocytes are separated into premature LM1 and mature LM2 reminiscent of the CC1 and CC2 clusters (Fig. 2a, b).

We next sought to validate the expression of subclusters in the lymph gland (Supplementary Data 1-2). In vivo, PH2 and PH3 express lower levels of *col* (col^low) than in the PSC and co-localize with *Tep4-gal4* (Supplementary Fig. 2g). Outer cells of col^low are *Tep4-gal4*/*Ance*^MiMiC single-positives and represent PH4 and PH5 (Supplementary Fig. 2g-h). At 120 h AEL, a few *Ilp6-gal4*+ cells appear within the *dome*+ medullary zone, representing putative PH6 (Supplementary Fig. 2i). In addition, we confirmed

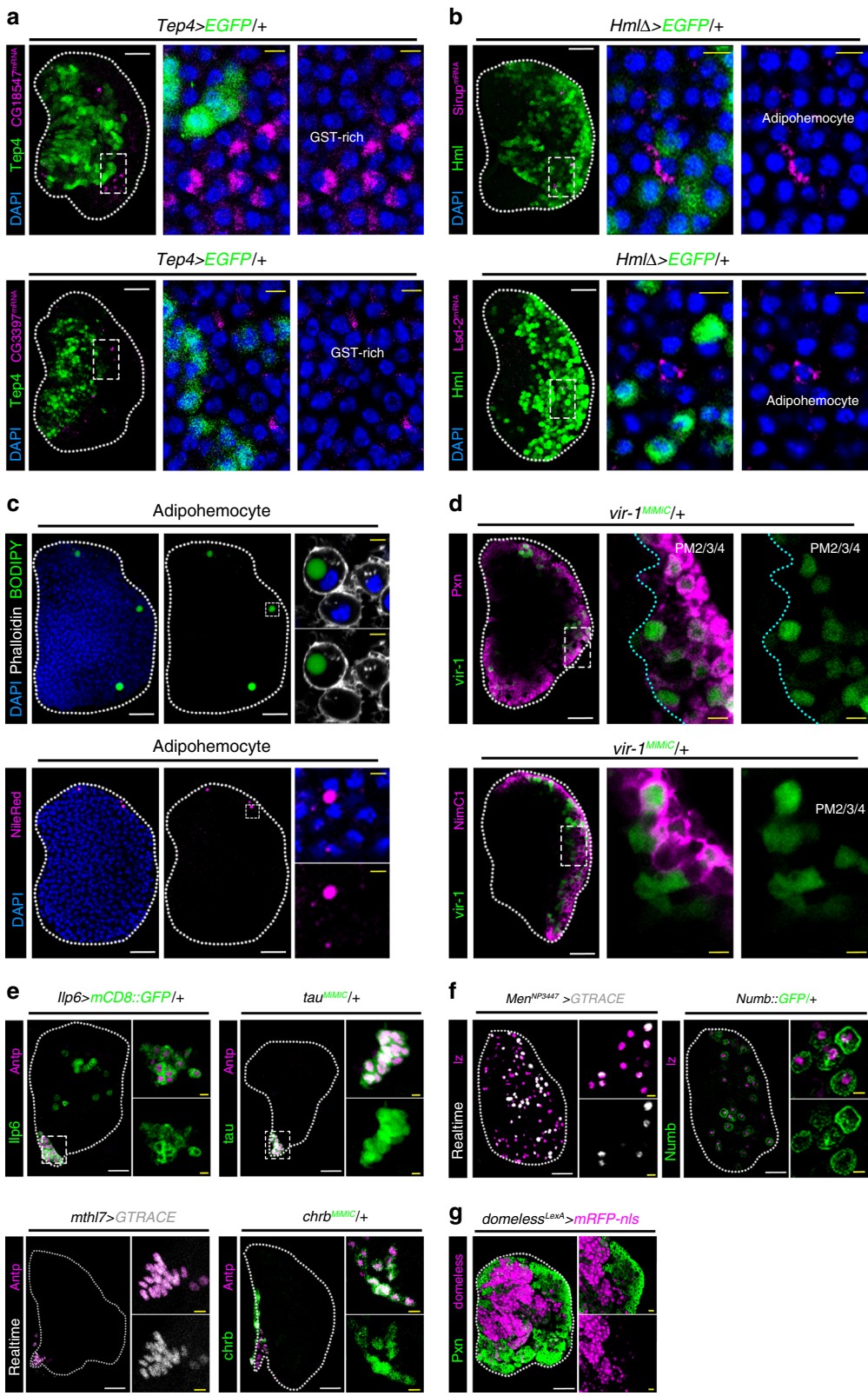

the endogenous expression of GST-rich and adipohemocyte subclusters. mRNA transcripts of *CG18547* or *CG3397*, two specific genes for the GST-rich cluster, are detected at the outer demarcation of *Tep4+* prohemocytes (Fig. 3a). Adipohemocytes represented by *Sirup* or *Lsd-2* are observed within the *Hml+* cortical zone (Fig. 3b). In addition to gene expression, we

discovered a few hemocytes containing neutral lipids (Fig. 3c), verifying the presence of adipohemocytes in the lymph gland. Finally, we visualized PM subclusters with *vir-1* and Pxn or NimC1 and established that a subset of *vir-1+* cells excludes Pxn or NimC1 expression analogous to the marker genes shown for PM3 or PM4 (Fig. 3d).

**Fig. 3 In vivo validation of subclusters and markers in the lymph gland. a** Expression of GST-rich-marker genes, *CG18547* and *CG3397*. mRNA transcripts of *CG18547* (top; magenta) or *CG3397* (bottom; magenta) are visualized at the outer demarcation of *Tep4* (green; left). Magnified images: *Tep4* and *CG18547* or *CG3397*. **b** Expression of adipohemocyte-specific marker genes, *Sirup* or *Lsd-2*. A few *Sirup*-expressing (top; magenta) or *Lsd-2* (bottom; magenta)-expressing cells are observed within the *Hml*+ (green) cortical zone. However, *Sirup*-positive or *Lsd-2*-positive hemocytes are devoid of *Hml*. Magnified images: *Hml* and *Sirup* or *Lsd-2*. **c** Lipid droplet-containing hemocytes in the lymph gland. When visualized with BODIPY (top; green) or Nile Red (bottom; magenta) lipid probes, a few hemocytes (Phalloidin, white) at the cortical zone hold a lipid droplet. Magnified images: actin structures (Phalloidin, white), a lipid droplet, and DAPI. **d** Plasmatocyte marker, *vir-1* (green; *vir-1*MiMiC) partially co-localizes with *Pxn*+ (top; magenta) or NimC1 (bottom; magenta). Magnified images: *vir-1* and *Pxn* or NimC1. **e** PSC markers in the lymph gland. Antp+ PSC cells (magenta) co-localize with *Ilp6* (green; *Ilp6-gal4* UAS-mCD8::GFP), *tau* (green; *tau*MiMiC), *mthl7* (white; *mthl7-gal4* UAS-GTRACE), or *chrb* (green; *chrb*MiMiC). *Ilp6-gal4* is expressed in other cell types including PH6 and crystal cells (Supplementary Fig. 2i). Magnified images: Antp and *Ilp6*, *tau*, *mthl7*, or *chrb*. **f** Crystal cell markers in the lymph gland. lz+ crystal cells (magenta) co-localized with *Men* (white; *Men-gal4* UAS-GTRACE; left) or *Numb* (green; *Numb*::GFP; right). lz co-localizes with a subset of *Men* while all the *Numb*+ and lz+ overlap. Magnified images: lz and *Men* or *Numb*. **g** *domeless-LexA* generated in this study (magenta; *domeless-LexA* LexAop-mRFP-nls) partially co-localizes with Pxn (green). Magnified images: *domeless* and Pxn. White scale bar, 30 µm; yellow scale bar, 3 µm. Lymph gland is demarcated by white dotted line. Blue indicates DAPI. The dotted box in **a–g** indicates a magnified region. Each panel at the right displays magnified views.

As an effort to augment current genetic tools, we generated or screened for markers showing specific gene expression in the lymph gland. We confirmed the expression of *Ilp6*, *tau*, *mthl7*, and *chrb* in the PSC, and *Men* and *Numb* in crystal cells (Fig. 2b; Supplementary Data 3–5; Fig. 3e, f). In addition to markers widely used[36], we uncovered reporter lines for representative markers such as *Ance*MiMiC and *dome-LexA* (Supplementary Data 3–5; Supplementary Fig. 2h; Fig. 3g). Together, we classified 17 transcriptionally distinctive subtypes of hemocytes in the developing lymph gland and assigned functional descriptions of each subset based on gene expression patterns. *Bona fide* markers elucidated in each subcluster collectively provide a valuable resource for further understanding of myeloid-like hemocyte development.

**Trajectory reconstitution and functional networks**. Next, we investigated the time sequence of lymph gland hematopoiesis by reconstruction of developmental trajectories using Monocle3[37]. For the trajectory analysis, we excluded PSC as PSC cells do not give rise to the rest of lymph gland hemocytes (Supplementary Fig. 3a)[19,24], and we set the PH1 subcluster as the start point based on the expression of *Notch*, *shg*, and high levels of mitotic genes (Fig. 2b, c). Pseudotime reconstitution of lymph gland hemocytes displays the main trajectory from prohemocytes to plasmatocytes along with divergent sub-trajectories towards crystal cells, adipohemocytes, GST-rich, and lamellocytes (Fig. 4a, b; Supplementary Movie 1). The trajectory corresponds well with on-and-off patterns of marker genes in the lymph gland (Supplementary Fig. 3b). Moreover, there is an excellent correlation between the real-time and the pseudotime trajectories when compared with segregated real-time hemocyte transcriptomes (Fig. 4b, c; Supplementary Fig. 3c).

In the major trajectory, we observed a linear trajectory from PH1 to PH3, projecting towards diverse differentiating states of prohemocytes including PH4-PH5 and GST-rich (Fig. 4a; Supplementary Fig. 3d-e). As an auxiliary route, PH6 is distinguishable from PH1-PH4 and gives rise to PM3, PM4, adipohemocytes, or lamellocytes at 120 h AEL (Fig. 4a, b; Supplementary Fig. 3d-e). In the later sequence, all the prohemocyte subclusters merge into PM1 in the trajectory (Supplementary Fig. 3d-e). PM1 accounts for the majority of plasmatocytes and a minor subset of PM1 starts to display *NimC1* expression (Supplementary Fig. 3b-d). A branch is observed following PM1, producing separable paths towards either the plasmatocyte PM2 or the crystal cell lineage (Supplementary Fig. 3d-e). We also observed the coupling of cell division and differentiation. In addition to the prohemocyte subclusters, PM1 exhibits high levels of cell cycle genes and appears immediately before a divergence to CC1 (Fig. 2c; Supplementary Fig. 3d-e).

To address the functional characteristics of the hematopoietic trajectory and associated subclusters, we examined gene-expression modules to determine whether subclusters share similar gene expression modules (Supplementary Fig. 3f; Supplementary Data 6). Strikingly, prohemocyte subclusters exhibit translation-related, metabolism-related, and developmental-related gene expression whereas plasmatocyte subclusters show relatively high levels of extracellular matrix and immune responsive genes. Crystal cell subclusters display high levels of genes involved in small molecule metabolism, and adipohemocyte and GST-rich subclusters show glyoxylate metabolism and DNA damage responsive gene modules, respectively (Supplementary Fig. 3f).

We next focused on the transition of prohemocytes into mature hemocytes and defined subclusters spanning the intermediate zone according to the trajectory analysis and modular configurations. Given that PH4, GST-rich, and PM1 subclusters exhibit moderate levels of *Tep4*, *Ance*, *Pxn*, and *Hml* while restricting mature hemocyte markers, as described in previous studies[17], we hypothesized that PH4, GST-rich, and PM1 correspond to intermediate cell types in transition prior to final differentiation. These subclusters are found at all timepoints (Supplementary Fig. 2b). We scored highly enriched genes in the potential intermediate subclusters and noticed that expression of *Nplp2*, a novel apolipoprotein[38], peaks at PH4, GST-rich, PM1, and CC1 and is lower in other subclusters (Fig. 2b). Visualizing *Nplp2-gal4*[39] in the lymph gland revealed a partial overlap of *Nplp2* with the medullary zone marker, *dome*Meso, or the cortical zone marker, *Hml* (Fig. 4d). Moreover, *Nplp2-gal4* expressing cells are exclusive of col^low but overlap with *Ance*MiMiC+ cells, demonstrating that *Nplp2* emerges subsequent to PH2/PH3 (Supplementary Fig. 3g-h). Furthermore, *Nplp2* is expressed independently of the late plasmatocyte marker, NimC1 (Supplementary Fig. 3i), indicating that *Nplp2* is expressed in the intermediate zone of the lymph gland, which corresponds to hemocytes transitioning towards ultimate differentiation. In addition, we verified that the cell cycle profile of *Nplp2-gal4*+ cells significantly differs from that of *Tep4-gal4* (Supplementary Fig. 3j). Altogether, the pseudotime trajectory analysis provides a detailed basis for prohemocyte differentiation. Also, we demonstrate the presence of transitioning subclusters, previously described as the intermediate zone, and their endogenous gene expression.

**Initiation of hematopoiesis in the lymph gland**. To investigate the primordial cell types during lymph gland hematopoiesis, we focused on the earliest PHs—PH1 and PH2—which initiate the entire trajectory. We observed that the majority of PH1 is detached from PH2/PH3 and a subset is connected to PH2 in the

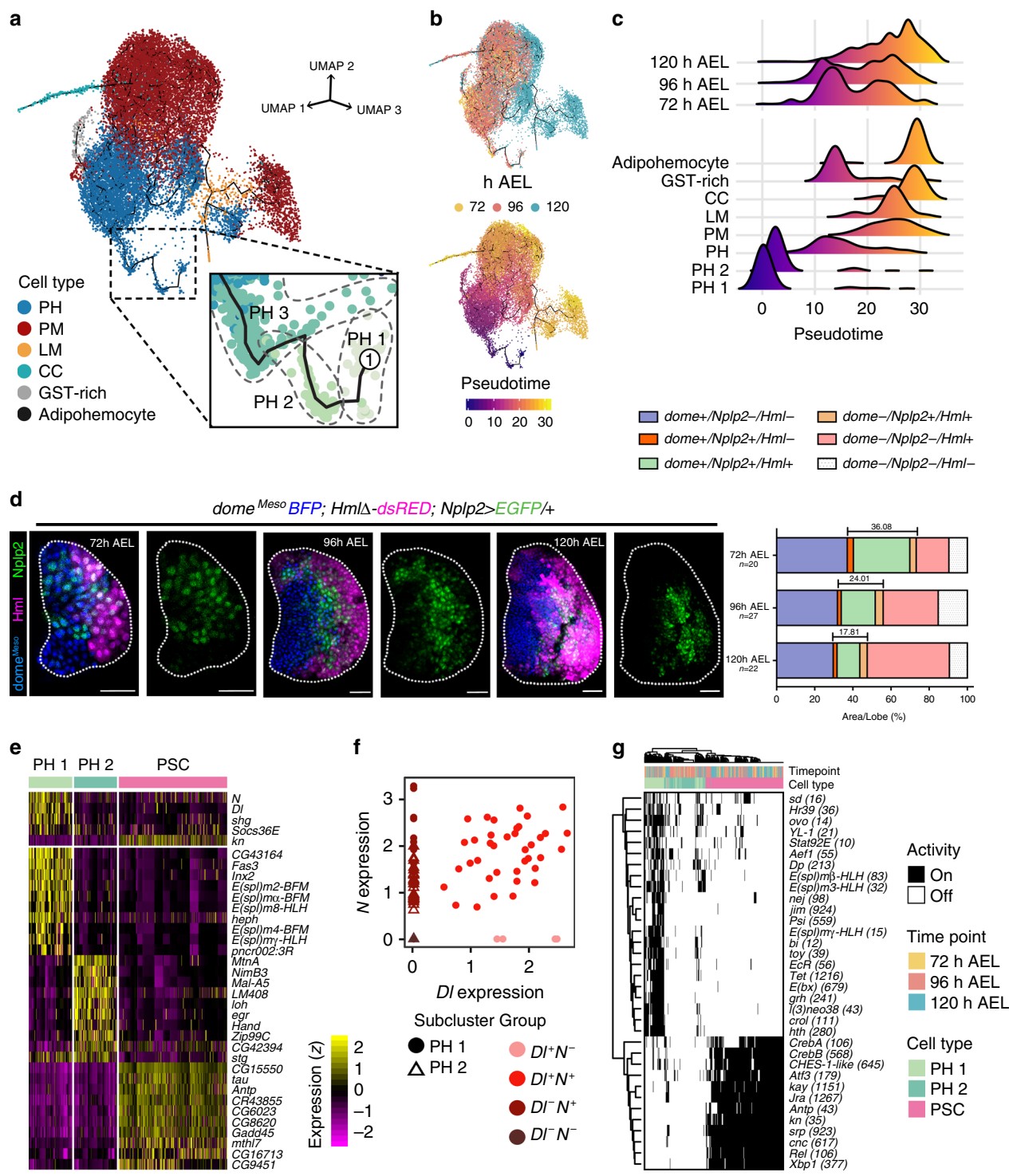

trajectory map (Fig. 4a; Supplementary Fig. 3b-d). Though PH1 and PH2 mark the earliest pseudotime, both clusters are found at all developmental timepoints (Fig. 4b, c; Supplementary Fig. 2b). We identified multiple signature genes in the PH1 subcluster (Figs. 2b, 4e). First, we discovered that *Notch* (*N*), its ligand, *Delta* (*Dl*), and the *E(spl)* family genes, downstream targets of the Notch pathway, are expressed in PH1 (Fig. 4e; Supplementary Fig. 4a). Interestingly, cells in PH1 and PH2 are in regular sequence according to on-and-off patterns of *Dl* and *N* (Fig. 4f; Supplementary Fig. 4b). Second, we found associations of MAPK

and Notch pathways with *Dl⁺N⁺* cells of PH1 by KEGG pathway analysis (Supplementary Fig. 4b). Third, we observed levels of *dome*, *hop*, *Stat92E*, and *Socs36E* in PH1, reflecting active JAK/STAT signaling (Fig. 4e; Supplementary Fig. 4a-b). Strikingly, the expression of Notch/Delta and JAK/STAT-related genes in PH1 decrease in the succeeding PH2, suggesting that PH1 cells undergo a drastic change. Fourth, we identified that PH1 does not express *col* while PH2 exhibits low levels of *col* expression, consistent with previous observations (Supplementary Fig. 4c)[20]. Lastly, high levels of cell cycle genes are detected in both PH1 and

**Fig. 4 Pseudotime trajectory analysis of lymph gland hematopoiesis. a** A three-dimensional landscape of the lymph gland hematopoiesis trajectory using Monocle3 ($n = 19,143$). Non-hematopoietic cells were excluded. Black line indicates the trajectory. Colors indicate the six major cell types used. The inset shows the three ancestral PH subclusters, PH1, PH2, and PH3. **b** Trajectories re-drawn by developmental timepoints (top) and calculated pseudotime (bottom). Colors indicate the three real-timepoints (top) and pseudotime (bottom). **c** Relative densities of hemocytes segregated by three timepoints (top) and cell types (bottom) along pseudotime. PH1 and PH2 are separated from other PH subclusters for higher resolution. Colors in density plots correspond to pseudotime, as in **b**. **d** Nplp2-gal4 (dome^Meso-EBFP; Nplp2-gal4 UAS-EGFP; HmlΔ-dsRED) at 72, 96, or 120 h AEL lymph gland. Nplp2-gal4 (green) partially overlaps with dome^Meso (blue), Hml (magenta) or both. The proportion of Nplp2-gal4 expressing cells diminishes over development (36.08% at 72 h; 24.01% at 96 h; 17.81% at 120 h AEL). Graph indicates quantitations of the combinational proportions of dome^Meso+ or dome^Meso−, Nplp2+ or Nplp2−, and Hml+ or Hml− cells in one lymph gland lobe at 72 ($n = 20$), 96 ($n = 27$), or 120 ($n = 22$) h AEL. The bracket indicates the proportions of Nplp2-gal4. White scale bar, 30 μm. White dotted line demarcates the lymph gland. **e** Heatmap representation of the 35 signature genes identified in PH1, PH2, and the PSC ($n = 77, 79,$ and 189 cells, respectively). The colored legend denotes the standardized level of the genes. **f** Four subgroups in PH1 and PH2 defined by the expression of Delta (Dl) and Notch (N). Colors show subgroups and shapes specify PH subclusters. X axis means Dl expression; Y axis, N expression. **g** Binary heatmap showing the activity of transcription factors in PH1, PH2, and the PSC predicted by SCENIC. Numbers in parentheses denote the count of downstream genes used to test the activity of transcription factors.

PH2, constituting the earliest PH subclusters actively proliferating in the lymph gland (Fig. 2c).

Next, we applied SCENIC to further establish gene regulatory networks of PH1 and PH2 cell populations with PSC as a comparison. SCENIC analysis on the PSC successfully proved the activity of known transcription factors including Antp[15] and col[18] (Fig. 4g). Moreover, we found that the PSC exhibits transcriptional activities of CrebA/B, Atf3, Jra/kay, and Xbp1 (Fig. 4g). The PH1 subcluster shows an entirely distinct set of transcription factors: sd in Hippo pathway, Stat92E in JAK/STAT pathway, E (bx), jim, and YL-1 in chromatin remodeling, EcR and crol in ecdysone pathway, the E(spl) family in Notch pathway, and Psi in Myc transcription (Fig. 4g).

We next performed spatial reconstructions for PH1 in vivo and profiled the expression of marker genes and related reporter lines newly identified in the subset. Interestingly, Stat92E::edGFP is expressed in the cells neighboring the PSC, which are neither Tep4+ medullary zone nor Antp+ PSC (Fig. 5a; Supplementary Fig. 4d-e). Similarly, Stat92E^Act cells show close contact with col+ cells without having apparent overlaps (Fig. 5b). Stat92E^Act cells incorporate EdU well as indicated in the analysis of cell cycle genes (Fig. 2c; Supplementary Fig. 4f). The number of Stat92E^Act cells increases over development but maintain relatively constant ratios (Fig. 5c). Furthermore, Stat92E^Act cells disappear upon genetic ablation of the PSC, which indicates that expression of Stat92E^Act PH1 is dependent upon the PSC (Fig. 5d). Yet, loss of upd3 in the PSC does not alter the expression of Stat92E^Act cells, suggesting that there is an alternative ligand for the STAT92E activation in PH1 (Supplementary Fig. 4g). We additionally detected Dl mRNA or Dl protein expression near the PSC similar to the pattern observed with Stat92E::edGFP (Fig. 5e; Supplementary Fig. 4h). The majority of Dl+ cells are Stat92E^Act; however, Dl covers a range broader than a few cell diameters than Stat92E^Act (Fig. 5f). Further, a few Dl-expressing cells exhibit co-localization with Su(H)-GBE coinciding with the Dl-N interaction shown in the computational analysis (Supplementary Fig. 4i). Consistent with the phenomenon observed with Stat92^Act cells, we observed that the loss of PSC significantly attenuates Dl expression (Supplementary Fig. 4j). We screened Gal4 lines of signature genes in PH1 and identified a Dl enhancer-trap that marks Antp− cells adjoining the PSC, which produce three representative types of hemocytes and consequently, the entire lymph gland (Fig. 5g; Supplementary Fig. 4k). To summarize, PH1, the initial subset of the pseudotime trajectory, indicates a subpopulation of prohemocytes that physically interacts with the PSC and is adjacent to col+ PH2. PH1 cells do not co-localize with conventional medullary zone or cortical zone markers but exhibit distinctive gene regulatory networks primarily steered by the Delta/Notch and JAK/STAT pathways. Moreover, these cells

retain potentials to give rise to hemocytes in the lymph gland during 72 to 120 h AEL (Fig. 5h). Thus, we conclude that PH1 is a precursor of prohemocytes reminiscent of mammalian hematopoietic stem cells (HSCs).

**Differentiation of lymph gland hemocytes upon wasp infestation.** We next investigated emerging heterogeneity and differentiation of the lymph gland hemocytes upon wasp infestation by scoring lymph gland hemocytes at 24 h post infestation (PI). At 24 h PI, lymph gland hemocytes remain in place, ultimately disintegrating before 48 h PI (Fig. 6a). We aligned and matched lymph gland hemocytes from 24 h PI (96 h AEL, $n = 10,158$) to control hemocytes using label transfer, that resulted in the annotation of six hemocyte clusters and 11 subclusters when compared to those from controls (Fig. 6b–e; Supplementary Table 5). Consistent with previous studies[24,40], wasp infestation significantly reduces crystal cells—iCC1 and iCC2 (Fig. 6c, e). A similar decline is readily observed in iPH1, which is confirmed by the reduction of Stat92E+ or Dl+ iPH1 in the lymph gland upon wasp infestation (Fig. 6e, f). In contrast, iPH4 shows an increase in numbers and relative proportions, implying an expansion of differentiating prohemocytes upon wasp infestation (Fig. 6e). Coinciding with the increase of differentiating cells, the lamellocyte and GST-rich populations, which are barely observed during normal development at 96 h AEL, expand upon wasp infestation (Fig. 6b–e). Lamellocytes derived upon infestation (iLMs) are subclustered into two groups: iLM1 and iLM2, which represent immature and mature iLMs, respectively (Fig. 6g). While other cell types undergo significant modifications upon wasp infestation, we did not detect any changes in the expression and number of iPSC upon wasp infestation (Fig. 6b–e, h). When the signature genes of each subcluster are compared to those from controls, gene expression patterns in general are not altered (Fig. 6h). However, some of the subclusters, iPH4 and iPM1 in particular, significantly enhance their cell cycle genes (Fig. 6h). These data indicate that active immunity causes a biased commitment of prohemocytes and plasmatocytes to the lamellocyte lineage and remodels the proportions of hemocyte populations.

To better understand how iLM differentiates in the lymph gland, we performed pseudotime trajectory analysis and examined gene modules of related subtypes. From the trajectory analysis, we discovered that the emergence of iLMs is associated with both iPHs and iPMs (Fig. 7a–d; Supplementary Data 7). The majority of iLMs are directly derived from iPH4 and iPM1 (route 1-2 in Fig. 7d), subclusters indicated as the intermediate cell populations. In addition, an alternative route is generated from a subset of iPM1 that expresses NimC1 (NimC1+ PM1) (route 3 in Fig. 7d), the most mature plasmatocyte subcluster at 96 h AEL

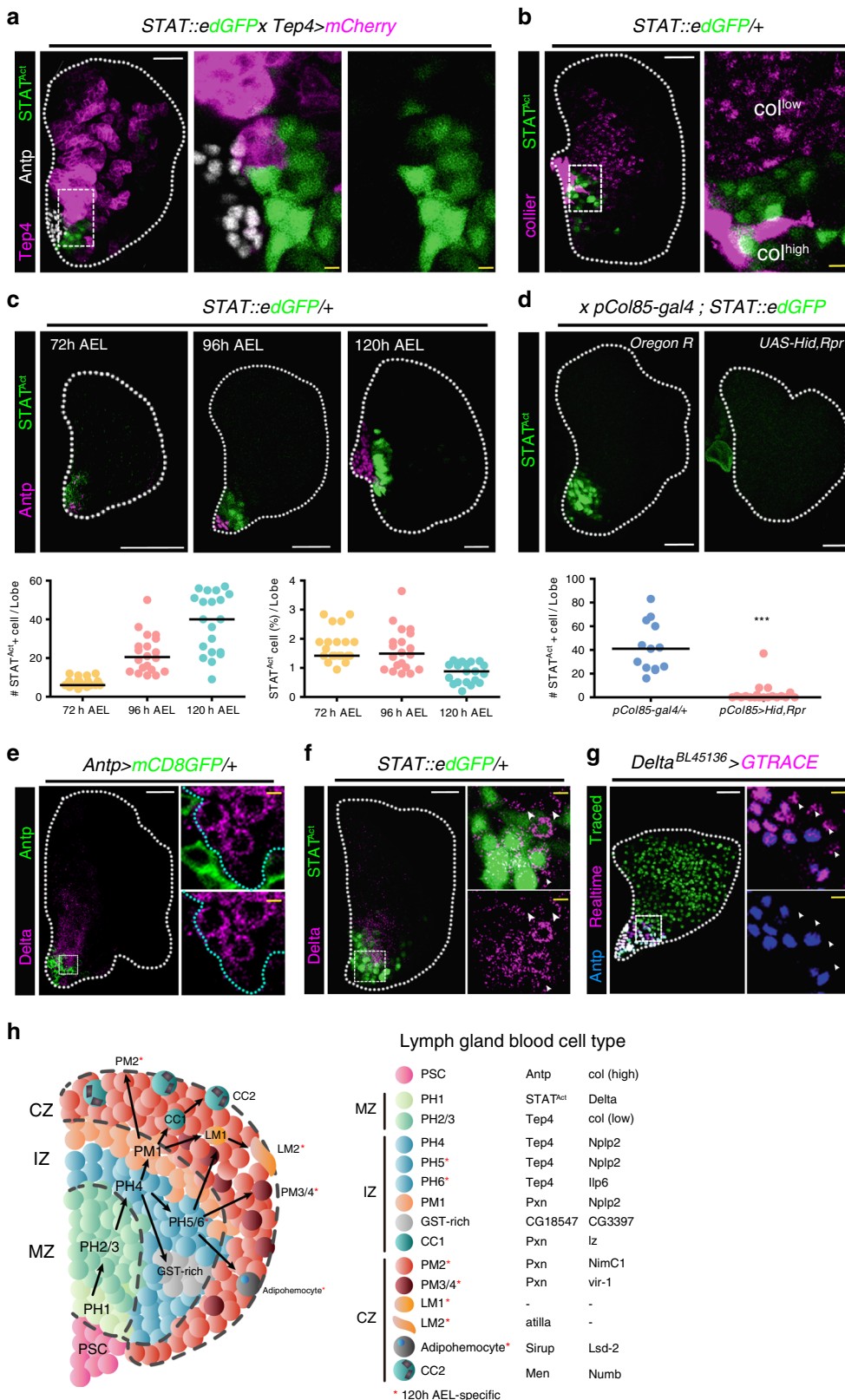

**h** Lymph gland blood cell type

| | | | |
|---|---|---|---|
| ● PSC | | Antp | col (high) |
| ● PH1 | MZ | STAT^Act | Delta |
| ● PH2/3 | | Tep4 | col (low) |
| ● PH4 | | Tep4 | Nplp2 |
| ● PH5* | | Tep4 | Nplp2 |
| ● PH6* | | Tep4 | Ilp6 |
| ● PM1 | IZ | Pxn | Nplp2 |
| ● GST-rich | | CG18547 | CG3397 |
| ● CC1 | | Pxn | lz |
| ● PM2* | | Pxn | NimC1 |
| ● PM3/4* | | Pxn | vir-1 |
| ● LM1* | CZ | - | - |
| ● LM2* | | atilla | - |
| ● Adipohemocyte* | | Sirup | Lsd-2 |
| ● CC2 | | Men | Numb |

* 120h AEL-specific

(Fig. 7b). We validated the data by tracing the intermediate zone or the cortical zone markers upon wasp infestation and confirmed that atilla⁺ iLMs are derived from either *Nplp2*⁺ intermediate hemocytes or *Hml*⁺ plasmatocytes (Fig. 7e). However, differentiating crystal cells and lamellocyte lineages are mutually exclusive (Fig. 7f). An association of gene-expression modules of each subcluster indicates the existence of two distinct trajectories to iLMs: iPH4-to-iLM and *NimC1*⁺ iPM1-to-iLM (Fig. 7d, right). The first iPH4-to-iLM wave is enriched with genes involved in hemocyte proliferation, DNA replication, and translation. The

**Fig. 5 Expression of PH1 in the lymph gland. a** $STAT92E^{Act}$ (green) and $Tep4^+$ (magenta) or $Antp^+$ (white) cells are mutually exclusive (*Tep4-gal4 UAS-mCherry; STAT92E::edGFP*). Magnified images: $STAT92E^{Act}$ and $Tep4^+$ cells near $Antp^+$ PSC. **b** $STAT92E^{Act}$ (green) do not co-localize with cells expressing high (PSC) or low (PHs) levels of *collier* (magenta). Magnified images: $STAT92E^{Act}$ and $col^+$ cells near the PSC. **c**. The number of $STAT92E^{Act}$ cells (green) increases during lymph gland development (72 ($n = 24$), 96 ($n = 17$), and 120 h ($n = 20$) AEL). $Antp^+$ (magenta) and $STAT92E^{Act}$ are separable at all timepoints. Graphs represent quantitation of the number (left) or the proportion (right) of $STAT92E^{Act}$ cells in one lymph gland lobe. **d**. Genetic ablation of the PSC (*pCol85-gal4; STAT92E::edGFP UAS-Hid, Rpr*) attenuates $STAT92E^{Act}$ (green) expression (top). Graph indicates quantitations of the number of $STAT92E^{Act}$ cells in one lymph gland lobe (bottom, Mann–Whitney test, ***$P < 0.0001$). *pCol85-gal4/+* ($n = 13$) or *pCol85-gal4 UAS-Hid, Rpr* ($n = 19$). **e** $Dl^+$ cells (magenta) are localized adjacent to the PSC (Antp, green). Magnified images: $Dl^+$ and $Antp^+$ cells. Cyan dotted lines demarcate $Dl^+$ cells. **f** $Dl^+$ cells (magenta) co-localize with $STAT92E^{Act}$ (green). Magnified view: $Dl^+$ and $STAT92E^{Act}$ cells (right). A few $Dl^+$ cells that do not express $STAT92E::edGFP$ are indicated (arrowhead). **g** Lineage tracing of $Dl^+$ cells (green, traced; magenta, real-time; blue, Antp). $Delta^{BL45136}$*-gal4 UAS-GTRACE* covers the entire lymph gland. Arrowheads represent $Dl^+$ cells next to the $Antp^+$ $Dl^+$ PSC. **h** Model of the lymph gland hematopoiesis. PH1 cells are adjacent to the PSC. PH1 and PSC or PH2 are mutually exclusive. There are multiple states of prohemocytes including GST-rich and intermediary PHs/PMs. Plasmatocytes represent a heterogeneous cell population including adipohemocytes and late plasmatocytes, PM3-4. Colors indicate subclusters. Putative subclusters in the medullary zone (MZ), the cortical zone (CZ), or the intermediate zone (IZ) are described (right). Subcluster-specific markers verified in this study are listed (right). Red asterisks indicate 120 h AEL-specific subclusters. White scale bar, 30 µm; yellow scale bar, 3 µm. White dotted line demarcates the lymph gland. The dotted box (**a–b**, **e–g**) indicates a magnified region. Each panel at the right displays magnified views. Median value is represented in graphs (**c**, **d**).

second $NimC1^+$ iPM1-to-iLM wave expresses metabolism-related genes, demonstrating at least two modes of iLM development in the lymph gland upon wasp infestation. Overall, we established that the lymph gland responds to wasp infestation by an expansion of differentiating hemocytes accompanied by subsequent differentiation of iLMs. In addition to the differentiation of intermediate populations indicated as iPH4-to-iLM in the trajectory, mature plasmatocytes, $NimC1^+$ iPM1, transdifferentiate into iLMs as an alternative route amplifying the magnitude of iLM formation.

**Comparison between two hematopoietic lineages**. To distinguish and compare hemocytes originating from two lineages[3], we compared the larval circulating hemocyte datasets[41] ($n = 3397$; Supplementary Table 6) sequenced using inDrops and Drop-seq to explore lineage-specific features of *Drosophila* hemocytes at 96 and 120 h AEL (see "Methods" section). As a result, we found that hemocytes from the lymph gland significantly overlap with those from circulation (Fig. 8a, b; Supplementary Fig. 5a). We then transferred subcluster labels of lymph gland hemocytes to circulating hemocytes, and recognized three common cell types: prohemocytes, plasmatocytes, and crystal cells, all of which consisted of five subclusters in circulation (Fig. 8b; Supplementary Fig. 5b; Supplementary Data 8). Lamellocytes, adipohemocytes, and the PSC are exclusively found in the lymph gland (Fig. 8b). Interestingly, 120 prohemocytes are identified in circulation, and 65 of them are labeled as PH1 with unique markers despite the absence of Notch and its downstream components (Fig. 8c; Supplementary Fig. 5b-c). A small number of PH4 is detected in circulation ($n = 55$) and displays subtle differences compared to the lymph gland (Supplementary Fig. 5c). Plasmatocytes in circulation share similarities only to PM1 of the lymph gland, implying the dominant plasticity of circulating PMs (Supplementary Fig. 5b-c). In addition, crystal cells in the lymph gland and in circulation are nearly identical except for a few genes (Supplementary Fig. 5c).

Next, we explored the collective signature gene expression of lymph gland and circulating hemocytes based on the relative expression levels. Besides the genes depicted due to uneven distribution of hemocytes from the two origins (Supplementary Fig. 5b), we identified lineage-enriched genes including *Ultrabithorax* (*Ubx*) (Fig. 8c; Supplementary Fig. 5a, c). When each subcluster was individually compared, circulating plasmatocytes display *Ubx* expression (Supplementary Fig. 5c-e). Similar differences are observed in crystal cells: *Pde1c*, *CAH2*, and *Naxd* are higher in circulating crystal cells whereas *Arc2*, *Oscillin*, *aay*,

and *fbp* are significantly higher in crystal cells from the lymph gland (Supplementary Fig. 5c). Taken together, hemocytes generated from the two independent lineages appear to be predominantly similar. At the same time, there are sufficiently distinctive variances in the levels of lineage-biased gene expressions by which we can distinguish their ancestries.

## Discussion

In this study, we report a comprehensive single-cell transcriptome analysis of 29,490 developing and immune-responsive myeloid-like hemocytes in *Drosophila* lymph glands. Our analysis provides insights into: (1) the development of hemocytes at the single-cell level, (2) the expression of hemocyte subpopulations including hematopoietic stem-like cells and adipohemocytes, (3) the differentiation mechanisms of myeloid-like hemocytes upon active immunity, and (4) similarities and differences between hemocytes derived from two distinct hematopoietic ancestries: embryo and larva. This study describes myeloid-like cells in *Drosophila* and their developmental trajectories at a single-cell level.

In this study, we used Drop-seq to build libraries from dissociated cells of the lymph gland. Given that hemocytes are highly susceptible to multiple stresses, we suspect that cell dissociation process might stress the lymph gland hemocytes. This is apparent in the unexpectedly high number of lamellocytes detected in the scRNA-seq (Fig. 1c). Generally, wild-type lymph glands rarely produce lamellocytes[1]. These lamellocytes plausibly differentiated following dissociation-induced stress response[42]. Alternatively, the number under normal condition may be undervalued due to the lack of markers to identify early lamellocytes. Another possibility is that lamellocytes, that are usually larger than other hemocytes, are better captured than other hemocytes in the Drop-seq. Nonetheless, to reduce the stress-induced bias, we introduced additional measures in our analyses, including the comparison of the single-cell transcriptome with bulk RNA-seq (see "Methods" section). As a result, our scRNA-seq datasets faithfully display single-cell transcriptomes of all known cell types, as well as two other previously undescribed cell types. GST-rich cells, enriched with ROS-responsive and DNA damage genes, emerge during prohemocyte development. Considering that genes enriched in GST-rich cells are also evident in the lymph gland bulk RNA-seq and GST-rich-specific marker genes are detected in wild-type lymph glands, this population cannot be considered a consequence of stressed hemocytes. Rather, this subtype may represent a state that prohemocytes experience during development, or may play an active role in ROS-mediated or

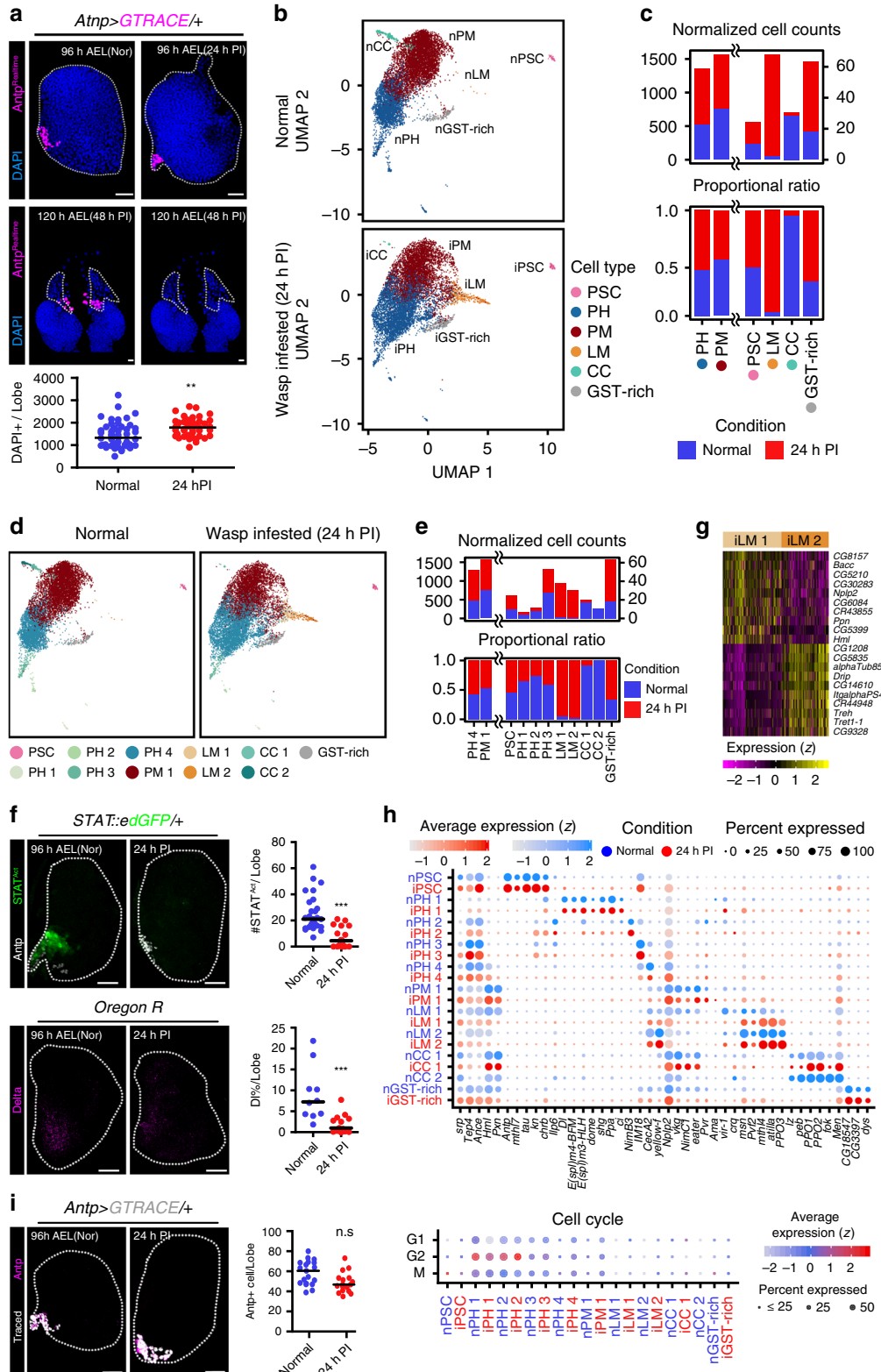

GABA-mediated stress responses[43,44]. Adipohemocytes, on the other hand, share hallmarks of both mature plasmatocytes and lipid metabolism, appearing only at 120 h AEL of the lymph gland. Macrophages in vertebrates readily take up lipids and lipoproteins, and accumulation of lipid-containing macrophages, called foam cells, is highlighted in various pathological conditions[45]. In *Drosophila*, the presence of lipid-containing hemocytes

has not been reported. Given our results, and that adipohemocytes are frequently observed in insects, including *Aedes aegypti*[30,31], it is possible that flies also conserve metabolism-oriented hemocytes to coordinate immunity and metabolism.

Prohemocytes have been widely considered to represent a uniform cell population based on the expression of marker genes, *domeless* or *Tep4*. However, recent studies have suggested their

**Fig. 6 Comparison of normal and wasp infested lymph glands. a** Lymph gland (*Antp-gal4 UAS-GTRACE*; Antp, magenta) remains at 24 h post infestation (PI) (DAPI, blue). Hemocytes disintegrate by 48 h PI. Graph indicates the number of DAPI$^+$ cells at 24 h PI. Normal ($n = 51$), 24 h PI ($n = 44$). Mann-Whitney test, **$P = 0.0011$. **b** UMAP projections of major cell types defined in normal (top, 'n') and wasp infested (bottom, 'i') lymph glands at 96 h AEL (thus, 24 h PI). **c** Relative proportion (top) or normalized cell counts (bottom) of major cell types in normal (blue) and wasp infested (red) lymph glands. **d** Two-dimensional projections of subclustered cells in normal (left) and wasp infested (right) lymph glands. **e** Normalized cell counts (top) or proportional ratio (bottom) of subclustered cells in normal (blue) or wasp infested (red) lymph glands. **f**. Wasp infestation reduces *STAT92E*$^{Act}$ (top; green) or Dl$^+$ (bottom; magenta) PH1 populations. Graphs represent quantitation of the number of *STAT92E*$^{Act}$ (top) or the proportion of Dl$^+$ (bottom) cells after infestation (24 h PI). Normal ($n = 27$) or 24 h PI ($n = 18$) for Top and Normal ($n = 10$), or 24 h PI ($n = 13$). Mann–Whitney test, ***$P < 0.0001$. **g** Top 20 genes in iLM1 or iLM2. Color bar indicates the level of scaled gene expression. **h** Dot plot expresses the level of marker genes in normal (blue) or wasp infested (red) lymph glands (top) using Wilcoxon Rank-Sum test. Scaled expressions of cell-cycle regulating genes in subclusters upon wasp infestation (bottom). Dot color: average expression levels. Dot size: percentage of cells expressing cell cycle genes in a subcluster. Cell count for each subcluster is listed in Supplementary Table 5. **i** PSC cell numbers do not change upon wasp infestation by 24 h PI (Antp, magenta; Traced, white; *Antp-gal4 UAS-GTRACE*). Graph indicates the number of Antp$^+$ cells. Normal ($n = 19$) or infested ($n = 17$). Mann–Whitney test, n.s.: not significant. White scale bar, 30 μm; Yellow scale bar, 3 μm. Lymph glands are demarcated by white dotted lines. Median value is shown in graphs. Colors in **b** and **d** indicate each cell type. Graphs indicate the number of cells in one lymph gland lobe.

heterogeneity based on uneven expressions of cell cycle markers or bifurcated col expressions[46,47]. In support of these studies, our unbiased subclustering of primary clusters identified different statuses of prohemocytes. First, prohemocytes differ in the expression of cell cycle regulators, implying an asynchrony of prohemocyte development and their states. This observation also accounts for the stochastic cell cycle patterns visualized with the UAS-FUCCI system[46]. Second, we observed dynamic expression patterns of development-related or DNA replication-related and proliferation-related genes in early or late prohemocytes, respectively. In addition, the 120 h AEL-specific PH6 denotes unique pathways including steroid biosynthesis-related genes. Lastly, the presence of prohemocytes with more differentiated states is also indicative of their dynamics. Although the presence of the intermediate zone has been recognized[17,43], the biological significance of various intermediary states and the previously undescribed functions of endogenous genes including *Nplp2* in these subclusters require further investigations.

As the most naïve subcluster identified in this study, PH1, demarcates a group of cells that has not been annotated by previous markers such as *Tep4*, *Antp* or *col*. Discovery of the hidden cell population–PH1, will shed light on understanding the hierarchy of prohemocyte differentiation and enhance the relevance of the lymph gland as a hematopoietic model. Roles for *Notch*, *Stat92E*, or *scalloped* in the earliest state of prohemocytes have been previously suggested by recent studies[18,48–50]. Moreover, clonal analyses have shown that cells adjacent to the PSC generate the largest population in the lymph gland[51]. These studies are consistent with our hypothesis that Notch/Delta and JAK/STAT$^+$ cells nearby the PSC sustain latent capacities to produce the entire lymph gland hemocytes. Since our analyses focus on the second to the third instar lymph gland, it will be important to further delineate an ancestor of PH1 and understand its developmental association with Notch$^+$ cells described in the first-instar lymph gland[49].

Comparative analyses on wild-type and wasp-infested lymph glands revealed that wasp infestation exerts a biased differentiation of hemocytes to the lamellocyte lineage. By 24 h PI, lymph glands physically remain in place; yet, cells within lymph glands undergo a dynamic differentiation towards early lamellocytes. Interestingly, iPH1 and iPH2 significantly reduce their numbers at 24 h PI while amplifying iGST-rich, iPH4, iLM1, and iLM2. The depletion of early prohemocytes and augmentation of the following subclusters could be tightly associated with the PSC considering its role in the PH1 maintenance. An expansion of iPH4 provides a designated pool for the iLMs and could be critical to sufficiently meeting the high demand for immune cells upon wasp parasitism similar to the circulation[52]. In addition to iPH4 committed to lamellocyte differentiation, *NimC1*$^+$ PM1

transdifferentiates into lamellocytes, revealing two independent routes for lamellocyte differentiation. Unlike other hemocytes, the iPSC remains stationary in its number and transcriptome profile. Hence, PSC may likely function through a post-transcriptional modification in controlling the immune response[20,24,50,53].

## Methods

***Drosophila* stocks and genetics**. The following *Drosophila* stocks were used in this study: *dome*$^{Meso}$*-EBFP2* (U.Banerjee)[36], *HmlΔ-gal4* (S.Sinenko)[54], *dome*$^{Meso}$*-gal4* (M.Zeidler)[55], *Tep4-gal4* (NIG, Japan)[56], *Nplp2-gal4* (KDRC, South Korea)[39], *pCol85-gal4* (M.Crozatier)[18], *Delta-gal4* (BL45136), *Antp-gal4* (U.Banerjee)[19], *Stat92E::edGFP* (N.Perrimon)[57], *HmlΔ-DsRed* (K.Brueckner)[58], *vir-1*$^{MiMiC}$ (BL34135), *Ance*$^{MiMiC}$ (BL59828), *UAS-Hid, Rpr* (Nambu JR)[59], *UAS-GTRACE* (C. Evans)[60], *nSyb-gal4* (BL51635), *Ilp6-gal4* (A.Brand)[61], *tau*$^{MiMiC}$ (BL37602), *mthl7-gal4* (generated in this study), *chrb*$^{MiMiC}$ (BL53813), *Men-gal4* (NIG, 113708), *Numb::GFP* (F.Schweisguth)[62], *UAS-EGFP* (BL5428), *UAS-mCD8GFP* (BL5137), *Ubx RNAi* (VDRC 37823), *lz-gal4*$^{DBD}$; *Pxn-gal4*$^{AD}$ and *dome-LexA* (generated in this study), *LexAop-mRFP-nls* (BL29956), *upd3 RNAi* (VDRC106869), *upd2Δupd3Δ* (BL55729), *upd3Δ* (BL55728), and *UAS-FUCCI* (BL55111)[63] and stocks examined are listed in Supplementary Data 3–4.

Fly stocks used in this study were maintained at 25 °C. *Oregon R* was used for the scRNA-seq as a wild type. Unless indicated, crossed flies were maintained at 25 °C with dextrose-cornmeal based normal food.

To synchronize larval stages, one hundred adult flies were kept on grape-juice agar plate for two hours. Hatched larvae were discarded at 23 h after egg laying (AEL), and those at 24 h AEL were collected and reared on normal corn-meal yeast media. To screen the gal4 lines in this study, we crossed each gal4 strain with *UAS-GTRACE* and identified those expressed in the lymph gland. To avoid stress conditions generated by crowding, we reared less than 50 larvae in one vial.

**Generation of fly stocks**. To generate gal4 fly lines, fly genomic DNA was amplified by primers indicated in Supplementary Data 3. Amplified genomic regions were ligated into *pAGal4+* (KDRC) or TOPO-TA vector (Invitrogen; K250020) for Gateway cloning. *pBPnlsLexAp65Uw* (Addgene 26230), *pBPZpGAL4DBDUw* (Addgene; 26233) or *pBPp65ADZpUw* (Addgene; 26234) was used as destination vector. Transgenic flies were generated by KDRC, South Korea.

**Dissociation of lymph glands into single cells**. One hundred to one hundred and fifty lymph glands were dissected at 72, 96, or 120 h AEL, respectively, in Schneider's medium (Gibco, 21720024). Dorsal vessel, ring gland and posterior lobes were detached from the primary lobe of the lymph gland; only the primary lobes of the lymph glands were used in this study. Primary lobes were kept in 200 μl ice-cooled Schneider's medium during dissection. After, centrifugation at 3000 rpm and 4 °C for 5 min was done. Supernatant was discarded and 300 μl of room temperature Schneider's medium was added to the lymph gland primary lobes. 300 μl of Papain (Worthington, LK003178) pre-heated to 37 °C, and 4.1 μl of Liberase TM (Roche, 5401119001) were added and gently mixed. Samples were incubated for 20 min with gentle agitation. At 5-min, 10-min, and 15-min timepoints of incubation, samples were mixed using 200p pipette. Enzymes were inactivated with 100 μl of ice-cooled Schneider's medium, and samples were kept on ice. Suspended cells were passed through a 40 μm cell strainer (Corning, 352340). Afterwards, centrifugation at 3000 rpm and 4 °C for 5 min was done. The supernatant was discarded, and 1× filtered sterile PBS was added to cells. The final concentration of cells was fixed to 300 cells per 1 μl for a total of 600 μl.

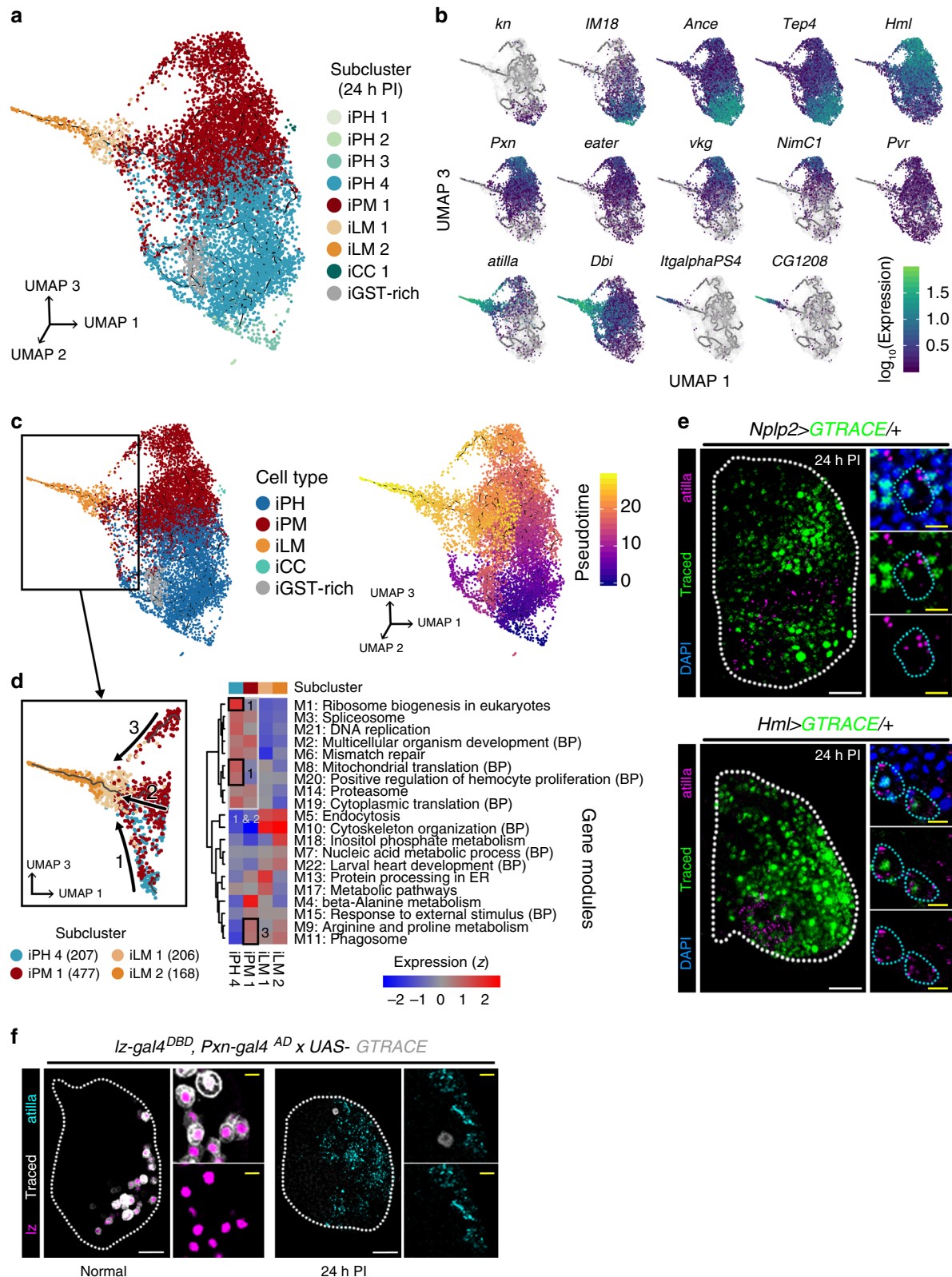

**Bulk RNA-seq of the lymph gland**. Procedures prior to the Papain treatment were performed to collect intact lymph glands at 72, 96, and 120 h AEL. Instead of Papain solution, TRIzol (MRC, TR118) was added to 100–150 lymph glands and RNA extraction was performed. More than 1 μg of RNA was prepared for each experiment. A cDNA library was constructed by 5′ and 3′ adapter ligation and loaded into a flow cell where fragments are captured into library adapters. Each fragment was then amplified through bridge amplification. After cluster generation

was completed, templates were sequenced by Illumina TruSeq. A single RNA-seq library was prepared from each developmental stage.

**Drop-seq and scRNA-seq**. All the Drop-seq and cDNA synthesis methods followed a previous study[27]. The concentration of beads was fixed to 300 beads per

**Fig. 7 Lymph gland hematopoiesis following wasp infestation. a** Three-dimensional pseudotime trajectory of subclustered cells from the normal lymph gland dataset. Colors indicate cell types. **b** Expression patterns of 14 marker genes– *kn* (*col*), *IM18, Ance, Tep4, Hml, Pxn, eater, vkg, NimC1, Pvr, atilla, Dbi, ItgalphaPS4*, and *CG1208*–in two-dimensional (UMAP 1 and 3) pseudotime trajectory. Color bar denotes the $\log_{10}$-scaled level of gene expression. Gray color indicates no expression. **c** A three-dimensional trajectory landscape of major cell types under wasp infestation (left), and additional representation of trajectory over calculated pseudotime using Monocle3 (right). Box indicates the cells used for subtrajectory analysis in **d**. Colors in legends show pseudotime (right). **d** Subtrajectory analysis of four subclusters—iPH4, iPM1, iLM1, and iLM2—detected in the trajectory to iLM (left). Two different waves, arrow 1/2 (iPH4/NimC1[−] iPM1, thus, intermediate cells) and arrow 3 (NimC1[+] iPM1), advance towards iLM with distinct gene modules (right). Shared gene modules between iPH or iPM with iLM are indicated in boxes. Colored scale represents the *z*-transformed mean expression level of gene modules. **e** Lamellocytes differentiate from intermediate iPHs (*Nplp2-gal4 UAS-GTRACE*) or iPMs (*Hml-gal4 UAS-GTRACE*) upon wasp infestation. *Nplp2[+]* iPHs (green, traced; blue, DAPI; top) or *Hml[+]* iPMs (green, traced; blue, DAPI; bottom) express atilla (magenta) upon wasp infestation. Insets indicate magnified views of atilla[+] cells. Cyan dotted lines within insets demarcate traced iLMs. **f** iLM and iCC lineages are separable. iLMs emerge independently of iCCs (*lz-gal4[DBD], Pxn-gal4[AD] UAS-GTRACE*). *lz-gal4[DBD], Pxn-gal4[AD] UAS-GTRACE* trace the majority of *lz*-expressing crystal cells (lz, magenta; traced, white). However, atilla[+] iLMs are *lz[DBD]/Pxn[AD]*-negative (atilla, cyan; traced, white). White scale bar, 30 μm; yellow scale bar, 3 μm. Lymph glands are demarcated by white dotted lines. Magnified images are shown on the right of each panel.

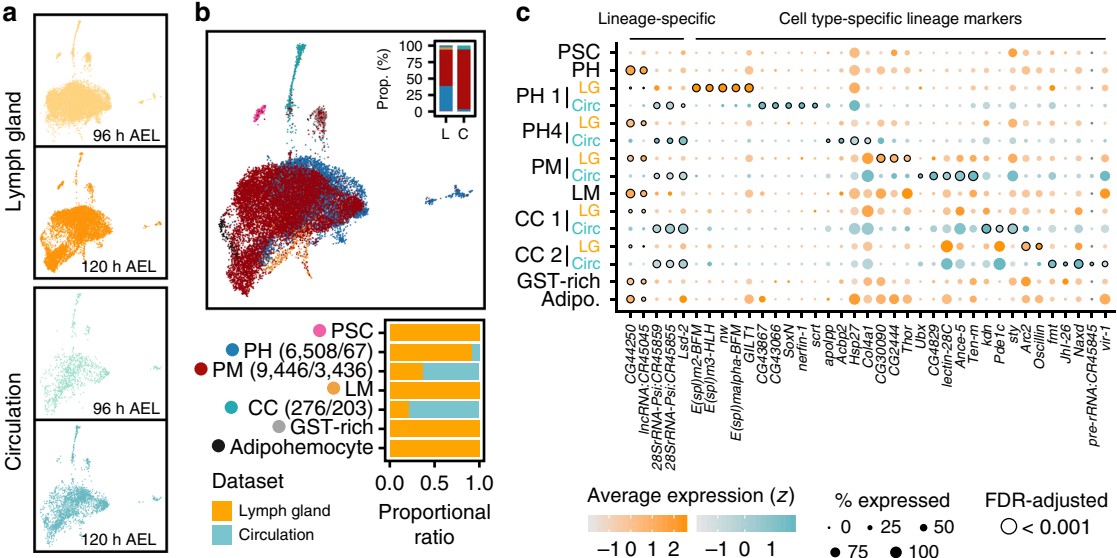

**Fig. 8 Transcriptome-wide comparisons between embryonic and larval lymph gland hemocytes in *Drosophila*. a** Two-dimensional projections of hemocytes in the lymph gland (top) and circulation (bottom) at 96 and 120 h AEL. **b** Combined projection (top) and proportions of major cell types (bottom) in the lymph gland (yellow) and in circulation (cyan). Inset (top) shows the ratio of major cell types between the lymph gland (L) and in circulation (C). **c** Dot plot of marker genes highly enriched in a lineage-specific or cell type-specific manner. The colors show the origin of the datasets (yellow, lymph gland; cyan, circulation). Significant marker genes with FDR-adjusted *P* < 0.001 were outlined.

1 μl. Around 10 min Drop-seq run was performed for each experiment. After cDNA synthesis, scRNA-seq was performed using Illumina NextSeq.

**Preprocessing and mapping of scRNA-seq data**. Raw scRNA-seq data were generated in paired-end reads following single-cell capture using Drop-seq: one end included a barcode and unique molecular identifier (UMI) sequences in 20 nucleotides (12 and 8 nts, respectively), and the other end, cDNA in 50 nts. The preprocessing and mapping of scRNA-seq data produced in this study followed the Drop-seq Core Computational Protocol version 1.2 (January 2016) and corresponding Drop-seq tools version 1.13 (December 2017) provided by the McCarroll Lab (http://mccarrolllab.org/dropseq/).

First, the reference genome (Fasta, BDGP 6.02) and transcriptome annotations (gtf, September 2014) required for the processing were downloaded from the Ensembl website (http://asia.ensembl.org/). Additional dictionary and refFlat files were generated using *picard* (*CreateSequenceDictionary*) and *ConvertToRefFlat* provided in the Drop-seq tools package, respectively. These reference data were prepared with the same prefix and stored in a single directory for later use. Simplified command lines are as follows:

Generation of a dictionary file:

*java -jar<path to Drop-seq tools picard>/picard.jar CreateSequenceDictionary R=<genome fasta> O=<output dictionary>*

Generation of a refFlat file:

*<path to Drop-seq tools>/ConvertToRefFlat \*
*ANNOTATIONS_FILE=<gft annotation> \*
*SEQUENCE_DICTIONARY=<dictionary file> O=<output refFlat>*

Once all the reference data was prepared, paired-end fastq files were converted to the bam format using *picard FastqToSam*.

*java -jar <path to Drop-seq tools picard>/picard.jar FastqToSam \*
*F1=<fastq 1> F2=<fastq 2> O=<output bam> SM=<library number>*

The unaligned bam files were subjected to the *Drop-seq_alignment.sh* script for alignment to genome. This shell script is a single pipeline that executes detection of barcode and UMI sequences, filtration and trimming of low-quality bases and adaptors or poly-A tails, and alignment of reads using *STAR* (2.5.3a).

*<path to Drop-seq tools>/Drop-seq_alignment.sh \*
*-g <path to STAR index> -r <genome fasta> -n <# of cells expected> \*
*-d <path to Drop-seq tools> -s <path to STAR> \*
*-o<path to output> -t<path to temporary output> -p<unaligned bam file>*

**Selection of cells by the total mapped reads**. To extract the number of cells having proper read counts, the aligned bam files generated from the previous section were summarized using *BAMTagHistogram* in the Drop-seq tools package. This program extracts the number of aligned reads per cell barcode which is subsequently used to plot the cumulative distribution of reads.

*<path to Drop-seq tools> / BAMTagHistogram I=<aligned bam> O=<output file> TAG=XC*

Cumulative read distribution plots were then explored, and the number of cells were inferred where a sharp decrease (referred as 'knee' by the author's documentation) in a slope occurs. The inferred cell number was determined as a minimal threshold number of aligned reads per cell for cell selection. To summarize, a minimum of 30,000 reads per cell for 72 h AEL library 6, 15,000 per cell for 72 h AEL library 3, 10,000 per cell for 72 h AEL libraries 1, 2, and 4; 5000

per cell for 96 h AEL libraries 1, 3, 5, all 120 h AEL libraries, and infested 96 h AEL libraries 1, 2, and 4; 4000 per cell for infested 96 h AEL 2, 3000 per cell for 96 h AEL library 4 and infested 96 h AEL library 3; 2000 per cell for 92 h AEL library 2 were chosen as thresholds. *DigitalExpression* provides UMI count matrix (selected cells by genes) using a mapped bam file and the minimum number of reads per cell as following.

    <path to Drop-seq tools>/DigitalExpression \
    I=<aligned bam> MIN_NUM_READS_PER_CELL=<read count threshold> \
    O=<output read count matrix> SUMMARY=<output summary>

The resulting output per library is written into a file with a name where a corresponding library number was added as a suffix to each barcode sequence with an AEL timepoint (e.g., barcode-72-1 or barcode-96-2) to avoid collision of barcode sequences between libraries. The number of expressed genes between libraries or sampling timepoints may vary because each library would have a different number of captured cells, different cell types, or uneven sequencing depth. In total, at least 13,612, 13,523, 14,277, and 13,658 genes, and 2505, 10,027, 11,702, and 10,939 cells were detected in one library at 72, 96, 120, or infested 96 h AEL, respectively. So, we used a union set of genes (15,540 genes) to merge three normal lymph gland datasets.

**scRNA-seq data analysis using Seurat 3.0.** Seurat is a universal software for scRNA-seq analyses including preprocessing, cell clustering, and dimension reductions. The current version (v3.0) of Seurat features dimension reduction using UMAP, integration of datasets produced with different modalities or conditions, and transfer of cell labels between datasets[28]. Detailed analyses steps are explained on the Seurat website (https://satijalab.org/seurat/), so we only describe the schematic workflow used in this study.

First, each library was filtered for low-quality cells, separately, by setting thresholds for UMI and gene counts. We used 5000 genes as an upper threshold and 400 genes as a lower threshold for normal lymph gland libraries and infested lymph gland library 1 and 2, and 200 genes as a lower threshold for other infested lymph gland libraries. Then we also filtered cells having UMI counts higher than two standard deviations from the mean UMI count to exclude multiplets. After filtration, 2399, 9496, 11,081, and 10,461 cells remained for 72, 96, 120, and infested 96 h AEL, respectively. All libraries in each sampling timepoint (h AEL) were merged and normalized, and cells expressing mitochondrial genes higher than 10% of total UMI count were removed (2322, 9411, 10,976, and 10,179 cells remained). After high mitochondrial cells were excluded, multiplets were inferred using Scrublet[64], which simulates artificial doublets using given expression matrix to predict multiplet artifacts. One, 11, and 52 cells were further filtered out from 72, 96, and 120 h AEL lymph glands, respectively. However, no cells were detected from the infested dataset. As a result, 2321 (72 h AEL), 9400 (96 h AEL), 10,924 (120 h AEL), and 10,179 cells (infested 96 h AEL) were subjected to downstream analyses.

Next, for the integration of cells from normal lymph glands at three timepoints (normal lymph gland integration), cells were aligned using *FindIntegrationAnchors ()* and *IntegrateData()* with default parameters, respectively. UMI counts were normalized, log-transformed, and scaled to properly integrate datasets, and 52 principle components (PCs) were used to explain the variability of the scaled UMI counts across cells. *t*-distributed stochastic neighbor embedding (*t*-SNE)[65] and UMAP plots were then generated using the selected numbers of PCs. Clustering was performed with resolution of 0.8 to get 19 clusters. Then again, clusters were aggregated to get broad cell types based on expression of known marker genes (Supplementary Fig. 1f). In summary, six and seven clusters were merged as collective PH and PM, respectively, to define the following six major cell types: PSC, LM, CC, DV, GST-rich, and adipohemocyte.

When the integrated normal lymph gland dataset was examined, we found that subclusters of PH and PM cells solely originated from 120 h AEL (Fig. 1d and Supplementary Fig. 1i). Thus, we analyzed our dataset using a different batch normalization method to test whether this trend is independent of our analysis strategy. For this, we corrected the sequencing library variable using *ScaleData* function along with the UMI count. We then selected the number of PCs to use (50 PCs) and performed t-SNE and UMAP analyses again. Similarly, a number of cells from 120 h AEL were separated from others while cell types, such as PSC, LM, or CC were mixed together (Supplementary Fig. 1j). An R package, *rgl*, was used for all 3-dimensional plots presented in this study.

**Pseudo-bulk RNA-seq trajectory analysis.** Normal lymph gland scRNA-seq libraries were examined to see whether they could be aligned in a single trajectory line as actual development timepoints. As we produced at least four sequencing libraries for each timepoint from independent sample preparations and sequencing, we performed trajectory analysis in pseudo-bulk RNA-seq samples, pooling all cells from each library. For this, valid cell barcodes identified in the previous analysis were collected and their UMI count matrices were retrieved. All UMI count values were then aggregated by genes to generate pseudo-bulk RNA-seq data. We applied Monocle 2[66] (http://cole-trapnell-lab.github.io/monocle-release/) for this trajectory analysis using 2758 highly variable genes or top 500 most differentially expressed (DE) genes out of 7596 genes (expressing more than one UMI in at least three cells) with default parameters. The highly variable genes were selected with criteria of *dispersion_empirical>=1 * dispersion_fit & mean_expression>=1*. In both analyses,

the results were similar, so the trajectory using variable genes was presented (Supplementary Fig. 1e).

**Initial subclustering analysis.** To exclude comtaminated cells, doublet-like cells in subclusters, and unintended cells that originate from only a single library, each of eight major cell types was subclustered separately using Seurat (Initial subclustering). For this, (1) cells designated to each cell type were retrieved from the integrated Seurat object, (2) their UMI counts were scaled and normalized for sequencing library and total UMI counts, and (3) the number of PCs were determined for dimension reduction and clustering analysis. With this procedure, we found that initial subclusters: 13 subclusters for PH (res = 0.8), 14 for PM (res = 0.9), two for LM (res = 0.3), three for CC (res = 0.1), and another three for adipohemocytes (res = 0.3) were detected. Other cell types (PSC, GST-rich, and DV) were not further clustered. Next, subclusters mainly originating from a single library (more than half) were removed from the datasets because they may have resulted from experimental artefacts. Two subclusters each for PH and adipohemocyte, three and one subclusters were excluded from PM, and CC, correspondingly (Supplementary Fig. 2b).

When the results were examined using known marker genes, one PM subcluster (PM11) that displayed a high level of ring gland marker gene, *phm*, was excluded from subsequent analyses. Particularly, of 11 remaining PH subclusters, one PH subcluster (PH1) displayed a high level of genes related to the Notch signaling pathway. We thus sought to investigate these PH1 and PH2 subclusters in more detail, along with PSC cells which are known to maintain early prohemocytes in lymph glands. For this, we reclustered PH1, PH2, and PSC cells together, and found 6 subclusters. However, one of these subclusters appeared to be neuronal cells as they expressed specific neuronal markers, such as *nSyb* or *Syt1* (Supplementary Fig. 2f). On the other hand, three PSC subclusters were detected but merged because they all shared the same signature genes (Fig. 4e). In addition, PH cells—annotated as PSC and 3 PSC cells—that comingled with PH1 or 2 in the subclustering analysis were removed, as their identities were inconsistent.

**Optimal subclustering analysis.** We noticed a number of PH or PM subclusters in our initial subclustering result lacked distinctive signature genes and expressed high level of genes related to immune response. To mitigate hemocyte activation signals could be induced by library preparation and to merge similar subclusters, we excluded genes highly upregulated or downregulated in scRNA-seq data compared to bulk-RNA-seq (≥10-fold; 3103 and 1727 genes, respectively), and reclustered PH or PM cells without contaminating neurons or ring gland. In this time, we hypothesized that an optimal resolution value would result in the maximal number of DEGs between subclusters after similar subclusters (Pearson r ≥ 0.95) were merged to form a super-group. To test this, we ranged resolution values from 0.1 to 3.0, increasing by 0.1, to make clusters and transformed UMI count matrix to pseudo-bulk and calculated correlation matrix using 2000 variable genes. We next iteratively correlated and excluded the least correlated subcluster (Pearson r < 0.90) until a minimum correlation coefficient exceeds 0.9. Then we again iteratively correlated and aggregated two most correlated subclusters (Pearson r ≥ 0.95) into a single super-group until the maximum correlation coefficient is lower than 0.95. Using our strategy, we identified six prohemocyte and four plasmatocyte subclusters. PH1 and 2 remained almost same as the previous result, while others except for two 120 h AEL specific subclusters were merged into a pan-PH group PH4. Majority of PM subclusters were grouped into a pan-PM group PM1 except for three 120 h AEL specific PM subclusters.

In summary, we defined a total of 17 subgroups from the normal lymph gland dataset (Fig. 2 and Supplementary Fig. 2); 6 for PH, 4 for PM, two each for LM and CC, one each for PSC, GST-rich, and adipohemocyte, along with three non-hematopoietic cell types (DV, RG, and Neurons).

**Cell cycling scores.** To calculate cell cycle scores, expression of cell cycle genes (*Cdk1*, *CycD*, and *CycE* for G1; *stg*, *CycA*, and *CycB* for G2; *polo*, *aurB*, and *Det* for M phase) were averaged for each cell. Then, cells expressing less than a third quartile (<0.784) of cell cycle scores (G1 + G2 + M) were filtered out and remaining cells were assigned to their original subclusters. We next constrained subclusters having cells more than 25% of their original population size to visualize only highly proliferative subclusters (Fig. 2c).

**Trajectory analysis using monocle 3.** To reconstruct lymph gland hematopoiesis in *Drosophila*, hematopoietic cells in the blood lineage were collected after filtering the PSC, DV, RG, and neuron subclusters (*n* = 19,143; Supplementary Table 4). We then followed the Monocle 3[37] analysis pipeline described in the website documentation (https://cole-trapnell-lab.github.io/monocle3/) using custom parameters predetermined from repetitive analyses. We normalized the dataset by log-transformation and size factor correction, following three covariates, sequencing library, UMI count, and mitochondrial gene contents (the proportion of mitochondrial genes in transcriptome), and scaled using the *preprocess_cds()* function with 75 PCs. UMAP dimension reduction (*reduce_dimension*) was performed with custom parameters *umap.min_dist*=0.4 and *max_components*=3, and clustering resolution (*cluster_cells*) was set to 0.001 which assigned all cells into a single partition. After graph learning was performed (*learn_graph*), the cells were ordered

using *order_cells()* to set a node embedded in the PH1 subcluster as a start point. All the trajectory graphs were visualized using the *plot_cells()* function with or without a trajectory graph.

Monocle 3 offers several approaches for differential expression analyses using regression or graph-autocorrelation. In this study, we identified co-regulated genes along the pseudotime by graph-autocorrelation and modularized them. To detect co-regulated genes, the graph-autocorrelation function *graph_test()* was specified with a "principal_graph" parameter and significant genes were selected (*q* value < 0.05). Modularization was performed using *find_gene_modules()* with default parameters but only passing a list of resolution values from $10^{-6}$ to 0.1 for automatic parameter selection. A total of 51 gene modules were detected from the normal lymph gland trajectory, however, three modules were excluded in that they were unable to be characterized by the enrichment analysis with biological process gene ontology terms or KEGG pathways using g:Profiler (https://biit.cs.ut.ee/gprofiler/gost).

The trajectory analysis of infested lymph gland (Fig. 7) followed a similar pipeline as that of normal lymph gland with slightly different parameters. The dataset was normalized and corrected for covariates, sequencing library, UMI count, and mitochondrial gene contents in the same manner, though using 50 PCs. We then used 0.5 for the minimum distance in UMAP and 0.005 for the clustering resolution. Gene modules were also explored using a complete dataset and the result generally agreed with the previous gene modules using normal lymph glands. So, we focused on the LM trajectory in the analysis. To collect LM trajectory we cropped cells in 2-dimensional UMAP graph (1,058 cells satisfying *UMAP 3* > 8 * *UMAP 1* + 16). One GST-rich cell was discarded in the analysis. Modularization of co-regulated genes was performed as previously described using these cells.

**Investigation of transcription factor activity using SCENIC**. To predict transcription factor regulatory networks in developing lymph glands, we performed SCENIC[32] analyses on the (1) normal lymph gland dataset excluding non-hematopoietic cells (*n* = 19,332) and (2) early PH subclusters with PSC (PH1, PH2, and PSC; *n* = 78, 81, and 189, respectively). We followed the general SCENIC workflow from gene filtration to binarization of transcription factor (TF) activity described by the authors (https://www.aertslab.org/#scenic), using a provided cis-Target reference based on *Drosophila melanogaster* release 6.02. For the latter analysis, we filtered genes using the following two criteria: (1) genes expressed higher than a UMI count threshold of 3*(3% of total cell count; 31.32), and (2) genes expressed in at least 3% of total cells (10.44 cells). We retrieved 3844 genes from these steps and performed the analysis.

When we investigated the normal lymph gland dataset, however, predictions for several well known TFs in cell types with relatively small sizes, such as *lz* for CC (1.44% of the input dataset) or *Antp* for PSC (0.97% of the input dataset), were affected by the presence of other major cell types. For example, *lz* was an active TF of CC only when PSC, adipohemocyte and GST-rich cell types were excluded from the analysis. We reasoned that the relative population size may affect the predictive power, and signals of TFs active in small populations would be frequently ignored. To tackle this issue, we randomly sampled 42 cells (two thirds of the smallest subcluster—LM2, *n* = 63) from each of 28 hematopoietic subclusters, then collected active TFs using a SCENIC workflow with slightly different gene filtration parameters (we modified 2.5% of the total cell count to 1.0% of total cell count in both gene filtration criteria). We iteratively performed 100 independent trials measuring frequencies for TFs and collected 45 out of 177 TFs predicted active at least 25 times. Then we generated pseudo-bulk profiles for cell types in each timepoint by aggregating scaled gene expression values retrieved from the Seurat object. During the pseudo-bulk generation, LM, CC, GST-rich, and adipohemocyte from 72 h AEL (*n* = 2, 4, 14, and 1, respectively) and LM and adipohemocyte from 96 h AEL (*n* = 14 and 5, respectively) were filtered because the number of cells in these groups was less than 0.1% (~19 cells) of the complete lymph gland dataset. Using pseudo-bulk profiles, we produced a heatmap describing TF expression in cell types for each timepoint (Supplementary Fig. 1k) using the R package *pheatmap*.

**Comparative analysis between normal and wasp infested lymph glands**. For the integration of infested lymph gland (24 h PI; 96 h AEL, *n* = 10,158) with normal dataset from 96 h AEL, those datasets were aligned using Seurat 3.0. functions, *FindIntegrationAnchors()* and *IntegrateData()*, then, cellular annotation was transferred using *FindTransferAnchors()* and *TransferData()* with default parameters. These recent achievements of Seurat enable users to analyze heterogeneous single-cell datasets minimizing unwanted technical effects and transfer cellular annotation from a reference dataset to others. Specifically, these approaches utilize canonical correlation analysis (CCA) to project datasets into same subspace, then define mutual nearest neighbors (MNNs; referred as 'anchors') between datasets to align cells or pass on information, such as cluster annotation or expression values. Again, UMI counts were normalized, log-transformed, and scaled to properly integrate datasets, and the number of PCs was determined (46 PCs). Subclustering labels defined in normal 96 h AEL lymph gland were transferred to infested lymph gland, and four minor subclusters were excluded because they were only found in the normal dataset; PH6 (*n* = 3), PM2 (*n* = 1), PM4 (*n* = 3), and adipohemocyte (*n* = 5). Three non-hematopoietic cell types (35 DV, 18 neurons, and 12 RG cells) were not considered in subsequent analysis.

**Comparative analysis between lymph gland and circulation**. A UMI count matrix of 120 h AEL circulating hemocytes described in Tattikota et al. eLife, 2020[41] was used. Given that the gene annotation version used in Tattikota et al.[41] was different from ours, an FBgn to Annotation ID conversion table was downloaded from FlyBase (https://flybase.org; fbgn_annotation_ID_fb_2019_03.tsv) to match gene IDs. After filtration of non-hematopoietic cell types (DV and RG) from the lymph gland dataset, and circulating hemocytes expressing high mitochondrial genes (higher than 20%), 17,122 cells from 96 and 120 h AEL lymph glands (*n* = 9339 and 7783, respectively) were aligned with 3955 hemocytes from Tattikota *et al.* and additional 96 and 120 h AEL circulation datasets (*n* = 1277, 1006, and 1671, respectively) using *FindIntegrationAnchors()* and *IntegrateData()* in Seurat. Subclustering labels in lymph glands were transferred to the circulation datasets using *FindTransferAnchors()* and *TransferData()* with default parameters. Prior to the comparison of aligned cell types, a number of subclusters in circulation datasets were filtered because they were either (1) subclusters mainly originated from a single library (more than half) or (2) minor subclusters (less than 1% of total circulation cells). After filtration, 3397 cells with five subclusters remained for subsequent analysis (PH1 = 65, PH4 = 55, PM1 = 3084, CC1 = 89, and CC2 = 104). To compare gene expression between circulating and lymph gland hemocytes, each cell type or subcluster was independently analyzed for differentially expressed genes using Seurat. Genes highly upregulated or downregulated in scRNA-seq data compared to bulk-RNA-seq (≥10-fold; 3095 and 1726 genes, respectively) were excluded from the DEG results because these genes may reflect non-biological cellular responses induced during scRNA-seq preparation (upregulated genes enriched with cell response to various stimulus; downregulated genes enriched with metabolic processes). Then, gene expression values from each dataset were averaged and natural log-transformed and top 10 DEGs were marked for visualization (Supplementary Fig. 5).

**Wasp infestation**. Larvae were infested at 72 h AEL with *Leptopilina boulardi* (G486). Wasps were removed after 8 h of co-culture and egg deposition was confirmed by direct observation of wasp eggs in the hemolymph during dissection. All infestation procedures were performed at 25 °C.

**Immunohistochemistry**. Lymph glands were dissected and stained as previously described[16]. The following primary antibodies were used in this study: α-lz (DSHB, 1:10, Mouse), α-Antp (4C3, DSHB, 1:10, Mouse), α-Dl (C594.9B, DSHB, 1:10, Mouse), α-L1 (I.Ando, 1:100, Mouse)[67], α-col (M.Crozatier, 1:400, Mouse)[24], α-Ubx (FP3.38, DSHB, 1:10, Mouse), α-nc82 (DSHB, 1:10, Mouse), α-NimC1 (I. Ando, 1:100, Mouse)[67], α-Pxn(1:2000, Rabbit)[68], α-GFP (Sigma Aldrich; G6539; 1:2,000, Rabbit) and α-F-actin (ThermoFisher; A34055, 1:100). Cy3-conjugated, FITC-conjugated, or Alexa Fluor 647-conjugated secondary antibody (Jackson Laboratory; 115-165-166, 711-165-152, 115-095-062, 711-095-152, 715-605-151) was used for staining at a 1:250 ratio. All samples were kept in VectaShield (Vector Laboratory) and imaged by a Nikon C2 Si-plus confocal microscope.

For α-Delta staining, a pre-absorption step was essential to reduce background in the lymph gland. For the pre-absorption step, 1:10 diluted (with 2% sodium azide) α-Dl antibody was incubated together with 9 fixed larval cuticles overnight at 4 °C. Lymph glands were dissected in Schneider's medium and cultured in Schneider's medium with 10 mM EDTA for 30 min. Samples were incubated with 3.7% formaldehyde for 1 h for fixation at 25 °C. After fixation, lymph glands were washed 3 times in 0.1% Triton X-100 in 1xPBS and blocked in 1% BSA/0.1% Triton X in 1× PBS for 1 h. Lymph glands were incubated overnight at 4 °C with α-Dl antibody. Lymph glands were washed 3 times in 0.1% Triton X in 1× PBS and then incubated with α-mouse secondary antibody with 1% BSA/0.1% Triton X in 1× PBS for 3 h at room temperature. After washing 3 times with 0.1% Triton X in 1× PBS, samples were mounted in Vectashield (Vector Laboratory) with DAPI and imaged by a Nikon C2 Si-plus confocal microscope.

**Bleeding hemocytes**. Larvae were vortexed with glass beads (Sigma G9268) for one minute before bleeding to detach sessile hemocytes[69]. Larvae were bled on a slide glass (Immuno-Cell Int.; 61.100.17) and hemocytes allowed to settle onto the slide at 4 °C for 40 min. Hemocytes were fixed with 3.7% formaldehyde and washed 3 times in 0.4% Triton X-100 in 1× PBS for 10 min and blocked in 1% BSA/0.4% TritonX in 1× PBS for 30 min. Primary antibody (α-Ubx; DSHB, 1:10) was added and samples incubated overnight at 4 °C. Hemocytes were washed 3 times in 0.4% Triton X in 1× PBS and then incubated with a secondary antibody with 1% BSA/ 0.4% Triton X in 1× PBS for 3 h at room temperature. After washing 3 times with 0.4% Triton X in 1× PBS, samples were kept in Vectashield (Vector Laboratory) with DAPI and imaged by a Nikon C2 Si-plus confocal microscope.

**BODIPY or NileRed staining**. Ten to fifteen 120 h AEL larvae lymph glands were dissected in Schneider's medium and incubated with BODIPY (ThermoFisher, D3922) or NileRed (ThermoFisher, N1142) for 30 min in room temperature. Lymph glands were fixed in 3.7% formaldehyde solution for 30 min and washed 3 times in 0.1% PBST (TritonX-100) for 5 min each. Lymph glands were treated with phalloidin (ThermoFisher, A34055, 1:100) after the wash step for an hour and again washed 3 times in 0.1% PBST (TritonX-100) for 5 min each. After the final wash, lymph glands were kept in VectaShield until mounting on a slide glass.

**Fluorescent in situ hybridization**. Lymph glands were dissected in the Schneider's medium less than 30 min and fixed in 4% formaldehyde solution for 30 min. Samples were washed with 0.3% PBST (Tween20) or hybridization solution. Probes were incubated for more than 24 h. An anti-*delta* probe was designed based on *delta* cDNA sequences (Forward primer: ATGTGCGAGGAGAAAGTGCT, Reverse primer: CGACTTGTCCCAGGTGTTTT). DIG-labeled probes were detected by α-DIG-biotin antibody (Jackson Immunoresearch; 200-062-156) and visualization was done using a SuperBoost™ Kit (ThermoFisher; B40933). The sense probe was used as a negative control. Samples were kept in VectaShield until mounting on a slide glass.

**SABER-FISH method**. All the procedures were done as described in the previous study[70] except for a few modifications. Lymph glands were dissected in the Schneider's medium less than 30 min and fixed in 4% formaldehyde solution for 30 min. After fixation, samples were washed with 0.3% PBST (Tween20) three times for 5 min. With the wash hybridization buffer (wHyb), samples were washed at least two times for 5 minutes and treated two different primary probes for 48 h at 43 °C. After probe hybridization, samples were washed with wHyb buffer or 2XSSCT buffer and samples were treated with imager probes. To amplify the signal, the HRP imager probe, instead of Cy3 imager probe, was used. For the HRP boost, SuperBoost™ Kit (ThermoFisher; B40933) was used. Samples were kept in VectaShield until mounting on a slide glass. Probes used in this study were listed in Supplementary Table 7.

**Quantitation, statistics and reproducibility**. All the experiments were analyzed from at least three independent experiments. Lymph gland images shown in Figures or Supplementary Figures indicate the most representative patterns. Stained or fluorescent cells were quantified and analyzed by IMARIS 8.3 software (Bitplane). Individual primary lobes were counted for this study. In vivo data was analyzed by the Mann–Whitney test after determining normality with the use of SPSS (version 24).

**Reporting summary**. Further information on research design is available in the Nature Research Reporting Summary linked to this article.

## Data availability

The authors declare that all data supporting the findings of this study are available within the article and its supplementary information files or from the corresponding author upon reasonable request. The datasets generated in this study are available in following repositories. Raw scRNA-seq and bulk RNA-seq reads have been deposited in the NCBI Gene Expression Omnibus (GEO) database under accession codes: GSE141273 and GSE141274, respectively, linked to a superseries GSE141275. Processed datasets can be mined through a web-tool [http://big.hanyang.ac.kr/flyscrna] that allows users to explore genes and cell types of interest.

## Code availability

In-house R and Python codes that were implemented in this study are available on GitHub [https://github.com/sangho1130/Dmel_Dropseq].

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

## Acknowledgements

The authors thank Dr. Greg S. Suh and all members of the Shim and the BIG labs for helpful discussions. The authors acknowledge the Bloomington, VDRC, DGRC, NIG, and KDRC Drosophila stock centers and the DSHB hybridoma bank. The authors thank the following individuals for stocks and reagents: Drs. C. Evans, U. Banerjee, S. Sinenko, M. Zeidler, M. Crozatier, K. Brueckner, Nambu JR, A. Brand, and F. Schweisguth. This work was supported by the Samsung Science and Technology Foundation under Project Number SSTF-BA1701-15 to J.S. and by the National Research Foundation (NRF) funded by the Ministry of Science and ICT under Project Numbers 2020R1A4A1018398 and 2018R1A2B2003782 to J.N., and 2016R1A5A2008630 to S.J.M. N.P. is an Investigator of the Howard Hughes Medical Institute.

## Author contributions

B.C., S.G.T., D.W.L., F.K., N.C., M.S., H.D., and S.Y.O. performed experiments; B.C., S.Y., S.G.T., Y.H., J.N., and J.S. analyzed data; S.Y.O. and S.J.M. provided technical support for Drop-seq; D.H.L., S.Y., and A.V.M. developed and set-up the scRNA-seq web-tool; B.C., S.Y., F.K., S.G.T., N.P., J.N., and J.S. contributed to writing the manuscript; N.P., J.N., and J.S. supervised the project; J.N., and J.S. conceived the idea.

## Competing interests

The authors declare no competing interests.

## Additional information

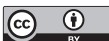

