## [Peer Review File · Nature Communications]

Reviewers' Comments:

Reviewer #1:

Remarks to the Author:

The authors use a single cell approach to characterize hematopoietic lineages in the lymph gland of the *Drosophila* larvae. Being not an expert on single cell transcriptomics, I cannot judge the quality of the data but it should be noted that this work requested the dissection of many lymph glands (a tiny organ) upon both unchallenged and immune conditions, which is time demanding and represents by itself a great achievement.

Overall, I found the paper interesting providing important insights on the hematopoietic process. I have minor recommendation to improve it.

1) Whilst agreeing that the lymph gland is a good model to study hematopoiesis, I would avoid to confuse the reader by using terminology of mammalian hematopoiesis for *Drosophila*.

Plasmacytes are myeloid-like cell, but not myeloid cells...The paper being quite long, I would recommend to delete the last part (Figure 7 and associated text, including some sentences in the introduction and the discussion) which provide an interkingdom comparison of *Drosophila* and vertebrate blood cells. The conclusions are ambiguous. It is not clear to me if there is really analogy or homology between these cells in mammals and insects. This would require to study hemocyte lineages in other phylogenetic groups. At this stage, this part weakens the finding.

2) The finding of a new class of hemocyte, termed adipohemocyte is interesting. It would be interesting to further characterize the lymph gland in relation to lipid uptake. Use of Nile red and Bodipy should allow the authors to probe lipid content of lymph gland cells.

3) The authors identify two routes that lead to lamellocyte differentiation. This is reminiscent of a previous publication (Anderl et al. Transdifferentiation and Proliferation in Two Distinct Hemocyte Lineages in *Drosophila melanogaster* Larvae after Wasp Infection. *PLoS Pathog.* 2016;12(7):e1005746. 2016) that also suggested the existence of lamelloblasts. Can the present study confirm or rather contradict this notion? This article and more globally, the existence of two paths leading to lamellocytes, could be further discussed.

Minor comment

Line 85. The term embryonic hemocyte is confusing. Maybe embryonic derived hemocyte or peripheral hemocytes.

Line 114 and elsewhere: remove myeloid or replace by myeloid like

Line 157 Adjust the sentence to associate the good genes within bracket to the appropriate statement (eater is a phagocytosis receptor not a starvation induced gene)

Line 234: Avoid the use of MZ CZ IZ jargon. Spell out.

Line 247-248: explain more what is this marker and what is its interest

Line 308: Npl2 and other genes. It would be better if the authors could mention the function of the genes they mentioned and reference them. Npl2 was thought to be a neuropeptide but a recent study has shown that this is a lipoprotein. This should be explicated. It was unclear to me which lines was used to assess the expression of this gene.

Line 349. This sentence is unclear to non-lymph gland aficionados

Line 374: would the term precursor be more appropriate than premature

Line 402 is mthl4 a new lamellocyte marker identified in this study or it was previously identified by other?

Line 407-408- these sentences are difficult to understand

Line 458: Ubx expression in peripheral hemocytes: is the function of Ubx in peripheral hemocyte known?

Line 466-504 and 584-597 delete (see above)

Line 518-519: the statement is wrong. There is hematopoiesis in peripheral hemocytes (sessile and circulating)

Line 552: immunogenic mean that trigger an immune response (ex. LPS, peptidoglycan). The term immune -responsive could be more adequate.

Reviewer #2:

Remarks to the Author:

Hematopoiesis in *Drosophila* takes place at two stages and the lymph gland constitutes the site of definitive hematopoiesis. The manuscript from Shim and collaborators presents the first single cell and bulk transcriptome analysis of the *Drosophila* lymph gland. This long-awaited and thorough analysis provides new tools in terms of transgenic lines and markers that will be valuable for the whole community.

The authors first produce bulk and single cell RNA seq data at three developmental stages. This analysis provides a number of novel markers for the PSC, for the prohemocytes, for the crystal cells and for the plasmatocytes. The authors also proceed with bioinformatic analyses to map the identified cells to the major zone of the lymph gland. This allows them to identify seven clusters characterized by distinct molecular signatures. This includes known cells, such as the prohemocytes, the plasmatocytes, the crystal cells and the PSC as well as novel cells such as the adipohemocytes and the GST-rich cells. These clusters are subsequently subdivided into subclusters, for a total of 28. They provide a first validation and a more detailed analysis of the PH1 and PH2 mitotic prohemocytes.

The authors then show that wasp infestation, which triggers an inflammatory response, gives rise to the production of an intermediate and a mature lamellocyte subcluster. Based on their data, they propose that some lamellocytes arise from plasmatocyte transdifferentiation whereas others through cell division, in agreement with published data.

The authors go on and compare the hemocytes originating from the lymph gland and those originating from the first hematopoietic wave that occurs in the embryo. This is a very interesting issue, as the embryonic and the larval derived hemocytes have so been dealt with in the same way. Based on the obtained results, the authors propose that the two waves produce relatively similar cells, although a handful of genes is specifically expressed in one of the two waves.

Finally, the authors compare the RNA seq data with data from the human atlas to identify differences and similarities with the immune cells from control and infested *Drosophila* larvae. This manuscript demonstrates for the first time that the cells of the *Drosophila* lymph gland are highly heterogeneous and provides evidence for lineage progression in this hematopoietic organ. The identification of novel cell types, markers and tools paves the way for further studies in the field.

Specific points to be addressed

Lamellocytes normally appear upon infestation or in mutants affecting the inflammatory response, however, the number of lamellocytes found in 120h AEL control lymph glands (Fig 1) is even higher than that of crystal cells. Could the authors comment on this? Do immunolabeling assays on 120h AEL lymph glands confirm the presence of lamellocytes at that stage?

Could the presence of lamellocytes represent a sign of a pre-inflammatory state in the sample used for the transcriptome analysis? See also the very last comment.

Related to this, did the treatments on the lymph glands affect cell viability and how does the total number of cells recovered for each stage compare with the number of cells present in the lymph gland at that stage?

Fig 1C: the t-SNE plot concerns the 22,645 cells from the three developmental stages. Since the lymph gland is undergoing extensive changes during those stages (see also Fig 1d, extended data Fig 2B), it is more appropriate to identify the clusters/subclusters present at each stage. This may provide a better definition of blood cell heterogeneity. One of the main features of the analysis is that few subclusters are defined by specific and unique markers. While this may reflect a real property of the cells populating the lymph gland, it may also depend on the heterogeneous material used for the study. Identifying the clusters and subclusters at each stage may also help providing a better developmental trajectory. For this reviewer, this is a very important point that could improve significantly the analysis of the different hemocyte populations.

Could the authors provide the number of cells present in each subcluster?

A number of genes associated to the cell cycle are expressed in specific prohemocyte subclusters but are also present, even at higher levels and in more cells, in specific plasmatocyte subclusters (PM3 and 4, Fig 2B). This should be discussed further.

Related to this, the hypothesis that the PH1 subcluster represents stem-like cells seems a bit farfetched without any other evidence on the mode of division (self-renewing, symmetric, asymmetric).

While it is beyond the scope of this manuscript to characterize all the populations of the lymph gland, some *in vivo* validations would have been appreciated, for example, for the cells that are hypothesized to correspond to the intermediate zone/plasmatocytes (page 13).

Some definitions seem a bit strong. For example, NPLP2 is given as a marker of the intermediate cell types (PH6, PH8, PH10, GST-rich and PM1, page 13), however, it is also expressed in the PM5 and PM6 clusters. Are these also intermediate cell types (see also the definition of PM6 as the most mature plasmatocyte subcluster at page 17)? What defines an intermediate cell type? A molecular signature? A geographical distribution?

The analysis of the lymph gland 24 hours post infestation needs a better description. Do the glands at this stage correspond to the lymph glands of control animals at 96h AEL? Is there any developmental delay upon infestation? Are the lymph glands at this stage always present, or some of them are (partially) hystolized?

'The majority of the lamellocytes are directly derived from iPH8...', however this subcluster does not change in terms of absolute number of cells or normalized counts. Could the author explain this?

Page 18: could the authors indicate the genes involved in hemocyte proliferation? Also, since the PH8 cluster of control animals is not enriched in genes associated with cell division, does infestation induce a proliferative potential in this subcluster?

Figure 6 describes a two-dimensional projection of hemocytes in the lymph gland and in circulation at 96 and 120 h AEL. Why merging the two stages? For the circulating hemocytes, the authors refer to an article submitted to BioRxiv. In that article, several techniques were used for the single cell RNA sequencing. Could the authors confine the comparison to the larval single cells data obtained by Drop-seq, which is the approach used in the present manuscript? Also, as the manuscript has to stand by itself, could the authors provide a more detailed comparison between the circulating cells and the lymph gland? Could they list the genes that allow the identification of the different subclusters in the circulating hemocytes? For the PH cluster composed of 67 cells in circulating cells (Fig 6a and c), what is the significance of a gene expressed in less than 50% cells? Also, is there a color coding for the expression intensity?

The comparison with the human atlas is potentially very interesting. As it stands now, however, it is not very informative; the authors need to expand this section. First, it seems difficult to identify cells based on a single marker or the authors should at least refer to published data using this criterion. Second, when they compare the fly and the human data to assign similarities, what is the number of genes involved? Can we see the list of those genes and their allocation to the different human cell types? Third, the authors have demonstrated that the prohemocytes are very heterogeneous cells and yet there is a significant similarity between PH (altogether) and HSC-MPP. To improve the significance of the analysis, could the authors reiterate the comparative analysis with the human cells by using either PH1 cells alone, which constitute the pool of dividing precursors, or the rest of the prohemocyte subclusters (that is, devoid of PH1 cells)? Fourth, could the authors provide a more significant evidence for the *Drosophila* hemocytes acquiring signatures of lymphoid lineages upon immune immunity? This statement seems to me a bit farfetched in the

present state.

The identification of the adipohemocytes is quite exciting. Could the authors perform a labeling for lipid droplets to provide a first validation of these bioinformatics data?

A reference should be added at page 15 line 360.

Minor points:

Could the authors clarify what 0, 2, 4, 6 indicate in extended data Fig 6a, c, d?

Page A9, lines 453-454 are unclear.

The discussion starts with the paragraph that summarizes the data presented in the manuscript. In some instances, the conclusions are a bit overstated.

The analysis at 24h after infestation likely reveals an ongoing response, since the LM1 and LM2 represent a little proportion of the lymph gland cells. This could be mentioned in the text. And finally, related to the very first comment, it seems that the lamellocytes LM1 identified in the control animals (without wasp infestation) share several features with plasmatocytes, suggesting that they are produced by transdifferentiation.

Reviewer #3:

Remarks to the Author:

In their manuscript, Cho et al. provide a single-cell resolution transcriptomic atlas of the *Drosophila* hematopoietic system. They focus their work on the lymph gland, the main hematopoietic organ in the larva, in which they could distinguish specific gene signatures for several subpopulations of progenitor and mature blood cells, revealing a much higher level of cellular heterogeneity than previously thought. They identify new types of hemocytes and they provide new reagents and markers to study lymph gland subpopulations, in particular a prospective stem-like population. In addition, they establish how these hemocyte populations develop during the third instar larvae and how they respond to wasp parasitism. Hence their results bring valuable information about the lineage relationships between the different blood cell populations produced in the lymph gland during normal development and in response to an immune challenge. To give a larger view of the larval hematopoietic system, they also explore the parallels and differences between the gene signature of lymph gland and peripheral larval blood cells. Finally, they provide some novel insights into the similarities between *Drosophila* and mammalian blood cell types by comparing their gene signature.

These findings are particularly important in the field of *Drosophila* hematopoiesis. They bring important and novel information concerning the regulation of blood cell development in the larval lymph gland. It makes no doubt that, even though it is essentially descriptive, this timely work will have a strong impact and will provide a solid basis for deeper analyses of *Drosophila* blood cells diversity and function.

The manuscript is in general very clear and rather convincing. It already contains a load of carefully performed experiments, associated with huge amounts of data to dig in, and a very thorough analysis of the single cell data. Yet, as presented in detail below, some points still really need to be substantiated by a few experiments and several aspects of the manuscript could easily be clarified.

Experimental points.

(1) One potential caveat of this single cell analysis of the lymph gland (an immune-responsive organ highly sensitive to stress) is that the cell dissociation procedure might (is likely to!) affect the gene expression profile of the hemocytes (and in a different manner depending on each subpopulation). Consistent with this hypothesis, in their 120h AEL samples, the authors observe roughly the same proportion of crystal cells and lamellocytes (Fig. 1D), while this cell type is normally not present in healthy larvae. It seems important that the authors take this problem into account and present their results with more caution.

(2) May be there is a trivial explanation but I was very surprised that with 100 to 150 dissected lymph glands (i.e. 50.000 to 400.000 cells, according to Fig 1B), the authors obtained valid sequencing data for only a few hundreds to a few thousand cells in each library (Ext data 1a). That gives a recovery rate of $\pm 1\%$, which seems worryingly low. How can it be explained?

(3) One major claim of the authors is the identification of new cell types (in particular the adipohemocytes, the GST-rich cells and the PH1/stem-like prohemocytes), with specific gene signatures. However, the presence of these populations in wild-type lymph gland is not always shown convincingly and further experiments are clearly needed to substantiate their conclusions. Accordingly, the following points should be addressed:

(3a) Concerning the GST-rich population: the authors only show (Fig. S1H) that the expression of the top 10 genes of the GST-rich cells is also detected in the bulk RNA-seq. That doesn't prove that they make a distinct population in wild-type lymph gland (rather than being artefactually induced in the dissociation process). Actually, the authors also tested a few *Drosophila* lines to analyse the expression pattern of some of these genes (Supl Table 3A), but none of these lines revealed an expression in wild-type lymph glands. In the absence of such an indication it is not possible to conclude that the GST-rich population is genuine. I prompt the authors to try other tools to validate the expression of the GST-rich cells in wild-type lymph gland, may be using RNA in situ hybridisation for the most specific markers or GFP-tagged version of the endogenous gene if the corresponding MiMiC line exist.

(3b) To reveal the adipohemocyte population in wild-type lymph glands, the authors used different GAL4 lines (crq-Gal4, Ama-Gal4 and Lsd2- Gal4; Fig 2). Whether these lines reflect the expression of the endogenous gene is subject to caution. In particular, it remains to be shown that the higher level of RedStinger observed in the prospective NimC1-low/adipohemocyte population (for crq-GAL4 and Lsd-2-GAL4; cf. line 246) reflects crq or Lsd-2 expression levels. Again, fluorescent in situ hybridisation or immunostaining (since an antibody is available for Crq) coupled with immunostaining against NimC1 should clarify the matter.

(3c) Along the same line, the authors claim (line 242; Fig. S2h) that zfh1 is a prohemocyte marker based on the use of a zfh1-GAL4 line (and on their scRNA-seq data). Given that antibodies against Zfh1 as well as a GFP-tagged version of the gene (cf PMID: 30002131) are available, I strongly advise the authors to use one of these tools to verify their conclusion (in our own experience, Zfh1 immunostaining revealed a rather ubiquitous pattern in the lymph gland, which was also observed with a zfh1-GAL4 line different from the one used here). More generally, the authors should be more cautious (and more precise) in their conclusions concerning the expression of different genes when it is based solely on GAL4 lines.

(3d) For the PH1 population, the authors show some more convincing evidence. However, they should clearly mention that the STAT92E::edGFP is a reporter of the JAK/STAT pathway and not a reporter of STAT92E expression. It is thus misleading to call these cells STAT92E+. Actually, the authors could also look at the endogenous STAT92E protein. Also, given that the PSC was proposed to express Upd3, it would be interesting to test the effect of a PSC-specific knock-down of upd3 on the activity of STAT92E-edGFP.

(4) The identification of the PH1/stem cell population is a major result. However, the authors should put in perspective their results with those of Dey et al. (ref 76), who described a putative Notch-expressing HSC population in the early larval lymph gland.

(5) The authors suggest that the Notch pathway is active in the PH1 cells but they do not bring strong evidence for this. As for the JAK/STAT pathway, they should investigate Notch pathway

activity using the equivalent Su(H)GBE-edGFP reporter (cf. He et al., PMID 31140975). They should also show whether Delta expression (or Notch activity) depends on the PSC, as shown for STAT92E-edGFP.

(6) Using the Gtrace lineage labelling technique, the authors show (Fig. 4G) that DI-GAL4+ cells close to the PSC “produce hemocytes of the entire lymph gland” (line 367) and they conclude that PH1 cells are reminiscent of mammalian HSC. Even though DI>GTRACE seems to label a large fraction of the lymph gland, it does not demonstrate unambiguously that PH1 cells (or DI>Gtraced cells) give rise to all the different types of hemocytes (i.e. that they have multilineage differentiation capacity). The authors must show, using immunolabelling with specific markers, that the major differentiated cell types (plasmatocytes, adipohemocytes, crystal cells and lamellocytes) are indeed labelled with DI>Gtrace.

(7) In their wasp infestation experiments, the authors found no change in the gene expression signature of the PSC cells. This is unexpected given the central role of these cells as a relay required to induce lamellocyte differentiation in the lymph gland, and several papers observed the (transcriptional) induction of different signalling pathways in these cells following wasp infestation (see for instance Louradour et al. PMID: 29091025; or Sinenko et al. PMID: 22134547). This possible inconsistency should be discussed.

(8) In Fig. 5C, Delta staining does not show any background outside the “medullary zone” in uninfested larvae, while there is quite some background in infected larvae or in Fig.4F In addition, DI staining does not seem to be mainly at the cell membrane (especially in Fig 5C) and the way the immunostaining is performed (i.e. with a 30' incubation of dissected lymph glands in 10mM EDTA) is strange (anti-DI from DSHB has been used in many studies but I did not see that they used a similar protocol). Since a 30' incubation with 2mM EDTA causes Notch pathway activation in cell culture (PMID: 17545467 and others), the authors should repeat their immunostaining without such treatment and/or use one of the published available DI-GFP knock-in lines (e.g. PMID: 26102525 or PMID: 31668010) to confirm their results concerning DI expression.

(9) Concerning the differences between circulating and lymph gland hemocytes. (a) Ubx staining (Fig. S6g,h) is clearly cytoplasmic. Do the author have any explanation for this? (b) A transcriptome of the circulating crystal cells has been published (Miller et al., PMID 27487438). How does it compare with crystal cells clusters identified in this work? Notably Miller et al observed Oscillin expression in circulating crystal cells whereas the authors suggest that this gene is specific to the lymph gland crystal cells.

Other points.

(1) Line 44-45: it seems odd to start the introduction with a reference to “memories of immunological events” since this is not a prevalent feature for insect blood cells.

(2) Line 62: I don't know why the authors state that the larval hematopoiesis is the “definitive” one. It has been shown that both embryo- and lymph gland-derived hemocytes contribute to the adult hematopoietic system, so both waves could be considered as definitive!

(3) Line 64-66: I'm not sure that freely circulating and sessile hemocytes should be described as distinct populations: sessile hemocytes can enter circulation and vice versa.

(4) Line 68-69: the reference is missing (Mandal et al. 2004). And only part of the lymph gland was shown to arise from hemangioblast-like cells.

(5) Line 68-70: the introduction of the overall lymph gland morphology/organisation is not very clear.

(6) Line 83: not every wasp species infest *Drosophila melanogaster*.

(7) Lines 87-88: lymph gland hemocytes do not “remain intact” at 24h post-infestation: they are already affected/start to differentiate in lamellocytes (see for example Louradour et al. 2017).

(8) Lines 97-99: some adequate references could be included.

(9) Lines 321-324: the corresponding references should be included (notably: Minakhina & Steward, 2010; Krzemien et al., 2010; Dey et al. 2016).

- (10) Lines 448, 449, 451 and 459: these refer to extended Data fig 6 (not 5).
- (11) Line 511: why do the authors think that they have revealed "genetic" differences in hemocytes derived from the two waves of hematopoiesis?
- (12) Line 513-515: actually, there was already a scRNA-seq paper for invertebrate blood cells (in mosquitoes: PNAS 2018; PMID 30038005). It should be cited.
- (13) Lines 526-528: as mentioned above, this is not very convincing.
- (14) Lines 568-570: this is vague. What did we learn from these publications concerning the role of these pathways in the LG (and potentially in HSC-like cell/progenitor maintenance)?
- (15) Lines 601-611: when possible, I would suggest that the authors cite the relevant publications for each stock (for instance He et al., for STAT92E-edGFP, or Evans et al. for UAS-Gtrace)
- (16) Lines 992-994: the number of biological replicates used for the bulk RNAseq should be indicated.
- (17) Line 354: as mentioned above, there is some confusion between the "expression of genes" and that of reporter lines.
- (18) Line 928: which publication is Sudhir et al.?
- (19) Line 1003: the strain of *Leptopilina boulardi* that was used should be mentioned.
- (20) Lines 1035-1038: it seems that the fixation step is missing.
- (21) Figure 4 panel b -right panel: I observed an alteration of the red channel in a part of the panel. This is probably due to a mishandling during figure preparation and should be corrected.
- (22) Extended data table S2: the legend is not clear. What are the values in the columns C to G and how were they calculated?

Lucas Waltzer

Reviewer #4:

Remarks to the Author:

Summary:

In this paper, Cho, Yoon et al profile the transcriptomes of the *Drosophila* lymph gland different cell types at different stages of development (72, 96, and 120 hours after egg laying) using Drop-Seq to achieve single cell resolution. They then use their single-cell atlas to address several interesting questions regarding the development and evolution of these cell types.

The authors first use known markers to identify the different cell types of the *Drosophila* lymph nodes. They identify all cell types, as well as 2 more that they name adipohemocytes and GST-rich. They then subcluster these cell types into smaller clusters. They find that these subclusters represent either populations at different cell cycle stages or populations at different levels of maturity. They use this information as well as a trajectory inference algorithm (Monocle) to define the developmental trajectories of the different cell types.

The authors then look at the differences that arise in the lymph node cell type composition upon wasp infestation, as well as the transcriptional differences between the same cell types before and after infestation. They show that lamellocytes are generated by prohemocytes and plasmatocytes. The authors then compare the hemocytes that are in circulation (that are generated during the embryonic hematopoiesis) with the ones that reside in the lymph gland (generated by the larval definitive hematopoiesis). They find that the two populations are very similar, with few differences that are not very convincing. Finally, they compare the *Drosophila* hemocytes with the human immune cells based on ~6,500 orthologous genes. They find that the *Drosophila* hemocytes correspond pretty clear to the human myeloid immune cells. Interestingly, they observe that *Drosophila* hemocytes upon infestation acquire characteristics of human lymphoid cells.

This paper is a great resource as single-cell transcriptomics paper and it addresses several interesting questions. However, although it touches upon different subjects, it falls short of proving any of the claims that are made. Moreover, many of the observations that are very interesting (i.e. new cell types, different subtypes, slight differences between the different *Drosophila* hemocyte

lineages, similarities and differences to human cells) are very superficially presented and are not convincing. Although the dataset has a lot of potential, it is often presented as a list of genes at different types/subtypes/stages that are not interesting to read, except by hemocyte aficionados. I would recommend publication as a resource paper, after a) toning down a few of the claims that are made (please see Major points), b) highlighting that it should be treated like a resource, and c) re-writing it to make it more appropriate for a broad audience.

Major points:

- 1) The GST-rich and adipohemocyte populations are not convincing. The authors need to show these populations in the tissue with appropriate counterstains. Otherwise, they could mention that the data suggest their presence, but it remains to be shown where they are. Also, it is unclear what the comparison to the bulk is trying to achieve. Of course, if there are single-cells expressing these genes, the bulk tissue will also express them. This does not prove that these are a specific cell type.
- 2) Related to the previous point, most of the antibody stainings that are presented are poor and lack appropriate counterstains.
- 3) The identification of subclusters is meaningless unless the authors provide in situ stains that verify these subclusters. Moreover, given the fact that these subclusters separate cells at different stages of the cell cycle or their development, I would be very cautious in calling them subclusters – they might as well be a continuum that the authors are arbitrarily separating them in clusters. I also had a hard time understanding from the Methods how they decided the number of the subclusters, given the fact that by increasing or decreasing the resolution, one can have as many subclusters as they want. They claim that they manually inspected the tSNE plots, if I understood correctly. I would remove this part.
- 4) The lists of genes that are presented at different stages of the paper are very tiring and seem useless. They can all be summarized in Tables and removed from the main text.
- 5) The differences that the authors identified between the two different hemocyte lineages (a couple rRNAs and a couple CGs) are very unconvincing. Then, the authors try to compare specific cell types between the two lineages and come up with a couple more differences. Unless they provide stainings of the different populations to show these differences, I do not think that these differences are convincing.

Minor points:

- 1) Why do the authors use these developmental stages (72, 96, and 120 hours AEL)? Is there anything significant in these stages in terms of hemocyte development?
- 2) The Introduction is very hard to read for non-experts - it would benefit greatly from a schematic.
- 3) The authors focus their analysis on cells with many reads/genes per cell. While this is good, they might be selecting for highly transcribing cells. If they relax their criteria to choose specific cells, do they still see the same populations?
- 4) Figure 1c has a couple typos (e.g. missing comma in 8218 and crystal cells)
- 5) It is unclear what the SCENIC output offers to this paper. I would remove or explain better.

Point-by-point response

We would like to thank all the reviewers for their invaluable comments and suggestions. We have now addressed all the comments, which substantially improved the quality of our manuscript and further supported the conclusions of our key findings. Please see below our point-by-point rebuttal to the comments raised by the reviewers.

* Reviewers' concerns = black

* Response = blue

Reviewers' comments:

Reviewer #1 (Remarks to the Author):

The authors use a single cell approach to characterize hematopoietic lineages in the lymph gland of the *Drosophila* larvae. Being not an expert on single cell transcriptomics, I cannot judge the quality of the data but it should be noted that this work requested the dissection of many lymph glands (a tiny organ) upon both unchallenged and immune conditions, which is time demanding and represents by itself a great achievement.

Overall, I found the paper interesting providing important insights on the hematopoietic process. I have minor recommendation to improve it.

We thank the reviewer for the kind words and constructive comments.

1) Whilst agreeing that the lymph gland is a good model to study hematopoiesis, I would avoid to confuse the reader by using terminology of mammalian hematopoiesis for *Drosophila*. Plasmacytes are myeloid-like cell, but not myeloid cells...

As per the reviewer's suggestion, we deleted or changed the term 'myeloid' to 'myeloid-like'.

The paper being quite long, I would recommend to delete the last part (Figure 7 and associated text, including some sentences in the introduction and the discussion) which provide an interkingdom comparison of *Drosophila* and vertebrate blood cells. The conclusions are ambiguous. It is not clear to me if there is really analogy or homology between these cells in mammals and insects. This would require to study hemocyte lineages in other phylogenetic groups. At this stage, this part weakens the finding.

This is a fair critique and we agree that the inter-phylum comparison shown in Figure 7 would benefit from additional analyses with comparisons to other phylogenetic groups. Thus, we have deleted Figure 7 and the associated text in the revised manuscript. We hope to expand the analyses in the future.

2) The finding of a new class of hemocyte, termed adipohemocyte is interesting. It would be interesting to further characterize the lymph gland in relation to lipid uptake. Use of Nile red and Bodipy should allow the authors to probe lipid content of lymph gland cells.

This is a good point raised by the reviewer, and we should have included the results in our previous version. To sufficiently confirm the presence of adipohemocytes in the lymph gland, we validated two representative transcripts enriched in adipohemocytes – Sirup and Lsd-2. Both transcripts are detected within the cortical zone; however, Sirup or Lsd-2 expressing cells are devoid of the cortical zone marker, Hml (Extended Data Fig. 3b). This pattern is reminiscent of the computational analysis shown in Fig. 2b. In addition to gene expression, we now show that a few hemocytes in the lymph gland indeed enclose Nile red- or BODIPY-positive lipid droplets (Fig. 2d; Extended Data Fig. 3c), identifying novel lipid-containing hemocytes in the lymph gland.

Together, these data suggest the expression of adipohemocytes in the lymph gland and support the computational analysis (Reviewer's Figure 1).

Figure 1 Validation of adipohemocytes

(top: BODIPY or Nile Red staining, bottom: mRNA expression of specific genes)

3) The authors identify two routes that lead to lamellocyte differentiation. This is reminiscent of a previous publication (Anderl et al. Transdifferentiation and Proliferation in Two Distinct Hemocyte Lineages in *Drosophila melanogaster* Larvae after Wasp Infection. PLoS Pathog. 2016;12(7):e1005746. 2016) that also suggested the existence of lamelloblasts. Can the present study confirm or rather contradict this notion? This article and more globally, the existence of two paths leading to lamellocytes, could be further discussed.

The elegant study performed by Hultmark and colleagues has shown that there are two parallel lamellocyte lineages in the larval circulation, with one type of lamellocyte generated by transdifferentiation of plasmatocytes and the other arising from a designated pool of infection-induced lamelloblasts¹.

Here, we show two comparable routes of lamellocyte differentiation in the lymph gland: 1) differentiation from prohemocytes and 2) transdifferentiation of plasmatocytes. Notably, we observed a significant expansion of iPH4 (Extended Data Fig. 6b-c) and its biased differentiation to the lamellocyte lineage in the trajectory analysis (Fig. 5e). Moreover, cells in NimC1⁺ iPM1, the most differentiated population in the 96 h AEL lymph gland, transdifferentiate into lamellocytes (Fig. 5e). Thus, our findings correspond well with the previous study¹ and affirms the existence of two universal routes for the lamellocyte differentiation, both of which can occur in the larval circulation and the lymph gland. Despite the similarities, however, we note that lymph glands remain in place at 24 hours post infestation (Extended Data Fig. 6a) and is independent of lamelloblasts and lamellocytes detected in the circulation at that time.

Minor comment

Line 85. The term embryonic hemocyte is confusing. Maybe embryonic derived hemocyte or peripheral hemocytes.

We changed modified 'embryonic hemocyte' to 'embryonically-derived hemocyte'.

Line 114 and elsewhere: remove myeloid or replace by myeloid like

We deleted or changed "myeloid" to "myeloid-like".

Line 157 Adjust the sentence to associate the good genes within bracket to the appropriate statement (eater is a phagocytosis receptor not a starvation induced gene)

We changed the sentence accordingly.

Line 234: Avoid the use of MZ CZ IZ jargon. Spell out.

We spelled out the full terms.

Line 247-248: explain more what is this marker and what is its interest

We have clarified the markers and their potential usefulness.

Line 308: Npl2 and other genes. It would be better if the authors could mention the function of the genes they mentioned and reference them. Npl2 was thought to be a neuropeptide but a recent study has shown that this is a lipoprotein. This should be explicated. It was unclear to me which lines was used to assess the expression of this gene.

We have added references for Nplp2² and Nplp2-gal4³ in the revised manuscript. A putative function for Nplp2 as a lipoprotein is now included. Nplp2-gal4 was first generated by the Kim lab³ and is now maintained at the Korean Drosophila Resource Center (KDRC). The stock can be acquired upon request.

Line 349. This sentence is unclear to non-lymph gland afiliados

We added examples of PSC-specific transcription factors to help make the sentence clear to a broader readership.

Line 374: would the term precursor be more appropriate than premature

We changed 'premature' to 'precursor' in line 374.

Line 402 is mthl4 a new lamellocyte marker identified in this study or it was previously identified by other?

mthl4 is a new gene identified in this study. Additional studies will be required to unravel novel functions of mthl4 in lamellocytes.

Line 407-408- these sentences are difficult to understand

We modified the sentences.

Line 458: Ubx expression in peripheral hemocytes: is the function of Ubx in peripheral hemocyte known?

Expression of Ubx in peripheral hemocytes is first shown in this study and future studies will explore novel functions of Ubx in hemocyte development and immunity.

Line 466-504 and 584-597 delete (see above)

We deleted Figure 7 and the associated text as suggested.

Line 518-519: the statement is wrong. There is hematopoiesis in peripheral hemocytes (sessile and circulating)

We edited the statement.

Line 552: immunogenic mean that trigger an immune response (ex. LPS, peptidoglycan). The term immune -responsive could be more adequate.

We deleted the term.

--

Reviewer #2 (Remarks to the Author):

Hematopoiesis in *Drosophila* takes place at two stages and the lymph gland constitutes the site of definitive hematopoiesis. The manuscript from Shim and collaborators presents the first single cell and bulk transcriptome analysis of the *Drosophila* lymph gland. This long-awaited and thorough analysis provides new tools in terms of transgenic lines and markers that will be valuable for the whole community.

The authors first produce bulk and single cell RNA seq data at three developmental stages. This analysis provides a number of novel markers for the PSC, for the prohemocytes, for the crystal cells and for the plasmatocytes. The authors also proceed with bioinformatic analyses to map the identified cells to the major zone of the lymph gland. This allows them to identify seven clusters characterized by distinct molecular signatures. This includes known cells, such as the prohemocytes, the plasmatocytes, the crystal cells and the PSC as well as novel cells such as the adipohemocytes and the GST-rich cells. These clusters are subsequently subdivided into subclusters, for a total of 28. They provide a first validation and a more detailed analysis of the PH1 and PH2 mitotic prohemocytes.

The authors then show that wasp infestation, which triggers an inflammatory response, gives rise to the production of an intermediate and a mature lamellocyte subcluster. Based on their data, they propose that some lamellocytes arise from plasmatocyte transdifferentiation whereas others through cell division, in agreement with published data.

The authors go on and compare the hemocytes originating from the lymph gland and those originating from the first hematopoietic wave that occurs in the embryo. This is a very interesting issue, as the embryonic and the larval derived hemocytes have so been dealt with in the same way. Based on the obtained results, the authors propose that the two waves produce relatively similar cells, although a handful of genes is specifically expressed in one of the two waves.

Finally, the authors compare the RNA seq data with data from the human atlas to identify differences and similarities with the immune cells from control and infested *Drosophila* larvae.

This manuscript demonstrates for the first time that the cells of the *Drosophila* lymph gland are highly heterogeneous and provides evidence for lineage progression in this hematopoietic organ. The identification of novel cell types, markers and tools paves the way for further studies in the field.

We thank the reviewer for the kind words and constructive review. Please see below the responses and changes we made to clarify specific points.

Specific points to be addressed

1) Lamellocytes normally appear upon infestation or in mutants affecting the inflammatory response, however, the number of lamellocytes found in 120h AEL control lymph glands (Fig 1) is even higher than that of crystal cells. Could the authors comment on this? Do immunolabeling assays on 120h AEL lymph glands confirm the presence of lamellocytes at that stage? Could the presence of lamellocytes represent a sign of a pre-inflammatory state in the sample used for the transcriptome analysis? See also the very last comment.

In line with your concerns, we were surprised by the high number of lamellocytes in the lymph gland. To determine the number of lamellocytes normally present in wild-type lymph glands, we counted the number of naturally occurring lamellocytes using atilla/L1, L2, or msn-mCherry as markers and discovered that ~7% of lymph glands have lamellocytes in normal growing conditions (Reviewer's Figure 2). Thus, it is apparent that lamellocytes occasionally develop in the wild-type lymph gland. However, the number of lamellocytes detected in our data is higher than what is expected.

Figure 2 Development of lamellocytes in wild-type lymph glands

$$\frac{\text{Lymph gland expressing lamellocyte}}{\text{Total lymph gland lobes observed}} = \frac{18}{264} = 6.81\%$$

There are three possible explanations for the higher number of lamellocytes. First, it is possible that the inflammatory responses due to the cell dissociation or the Drop-seq single-cell sorting causes lamellocyte differentiation. Second, lamellocytes might be enriched by the Drop-seq flow. In our experience, the capture rate largely depends on cell size and bigger cells show a better selectivity in the Drop-seq flow. This may cause an increase in lamellocyte number in the scRNA-seq data. The third possibility is that the lamellocyte markers we use are insufficient to visualize premature lamellocytes developing in the lymph gland. We would like to note that the majority of LMs in the wild-type lymph gland are premature LM1 that lack mature lamellocyte markers such as atilla. Although we have tested several representative markers, including atilla/L1, L2, and msn, these antibodies/reporter may not be sufficient to distinguish the earliest type of lamellocytes or reveal differentiating lamellocytes.

Interestingly, some genes are more enriched in scRNA-seq than in bulk RNA-seq data, which are summarized as immune response or metabolic process-regulating genes. We further confirmed that these genes can be induced or reduced by dissociation of lymph glands. Thus, we excluded genes that were highly up- or down-regulated compared to bulk RNA-seq (10-fold) from downstream analyses (Reviewer's Figure 3). We describe our subclustering procedures in detail in the Materials and Methods. We now discuss this issue in the revised manuscript.

Figure 3 Exclusion of genes biased to scRNA-seq or bulk sample RNA-seq

2) Related to this, did the treatments on the lymph glands affect cell viability and how does the total number of cells recovered for each stage compare with the number of cells present in the lymph gland at that stage?

We scored viable cells before running them through Drop-seq to check for possible damage of hemocytes after lymph gland dissociation. Moreover, we included an additional quality control measure during the initial analyses by removing cells with high mitochondrial

transcript content⁴. Specifically, cells with mitochondrial gene contents elevated higher than 10% for the lymph gland or 20% for circulating hemocytes (red lines) of total expression were excluded in our analysis (Reviewer's Figure 4).

Figure 4 Exclusion of cells with high mitochondrial contents

Comparing the total number of cells initially dissected with the number of cells recovered by computational analyses, we observed that the recovery rate is 1~2% in total. The numbers of cells yielded after sequencing are 5.5X, 6.8X, and 2.4X of one lymph gland lobe at 72, 96, and 120 hour AEL (number of lymph gland hemocytes, Fig. 1b; total number of cells recovered by analysis, Extended Data Fig. 1a) (Reviewer's Figure 5).

Figure 5 The number of hemocytes captured (left) and the actual number of hemocytes per one lobe (right)

3) Fig 1C: the t-SNE plot concerns the 22,645 cells from the three developmental stages. Since the lymph gland is undergoing extensive changes during those stages (see also Fig 1d, extended data Fig 2B), it is more appropriate to identify the clusters/subclusters present at each stage. This may provide a better definition of blood cell heterogeneity. One of the main features of the analysis is that few subclusters are defined by specific and unique markers. While this may reflect a real property of the cells populating the lymph gland, it may also depend on the heterogeneous material used for the study. Identifying the clusters and subclusters at each stage may also help providing a better developmental trajectory. For this reviewer, this is a very important point that could improve significantly the analysis of the different hemocyte populations.

Based on the reviewer's suggestion, we performed clustering of cells at each stage and compared the resulting clusters with our original clusters from all stages. We did not find any significant differences at the cluster- or marker-level (Reviewer's Figure 6-7; Extended Data Fig. 2c-d).

Stage-specific subclustering of PHs or PMs shows that all subclusters at each stage are already defined by clustering of the combined stages (Extended Data Fig. 2c-d). For example, subclusters of PH cells at 72, 96, and 120 h AEL match to early PH subclusters, PH1 and PH2/PH3, or 120 h AEL-specific subcluster, PH6. Stage-specific subclusters of PM cells are either mature/late stage cells (NimC1 expressing or 120 h AEL specific cells) or PH-PM cells (Ance expressing, intermediate cells). In addition, subclustering of all stages finds more subclusters at 96 and 120 h, supported by unique gene markers (Extended Data Fig. 2c-d).

Further, we confirmed that cells of each subcluster at different time points correspond well, except for 120 h AEL-specific subclusters that do not have matched subcluster at earlier time points (see Extended Data Fig. 2b: PH1-PH4, PM1, PSC, CC, GST-rich, and LM).

Regarding the issue of original subclusters lacking unique markers, we have developed a new iterative subclustering method that combines continuums with similar signatures, allowing us to define optimal subclusters with unique markers (see Materials and Methods for details). We now have fewer subclusters (11 to 6 for PH; 10 to 4 for PM) than we had in

our original PH and PM subclusters, and these subclusters have distinct marker sets (see our response below).

4) Could the authors provide the number of cells present in each subcluster?

We provide the number of cells in each subcluster in Extended Data Table 2.

5) A number of genes associated to the cell cycle are expressed in specific prohemocyte subclusters but are also present, even at higher levels and in more cells, in specific plasmatocyte subclusters (PM3 and 4, Fig 2B). This should be discussed further.

Yes, the reviewer is correct. To adequately address reviewer 4's point 3 and highlight subclusters with active cell cycle genes, we redefined PH and PM subclusters and reanalyzed the expression of cell cycle regulators in the current manuscript. As a result, we found that PH1-PH4 are mitotic, consistent with the previous analysis, and that PM1, which includes PM3 and PM4 in the previous manuscript, exhibits high cell cycle genes amongst PMs. Considering that PM1 emerges prior to the final differentiation and denotes high cell cycle genes, we hypothesize that the proliferation of PM1 could be associated with the differentiation of plasmatocytes or crystal cells. We have added a description of this in the results section of the revised manuscript.

6) Related to this, the hypothesis that the PH1 subcluster represents stem-like cells seems a bit farfetched without any other evidence on the mode of division (self-renewing, symmetric, asymmetric).

We now show that a subset of PH1 cells incorporates EdU, demonstrating that these cells undergo active cell division, as analyzed. Intriguingly, the majority of EdU-expressing cells are next to *STAT92E^{Act}* PH1, suggesting that early prohemocytes actively proliferate in the lymph gland.

Unfortunately, we have not been able to observe a specific mode of division due to a lack of reagents. As an alternative, we counted the number of PH1 cells at each stage and found that the number of PH1 cells increases, such that the lymph gland keeps a constant ratio of PH1 throughout lymph gland development (Fig. 4c). These data imply that PH1 may at least symmetrically divide to maintain the constant proportion, which can be accompanied by differentiation with an alternative division mode. Furthermore, the number of PH1 cells significantly reduces upon wasp infestation (Fig 5c), suggesting that these serve as a pool of cells that differentiate upon immune challenge. It will be essential to further characterize the nature of the PH1 cluster and its underlying control mechanisms in the future.

7) While it is beyond the scope of this manuscript to characterize all the populations of the lymph gland, some in vivo validations would have been appreciated, for example, for the cells that are hypothesized to correspond to the intermediate zone/plasmatocytes (page 13).

We thank the reviewer for constructive suggestions.

1) Intermediate zone: We now further characterize the expression in the intermediate zone by probing spatial associations of Nplp2+ cells with additional marker genes. In the revised manuscript, we profiled the expression of Nplp2-gal4+ cells at 72, 96, and 120h AEL and established how the prospective intermediate zone emerges over development (Fig. 3d). At 72 h, Nplp2-gal4 expressing cells occupy ~40% of the lymph gland and partially overlay with Hml-RFP, dome-eBFP, or both. As the lymph gland grows, the proportion of Nplp2-gal4+ cells decreases while remaining in between the Hml+ cortical zone and the Dome+ medullary zone. Moreover, we verified that Nplp2-gal4 positive cells localize right next to

collier-positive hemocytes (PH2 and PH3) but these are separable from one another (Extended Data Fig. 4g). Also, the inner edge of the *Nplp2-gal4+* cell boundary co-localizes with the outer demarcation of *Ance^{MiMIC}* (Extended Data Fig. 4h). We also confirmed that *Nplp2-gal4+* cells and *NimC1+* plasmatocytes are exclusive (Extended Data Fig. 4i). Finally, we profiled cell cycle phases of *Nplp2-gal4*-expressing cells and found that *Nplp2-gal4+* cells exhibit significantly different cell cycle patterns when compared to *Tep4-gal4* using *UAS-FUCCI* (Extended Data Figure. 4j). Together, these results suggest that the intermediate zone arises following PH3 and persists until the final differentiation (Reviewer's Figure 8).

Figure 8 Expression of *Nplp2* in the intermediate zone

In addition to the intermediate zone, we further verify the expression of adipohemocytes (Extended Data Fig. 3b) and GST-rich cells (Extended Data Fig. 3a).

2) Adipohemocytes: To sufficiently confirm the expression of adipohemocytes, we validated two representative transcripts enriched in adipohemocytes – *Sirup* and *Lsd-2* – in the lymph gland. Both transcripts are detected within the cortical zone; however, *Sirup* or *Lsd-2* expressing cells are devoid of the cortical zone marker, *Hml* (Extended Data Fig. 2b). This pattern is reminiscent of the computational analysis shown in Fig. 2b. In addition to gene expression analysis, we now also show that a few hemocytes in the lymph gland indeed enclose Nile red- or BODIPY-positive lipid droplets (Fig. 2d; Extended Data Fig. 3c), identifying a novel lipid-containing hemocyte in the lymph gland (Reviewer's Figure 1).

3) GST-rich: We validated the expression of two GST-rich-specific transcripts including *CG18547* and *CG3397* by SABER-FISH. Both *CG18547* and *CG3397* sporadically localize adjacent to *TepIV*-positive early prohemocytes (Extended Data Fig. 3a). Neither *CG18547*-positive cells nor *CG3397*-positive cells co-localizes with *TepIV-gal4*-expressing cells, yet they are closely adjoined. These results indicate that the GST-rich population is present in the lymph gland and likely emerges following the *TepIV*⁺ PH differentiation (Reviewer's Figure 9).

Figure 9 Validation of GST-rich subcluster

4) Prohemocyte subclusters: With the use of known or new marker genes, we defined the spatial organization of PH subclusters in the lymph gland. In addition to the STAT92E^{active}/Delta-positive PH1, we established that col^{low}/Tep4-gal4⁺ cells best represent PH2 and PH3 cells (Extended Data Fig. 2g) and Tep4-gal4⁺ cells lacking col^{low} expression are well-suited to PH4/PH5 (Extended Data Fig. 2g-h). Furthermore, we confirmed that there are a few Ilp6⁺/dome⁺ cells in the medullary zone, indicating the presence of the novel and rare cell type in the late lymph gland (Extended Data Fig. 2i) (Reviewer's Figure 10).

Figure 10 Validation of PH sub-population

Together, these data suggest that marker genes for each subcluster are expressed in the lymph gland and indicate the dynamic nature of lymph gland differentiation.

8) Some definitions seem a bit strong. For example, NPLP2 is given as a marker of the intermediate cell types (PH6, PH8, PH10, GST-rich and PM1, page 13), however, it is also expressed in the PM5 and PM6 clusters. Are these also intermediate cell types (see also the definition of PM6 as the most mature plasmacyte subcluster at page 17)? What defines an intermediate cell type? A molecular signature? A geographical distribution?

Previous studies have shown that some of the Hemolectin-positive differentiating hemocytes co-express a representative prohemocyte marker, domeless-gal4, and defined this population as the intermediate zone⁵. As indicated in the response above (point 7), The expression of Nplp2-gal4 matches well with the previous definition of the intermediate zone.

We agree with the reviewer that Nplp2 transcripts exhibit a gradual increase and decrease over hemocyte development and are not strictly restricted to the intermediate subclusters we defined (PH4, PM1, and GST-rich). In the revised manuscript, we clarify that Nplp2-gal4 is a

putative marker for the intermediate zone that reflects the highest level of *Nplp2* mRNA expression.

9) The analysis of the lymph gland 24 hours post infestation needs a better description. Do the glands at this stage correspond to the lymph glands of control animals at 96h AEL? Is there any developmental delay upon infestation? Are the lymph glands at this stage always present, or some of them are (partially) hystolized?

We infected fly larvae with *Leptopilina boulardii* wasps at 72 h AEL and let them grow for another 24 h, which corresponds to 96 h AEL of developing unchallenged larvae. We examined that the number of cells in one lymph gland lobe after 24 h PI is comparable to that of controls at 96 h AEL (Extended Data Fig. 6a), suggesting that lymph gland development is not significantly altered by the infestation. This data also confirms that the lymph gland is yet to disintegrate at this stage. These results are now included in the Results and Discussion sections (Reviewer's Figure 11).

Figure 11 The number of cell in the lymph gland after 24 h or 48 h post infestation

10) 'The majority of the lamellocytes are directly derived from iPH8...', however this subcluster does not change in terms of absolute number of cells or normalized counts. Could the author explain this?

With the new subclustering analysis, we verified that the number or proportion of iPH4 (iPH8 in the original version) increased upon wasp infestation (Extended Data Fig. 6b-c). Also, we noticed that iPH4 shows high hemocyte proliferation-, DNA replication-, and ribosome biogenesis genes upon wasp infestation (Fig. 5e; Extended Data Table 5b). We assume that the loss of early PHs, including iPH1 and iPH2, may trigger the proliferation of succeeding iPH4. The increase in the number/proportion of iPH4 is reminiscent of what was described in a previous study for lamelloblasts¹ (Reviewer's Figure 12).

Figure 12 Proportional change of iPH4 population and its gene module

11) Page 18: could the authors indicate the genes involved in hemocyte proliferation? Also, since the PH8 cluster of control animals is not enriched in genes associated with cell division, does infestation induce a proliferative potential in this subcluster?

We now show prospective cell cycle phases of subclusters upon wasp infestation in Extended Data Fig. 6e and the genes involved are shown in Extended Data Table 5b. As indicated in the response above, we confirmed that wasp infestation activates proliferation of iPH4 and iPM1 (Reviewer's Figure 13). We thank the reviewer for insightful and detailed analyses.

Figure 13 Subcluster markers and cell cycle phases upon wasp infestation

12) Figure 6 describes a two-dimensional projection of hemocytes in the lymph gland and in circulation at 96 and 120 h AEL. Why merging the two stages?

We now separate two stages in Figure 6a to enhance the clarity of our data presentation (Reviewer's Figure 14).

Figure 14 Comparison of lymph gland and circulating hemocyte transcriptome at 96 or 120 h AEL

13) For the circulating hemocytes, the authors refer to an article submitted to BioRxiv. In that article, several techniques were used for the single cell RNA sequencing. Could the authors confine the comparison to the larval single cells data obtained by Drop-seq, which is the approach used in the present manuscript?

Recent scRNA-seq analytic tools provide several data integration strategies (e.g. Seurat alignment, Harmony etc.) useful to combine multiple datasets produced using different platforms, labs, or samples. In this study, we performed Seurat alignment and label transfer, which utilizes canonical correlation analysis and neighbor searching in latent space to “align” datasets. Using these algorithms, we could successfully integrate samples and find similar subclusters between two origins of fly hemocytes. According to the reviewer’s suggestion, we also re-analyzed and compared the results of Drop-seq datasets with those of the combined datasets. Overall, the results were very similar to those of all combined datasets, with subtle differences in a marker gene levels. We include brief results of analysis of Drop-seq datasets below (Reviewer’s Figure 15).

Figure 15 Comparison of Drop-seq dataset

14) Also, as the manuscript has to stand by itself, could the authors provide a more detailed comparison between the circulating cells and the lymph gland? Could they list the genes that allow the identification of the different subclusters in the circulating hemocytes?

Thank you for the interesting comment. To match cell types in the lymph gland to circulating hemocytes, we performed label transfer analysis using the Seurat v3 R package. This strategy utilizes canonical correlation analysis (CCA) to project datasets into the same subspace, then define mutual nearest neighbors (MNNs; referred to as ‘anchors’ by the authors) between datasets to align cells or pass on information, such as cluster annotation or expression values. Because the analysis is performed in a latent space where axes are defined by different combinations of features (in this case, genes) and neighbor searching in that space, it is infeasible to identify particular genes or gene sets. We briefly describe this process in the Materials and Methods.

The detailed analyses of subclusters in the circulating hemocytes were described in our collaborative work with the Perrimon lab, recently published in eLife⁶. In addition, the list of lineage-specific signature genes is included in Extended Data Table 6.

15) For the PH cluster composed of 67 cells in circulating cells (Fig 6a and c), what is the significance of a gene expressed in less than 50% cells? Also, is there a color coding for the expression intensity?

All genes identified in Fig. 6c are significant DEGs in each cell type of origin. Gene expression levels and FDR-adjusted significance were indicated in the figure as the reviewer suggested. We accordingly inserted a color-coded legend for average expression in the revised Fig. 6c (please find Reviewer's Figure 16 for your reference).

Figure 16 Dot plot of marker genes highly enriched in a lineage-specific or cell type-specific manner.

16) The comparison with the human atlas is potentially very interesting. As it stands now, however, it is not very informative; the authors need to expand this section. First, it seems difficult to identify cells based on a single marker or the authors should at least refer to published data using this criterion. Second, when they compare the fly and the human data to assign similarities, what is the number of genes involved? Can we see the list of those genes and their allocation to the different human cell types? Third, the authors have demonstrated that the prohemocytes are very heterogeneous cells and yet there is a significant similarity between PH (altogether) and HSC-MPP. To improve the significance of the analysis, could the authors reiterate the comparative analysis with the human cells by using either PH1 cells alone, which constitute the pool of dividing precursors, or the rest of the prohemocyte subclusters (that is, devoid of PH1 cells)? Fourth, could the authors provide a more significant evidence for the *Drosophila* hemocytes acquiring signatures of lymphoid lineages upon immune immunity? This statement seems to me a bit farfetched in the present state.

We deleted Figure 7 and the corresponding text in the revised manuscript, as we decided to expand the phylogenetic study in the future (please see Reviewer 1, comment 1). We appreciate the reviewer's valuable comments.

17) The identification of the adipohemocytes is quite exciting. Could the authors perform a labeling for lipid droplets to provide a first validation of these bioinformatics data?

As indicated in the response to point 7, we further verified the presence of lipid droplets with Phalloidin staining and observed that the BODIPY-positive lipid droplet is indeed included in a hemocyte. The new data are added in Extended Data Fig. 3c and Figure 2d (Reviewer's Figure 1).

18) A reference should be added at page 15 line 360.

Line 360 “Furthermore, *Stat92E*⁺ cells are gone upon genetic ablation of the PSC, which indicates that expression of *Stat92E*⁺ PH1 is dependent upon the PSC (Fig. 4d).” indicates a result from our study and thus does not seem to require a reference citation. We thank the reviewer for thoughtful comments.

Minor points:

Could the authors clarify what 0, 2, 4, 6 indicate in extended data Fig 6a, c, d?

The X-Y values of Extended Data Fig. 7a and c (previous Extended Data Fig. 6a, c, d) indicate the natural logarithm expression levels of genes estimated using pseudo-bulk RNA-seq (from scRNA-seq). To avoid the error of natural logarithm, we added a pseudo-value, 1, to the expression level. We have now indicated the X-Y labels in the figures.

Page A9, lines 453-454 are unclear.

We edited the sentence.

The discussion starts with the paragraph that summarizes the data presented in the manuscript. In some instances, the conclusions are a bit overstated.

We changed the conclusions to faithfully summarize our findings.

The analysis at 24h after infestation likely reveals an ongoing response, since the LM1 and LM2 represent a little proportion of the lymph gland cells. This could be mentioned in the text.

We added the idea that 24 h PI denotes an ongoing state of innate immune responses in the Discussion section.

And finally, related to the very first comment, it seems that the lamellocytes LM1 identified in the control animals (without wasp infestation) share several features with plasmatocytes, suggesting that they are produced by transdifferentiation.

We added a discussion of lamellocytes in the control animals.

--

Reviewer #3 (Remarks to the Author):

In their manuscript, Cho et al. provide a single-cell resolution transcriptomic atlas of the *Drosophila* hematopoietic system. They focus their work on the lymph gland, the main hematopoietic organ in the larva, in which they could distinguish specific gene signatures for several subpopulations of progenitor and mature blood cells, revealing a much higher level of cellular heterogeneity than previously thought. They identify new types of hemocytes and they provide new reagents and markers to study lymph gland subpopulations, in particular a prospective stem-like population. In addition, they establish how these hemocyte populations develop during the third instar larvae and how they respond to wasp parasitism. Hence their results bring valuable information about the lineage relationships between the different blood cell populations produced in the lymph gland during normal development and in response to an immune challenge. To give a larger view of the larval hematopoietic system, they also explore the parallels and differences between the gene signature of lymph gland and peripheral larval blood cells. Finally, they provide some novel insights into the similarities between *Drosophila* and mammalian blood cell types by comparing their gene signature.

These findings are particularly important in the field of *Drosophila* hematopoiesis. They bring important and novel information concerning the regulation of blood cell development in the larval lymph gland. It makes no doubt that, even though it is essentially descriptive, this timely work will have a strong impact and will provide a solid basis for deeper analyses of *Drosophila* blood cells diversity and function.

The manuscript is in general very clear and rather convincing. It already contains a load of carefully performed experiments, associated with huge amounts of data to dig in, and a very thorough analysis of the single cell data. Yet, as presented in detail below, some points still really need to be substantiated by a few experiments and several aspects of the manuscript could easily be clarified.

We thank the reviewer for appreciating the work and raising valid questions for further clarification of the manuscript.

Experimental points.

(1) One potential caveat of this single cell analysis of the lymph gland (an immune-responsive organ highly sensitive to stress) is that the cell dissociation procedure might (is likely to!) affect the gene expression profile of the hemocytes (and in a different manner depending on each subpopulation). Consistent with this hypothesis, in their 120h AEL samples, the authors observe roughly the same proportion of crystal cells and lamellocytes (Fig. 1D), while this cell type is normally not present in healthy larvae. It seems important that the authors take this problem into account and present their results with more caution.

Yes, the reviewer is correct. We were aware that the cell dissociation process or the Drop-seq single-cell sorting may induce immediate early stress/immune-responsive genes given the dynamic nature of hemocytes.

1) Induction of immune responsive genes: We dissected and treated lymph glands with extra care and finished the whole procedure (dissection to single-cell library construction) within an hour. Despite the well-controlled process, we did find that immune-responsive genes are significantly induced in the single-cell transcriptome data.

Some genes are differentially expressed in scRNA-seq as compared with bulk RNA-seq data, which are summarized as immune response or metabolic process-regulating genes. We further confirmed that these genes can be induced or reduced by dissociation of lymph glands. Thus, we excluded from downstream analysis those genes that were highly up- or down-regulated compared to bulk RNA-seq (10-fold). We re-analyzed subclustering by aggregating similar subclusters and noticed that subclusters highly expressing immune-responsive genes yet lacking distinct signature genes are removed (Reviewer's Figure 3). We describe our subclustering procedures in detail in the Materials and Methods.

2) Lamellocytes in the lymph gland: Please see our response to Reviewer 2, concern 1.

We agree that the procedure might have affected gene expression and induced the lamellocyte differentiation. We added a discussion of this topic in the revised manuscript.

(2) May be there is a trivial explanation but I was very surprised that with 100 to 150 dissected lymph glands (i.e. 50.000 to 400.000 cells, according to Fig 1B), the authors obtained valid sequencing data for only a few hundreds to a few thousand cells in each library (Ext data 1a). That gives a recovery rate of $\pm 1\%$, which seems worryingly low. How can it be explained?

While applying hemocyte samples to the Drop-seq, we noticed that careful handling of the cell flow is critical and that the capturing rate largely relies on the flow speed. We also ran the Drop-seq for circulating hemocytes (now published in Tattikota et al., 2020⁶) and observed similar capture rates.

(3) One major claim of the authors is the identification of new cell types (in particular the adipohemocytes, the GST-rich cells and the PH1/stem-like prohemocytes), with specific gene signatures. However, the presence of these populations in wild-type lymph gland is not always shown convincingly and further experiments are clearly needed to substantiate their conclusions. Accordingly, the following points should be addressed:

We thank the reviewer for constructive suggestions. Owing to the critique, we have further validated expression of representative marker genes in each cell type using the SABER-FISH assay. We hope the reviewer finds the mRNA expressions indicative of each cell type valuable and in agreement with our assessment.

(3a) Concerning the GST-rich population: the authors only show (Fig. S1H) that the expression of the top 10 genes of the GST-rich cells is also detected in the bulk RNA-seq. That doesn't prove that they make a distinct population in wild-type lymph gland (rather than being artefactually induced in the dissociation process). Actually, the authors also tested a few *Drosophila* lines to analyse the expression pattern of some of these genes (Supl Table 3A), but none of these lines revealed an expression in wild-type lymph glands. In the absence of such an indication it is not possible to conclude that the GST-rich population is genuine. I prompt the authors to try other tools to validate the expression of the GST-rich cells in wild-type lymph gland, may be using RNA in situ hybridisation for the most specific markers or GFP-tagged version of the endogenous gene if the corresponding MiMiC line exist.

We validated the expression of two GST-rich-specific transcripts, CG18547 and CG3397, by SABER-FISH. Both CG18547 and CG3397 sporadically localize adjacent to Tep4-positive early prohemocytes (Extended Data Fig. 3a). Neither CG18547-positive cells not CG3397-positive cells co-localize with Tep4-gal4-expressing cells yet they are closely adjoined. These results indicate that the GST-rich population is present in the lymph gland and likely emerges following Tep4⁺ PH differentiation (Reviewer's Figure 9).

(3b) To reveal the adipohemocyte population in wild-type lymph glands, the authors used different GAL4 lines (crq-Gal4, Ama-Gal4 and Lsd2- Gal4; Fig 2). Whether these lines reflect the expression of the endogenous gene is subject to caution. In particular, it remains to be shown that the higher level of RedStinger observed in the prospective NimC1-low/adipohemocyte population (for crq-GAL4 and Lsd-2-GAL4; cf. line 246) reflects crq or Lsd-2 expression levels. Again, fluorescent in situ hybridisation or immunostaining (since an antibody is available for Crq) coupled with immunostaining against NimC1 should clarify the matter.

To sufficiently confirm the presence of adipohemocyte, we validated two representative transcripts enriched in adipohemocytes – Sirup and Lsd-2 – in the lymph gland. Both transcripts are detected within the cortical zone; however, Sirup or Lsd-2 expressing cells are devoid of the cortical zone marker, Hml (Extended Data Fig. 3b). This pattern is reminiscent of the computational analysis shown in Fig. 2b. In addition to the gene expression analysis, we now show that a few hemocytes in the lymph gland indeed enclose Nile red- or BODIPY-positive lipid droplets (Fig. 2d; Extended Data Fig. 3c), identifying a novel lipid-containing hemocyte in the lymph gland (Reviewer's Figure 1).

(3c) Along the same line, the authors claim (line 242; Fig. S2h) that zfh1 is a prohemocyte marker based on the use of a zfh1-GAL4 line (and on their scRNA-seq data). Given that

antibodies against Zfh1 as well as a GFP-tagged version of the gene (cf PMID: 30002131) are available, I strongly advise the authors to use one of these tools to verify their conclusion (in our own experience, Zfh1 immunostaining revealed a rather ubiquitous pattern in the lymph gland, which was also observed with a *zfh1*-GAL4 line different from the one used here). More generally, the authors should be more cautious (and more precise) in their conclusions concerning the expression of different genes when it is based solely on GAL4 lines.

We agree with the reviewer that enhancer-based expression could be misleading and may not be adequate to reflect the endogenous gene expression. The purpose of presenting *zfh1*-gal4 was to report a new enhancer trap line that displays lymph gland expression and can serve as a resource for the community. Due to the lack of reagents under current circumstances, we cannot further characterize and fully understand the endogenous expression patterns of *zfh1* in the lymph gland. Hence, we deleted the *zfh1*-gal4 image in the current manuscript and hope to carry out a comprehensive analysis in the future.

(3d) For the PH1 population, the authors show some more convincing evidence. However, they should clearly mention that the STAT92E::edGFP is a reporter of the JAK/STAT pathway and not a reporter of STAT92E expression. It is thus misleading to call these cells STAT92E+. Actually, the authors could also look at the endogenous STAT92E protein. Also, given that the PSC was proposed to express *Upd3*, it would be interesting to test the effect of a PSC-specific knock-down of *upd3* on the activity of STAT92E-edGFP.

As the reviewer pointed out, STAT92E::edGFP is a reporter for the JAK/STAT pathway, and therefore, we changed the term “STAT92E-positive” to “STAT92E-active (STAT92E^{act})” to provide a more precise description.

We now corroborate whether the expression of STAT92E reporter is dependent upon *upd3* ligand in the PSC. Surprisingly, loss of *upd3* (*upd3*Δ^Y; STAT92E::edGFP and *upd2*Δ^Y; *upd3*Δ^Y; STAT92E::edGFP) or expression of an RNAi reagent that targets *upd3* in the PSC (*Antp*-gal4 UAS-*upd3* RNAi; STAT92E::edGFP) does not alter STAT92E activity, indicating that *upd3* is not a major trigger of STAT92E activation (Extended Data Figure 5g). Given our analysis showing that the genetic ablation of PSC (*col*-gal4 UAS-*hid*,*rpr*; STAT92E::edGFP) causes a stark reduction of STAT92E activity as well as Delta protein expression (Extended Data Figure 5j; Figure 4d), it is clear that the PSC is required for PH1 maintenance. It will be interesting to further characterize a ligand essential for STAT92E activation and its role in the PH1. We now include the data in the revised manuscript (Reviewer’s Figure 17).

Figure17 Loss of *upd3* in the PSC does not alter the expression of STAT92E

(4) The identification of the PH1/stem cell population is a major result. However, the authors should put in perspective their results with those of Dey et al. (ref 76), who described a putative Notch-expressing HSC population in the early larval lymph gland.

We added a discussion of the previous study, Dey et al.

(5) The authors suggest that the Notch pathway is active in the PH1 cells but they do not bring strong evidence for this. As for the JAK/STAT pathway, they should investigate Notch pathway activity using the equivalent *Su(H)GBE-edGFP* reporter (cf. He et al., PMID 31140975). They should also show whether Delta expression (or Notch activity) depends on the PSC, as shown for *STAT92E-edGFP*.

In addition to Delta protein expression, we now show that a few DI-expressing cells co-localize with *Su(H)-GBE*-positive cells in the lymph gland (Extended Data Fig. 5i). This result corresponds well with the results of the computational analysis, which suggests that there are mixtures of Δ^+ , Δ^+/N^+ , and N^+ cells, and indicate an intriguing interaction between Delta and Notch in the PH1.

Moreover, we now show that loss of the PSC significantly alters Delta expression, as revealed by *STAT92E::edGFP* (Extended Data Fig. 5j). Future studies will allow us to gain additional insights into the intricate communication between Delta and Notch in PH1 and PH1 cell interactions with neighboring PSC cells (Reviewer's Figure 18).

Figure 18 Delta expression with Notch reporter (left) or upon the PSC ablation (right)

(6) Using the Gtrace lineage labelling technique, the authors show (Fig. 4G) that DI-GAL4+ cells close to the PSC “produce hemocytes of the entire lymph gland” (line 367) and they conclude that PH1 cells are reminiscent of mammalian HSC. Even though DI>GTRACE seems to label a large fraction of the lymph gland, it does not demonstrate unambiguously that PH1 cells (or DI>Gtraced cells) give rise to all the different types of hemocytes (i.e. that they have multilineage differentiation capacity). The authors must show, using immunolabelling with specific markers, that the major differentiated cell types (plasmacytes, adipohemocytes, crystal cells and lamellocytes) are indeed labelled with DI>Gtrace.

We now provide data showing that Delta-gal4 positive cells can give rise to the three representative hemocyte types, $NimC1^+$ plasmacytes, $PPO1^+$ crystal cells, or $L1^+$ lamellocytes. These data support our idea that Delta-gal4 positive cells retain the potential to generate all lymph gland cell types. These data are now included in Extended Data Fig. 5k (Reviewer's Figure 19).

k*Delta*^{BL45136}>*GTRACE*
Figure 19 Delta+ cells give rise to three types of hemocytes

(7) In their wasp infestation experiments, the authors found no change in the gene expression signature of the PSC cells. This is unexpected given the central role of these cells as a relay required to induce lamellocyte differentiation in the lymph gland, and several papers observed the (transcriptional) induction of different signalling pathways in these cells following wasp infestation (see for instance Louradour et al. PMID: 29091025; or Sinenko et al. PMID: 22134547). This possible inconsistency should be discussed.

The data was unexpected and lead to a number of possible hypotheses. First, the PSC might have gone through transcriptional modifications immediately after wasp infestation. In this case, we may not be able to see dramatic changes at the 24 h PI time point. Second, it is possible that the main modifications of the PSC are translational or post-translational, not transcriptional. This idea is supported by the results of a previous study⁷ that reported post-translational control of the EGF ligand/Spitz by Rhomboid and its role in the lamellocyte differentiation. We discuss this in the revised manuscript.

(8) In Fig. 5C, Delta staining does not show any background outside the “medullary zone” in uninfested larvae, while there is quite some background in infected larvae or in Fig.4F.

Yes, we see Delta background staining in the medullary zone. The pattern sometimes appears in cells at the medial-anterior side of Delta⁺ cells. We assume that this pattern could be caused by long Delta protein perdurance considering very low Delta mRNA expression in PH2 and PH3. Of note, Delta staining shown in Fig. 5c indicates the lymph gland at 96 h AEL but lymph glands in Fig. 4e-f are 120 h AEL.

In addition, DI staining does not seem to be mainly at the cell membrane (especially in Fig 5C) and the way the immunostaining is performed (i.e. with a 30' incubation of dissected lymph glands in 10mM EDTA) is strange (anti-DI from DSHB has been used in many studies but I did not see that they used a similar protocol). Since a 30' incubation with 2mM EDTA causes Notch pathway activation in cell culture (PMID: 17545467 and others), the authors should repeat their immunostaining without such treatment and/or use one of the published available DI-GFP knock-in lines (e.g. PMID: 26102525 or PMID: 31668010) to confirm their results concerning DI expression.

While improving the Delta staining protocol, we came across previous literature showing that EDTA treatment inhibits the metalloprotease Kuzbanian and consequently keeps the Delta extracellular domain intact in S2 cells⁸. Interestingly, EDTA reduces background staining and enhances the visibility of Delta in the lymph gland.

We corroborated the new staining protocol as the reviewer suggested. First, we visualized Notch activity after a 30-minute treatment of EDTA and confirmed that EDTA does not induce reporter expression (Reviewer's Figure 20, top). Second, we confirmed that Delta RNAi clones do not show anti-Delta staining (Reviewer's Figure 20, middle). Finally, we established that Delta staining with EDTA results in identical, or even better, staining patterns in other tissues including the brain or the disc (Reviewer's Figure 20, bottom).

Albeit preliminary, we assume that Delta in the PH1 is placed in a distinct extracellular environment and EDTA helps maintain Delta in the lymph gland.

Figure 20 Validation of anti-DI staining

(9) Concerning the differences between circulating and lymph gland hemocytes. (a) Ubx staining (Fig. S6g,h) is clearly cytoplasmic. Do the author have any explanation for this?

Yes, the reviewer is correct. We see Ubx staining in the cytoplasm of circulating hemocytes different from the embryo. We suspect that Ubx may undergo nuclear shuttling upon certain challenges or Ubx may carry out an unexpected function even without going into the nucleus. It will be interesting to understand the novel function of Ubx in hematopoiesis or immunity in the future.

(b) A transcriptome of the circulating crystal cells has been published (Miller et al., PMID 27487438). How does it compare with crystal cells clusters identified in this work? Notably Miller et al observed Oscillin expression in circulating crystal cells whereas the authors suggest that this gene is specific to the lymph gland crystal cells.

Owing to the reviewer's point, we further analyzed the relationship between the crystal cell transcriptome⁹ (Miller et al., Y axis in Reviewer Figure 21) and the lymph gland scRNA-seq (X axis in Reviewer Figure 21) and confirmed their correlation. Moreover, we compared the expression of Oscillin in circulating crystal cells with one in the lymph gland crystal cells and identified that circulating crystal cells also exhibit Oscillin expression but at significantly lower levels. We would like to note that a comparison between the circulation and the lymph gland relies on relative values of mRNA expression, not on absolute values as below (Reviewer's Figure 21).

Figure 21 Correlation between the crystal cell transcriptome and the lymph gland scRNA-seq

Other points.

(1) Line 44-45: it seems odd to start the introduction with a reference to “memories of immunological events” since this is not a prevalent feature for insect blood cells.

We rephrased the sentence.

(2) Line 62: I don't know why the authors state that the larval hematopoiesis is the “definitive” one. It has been shown that both embryo- and lymph gland-derived hemocytes contribute to the adult hematopoietic system, so both waves could be considered as definitive!

We changed “definitive” to “larval” or “lymph gland” hematopoiesis in the revised manuscript.

(3) Line 64-66: I'm not sure that freely circulating and sessile hemocytes should be described as distinct populations: sessile hemocytes can enter circulation and vice versa.

We removed 'distinct' and edited the sentence.

(4) Line 68-69: the reference is missing (Mandal et al. 2004). And only part of the lymph gland was shown to arise from hemangioblast-like cells.

We added the reference, Mandal et al., 2004.

(5) Line 68-70: the introduction of the overall lymph gland morphology/organisation is not very clear.

We rephrased the description and added a schematic to better explain the lymph gland (Figure 1a).

(6) Line 83: not every wasp species infest *Drosophila melanogaster*.

We specified the wasp species as *Leptopilina*.

(7) Lines 87-88: lymph gland hemocytes do not “remain intact” at 24h post-infestation: they are already affected/start to differentiate in lamellocytes (see for example Louradour et al. 2017).

We meant to describe that the lymph gland remains in place. We edited the sentence accordingly.

(8) Lines 97-99: some adequate references could be included.

We added references for lines 97-99.

(9) Lines 321-324: the corresponding references should be included (notably: Minakhina & Steward, 2010; Krzemien et al., 2010; Dey et al. 2016).

We added the references.

(10) Lines 448, 449, 451 and 459: these refer to extended Data fig 6 (not 5).

We corrected the typo in the figure. We thank the reviewer for careful assessment.

(11) Line 511: why do the authors think that they have revealed “genetic” differences in hemocytes derived from the two waves of hematopoiesis?

We changed ‘genetic’ to ‘expression’.

(12) Line 513-515: actually, there was already a scRNA-seq paper for invertebrate blood cells (in mosquitoes: PNAS 2018; PMID 30038005). It should be cited.

We added the reference.

(13) Lines 526-528: as mentioned above, this is not very convincing.

We now show the expression of GST-rich specific mRNAs (Extended Data Fig. 3a) and edited the sentence accordingly.

(14) Lined 568-570: this is vague. What did we learn from these publications concerning the role of these pathways in the LG (and potentially in HSC-like cell/progenitor maintenance)?

We edited the sentence.

(15) Lines 601-611: when possible, I would suggest that the authors cite the relevant publications for each stock (for instance He et al., for STAT92E-edGFP, or Evans et al. for UAS-Gtrace)

We added references.

(16) Lines 992-994: the number of biological replicates used for the bulk RNAseq should be indicated.

We performed only one bulk RNA-seq experiment for each timepoint; i.e. 72, 96, and 120 h AEL. This experiment was only to confirm the expression.

(17) Line 354: as mentioned above, there is some confusion between the “expression of genes” and that of reporter lines.

We changed the description.

(18) Line 928: which publication is Sudhir et al.?

We added a full citation.

(19) Line 1003: the strain of *Leptopilina boulardi* that was used should be mentioned.

We indicated the strain we used (G486).

(20) Lines 1035-1038: it seems that the fixation step is missing.

We added the fixation step in the method.

(21) Figure 4 panel b -right panel: I observed an alteration of the red channel in a part of the panel. This is probably due to a mishandling during figure preparation and should to be corrected.

We now add "col_low" in the figure to clarify the labeling. We do not see the alteration of the channel.

(22) Extended data table S2: the legend is not clear. What are the values in the columns C to G and how were they calculated?

We changed the labels to make them clear to a broader readership.

Lucas Waltzer

--

Reviewer #4 (Remarks to the Author):

Summary:

In this paper, Cho, Yoon et al profile the transcriptomes of the *Drosophila* lymph gland different cell types at different stages of development (72, 96, and 120 hours after egg laying) using Drop-Seq to achieve single cell resolution. They then use their single-cell atlas to address several interesting questions regarding the development and evolution of these cell types.

The authors first use known markers to identify the different cell types of the *Drosophila* lymph nodes. They identify all cell types, as well as 2 more that they name adipohemocytes and GST-rich. They then subcluster these cell types into smaller clusters. They find that these subclusters represent either populations at different cell cycle stages or populations at different levels of maturity. They use this information as well as a trajectory inference algorithm (Monocle) to define the developmental trajectories of the different cell types. The authors then look at the differences that arise in the lymph node cell type composition upon wasp infestation, as well as the transcriptional differences between the same cell types before and after infestation. They show that lamellocytes are generated by prohemocytes and plasmatocytes. The authors then compare the hemocytes that are in circulation (that are generated during the embryonic hematopoiesis) with the ones that reside in the lymph gland (generated by the larval definitive hematopoiesis). They find that the two populations are very similar, with few differences that are not very convincing. Finally, they compare the *Drosophila* hemocytes with the human immune cells based on ~6,500 orthologous genes. They find that the *Drosophila* hemocytes correspond pretty clear to the human myeloid immune cells. Interestingly, they observe that *Drosophila* hemocytes upon infestation acquire characteristics of human lymphoid cells.

This paper is a great resource as single-cell transcriptomics paper and it addresses several interesting questions. However, although it touches upon different subjects, it falls short of proving any of the claims that are made. Moreover, many of the observations that are very interesting (i.e. new cell types, different subtypes, slight differences between the different *Drosophila* hemocyte lineages, similarities and differences to human cells) are very superficially presented and are not convincing. Although the dataset has a lot of potential, it is often presented as a list of genes at different types/subtypes/stages that are not

interesting to read, except by hemocyte aficionados. I would recommend publication as a resource paper, after a) toning down a few of the claims that are made (please see Major points), b) highlighting that it should be treated like a resource, and c) re-writing it to make it more appropriate for a broad audience.

Major points:

1) The GST-rich and adipohemocyte populations are not convincing. The authors need to show these populations in the tissue with appropriate counterstains. Otherwise, they could mention that the data suggest their presence, but it remains to be shown where they are.

We thank the reviewer for constructive suggestions. We further verified the expression of marker genes for the intermediate zone, adipohemocytes, GST-rich, and most of the PH subclusters. Please see our response to Reviewer 2, concern 7.

Also, it is unclear what the comparison to the bulk is trying to achieve. Of course, if there are single-cells expressing these genes, the bulk tissue will also express them. These does not prove that these are a specific cell type.

We agree with the reviewer's comment. We originally intended to show the expression of the markers in bulk RNA-seq as well. We have now verified specific markers of adipohemocytes and GST-rich populations using SABER-FISH experiments in Extended Data Fig. 3a-c and Figure 2d (Reviewer's Figure 1 and 9).

2) Related to the previous point, most of the antibody stainings that are presented are poor and lack appropriate counterstains.

We now show all the images with representative lymph gland markers. For example, Extended Data Fig. 2g, 2h, 3a, 4j, and Fig. 4a, are displayed with a prohemocyte marker, *Tep4*; Extended Data Fig. 2i, 3b, 3d, 3g, 4i, 7d, 7e, and Fig. 3d are with a plasmatocyte marker, *Hml* or *Pxn*; Extended Data Fig. 3e, 5e, 5g, Fig. 4a, 4c, 4e, 4g, and 5c are with a PSC marker, *Antp*.

Images presented in the manuscript are comparable to images found in the literature for this field¹⁰⁻¹².

3) The identification of subclusters is meaningless unless the authors provide in situ stains that verify these subclusters. Moreover, given the fact that these subclusters separate cells at different stages of the cell cycle or their development, I would be very cautious in calling them subclusters – they might as well be a continuum that the authors are arbitrarily separating them in clusters.

We agree with the reviewer that the subclusters presented in this study could be hemocytes at different developmental stages that are all linked in a continuum. However, we would like to note that subclusters previously grouped exhibit expressions of distinct genes enough to be distinguished as subclusters. Indeed, previous studies have classified developmental stages of crystal cells or lamellocytes^{1,13} and recent studies on the single-cell transcriptome of circulating hemocytes also provide evidence for similar subclusters^{6,14}. Different from mature hemocytes, diverse state or group of prohemocytes is less recognized. Therefore, our classification would help to extend previous hypotheses in the field and suggest possible diversity in hemocyte development and differentiation.

To avoid continuum subclusters of hemocytes, we developed a new subclustering method that aggregates similar subclusters (pseudo-bulk Pearson correlation ≥ 0.95) to form a super-group, which greatly simplified the PH and PM populations. We compared the number of DEGs using different resolution values ranging from 0.1 to 3.0 and select a value that resulted in the largest number of DEGs. A resolution with maximum DEGs was chosen as

an optimal resolution. Using this method, we identified six prohemocytes and four plasmacyte subclusters. PH1 and PH2 remain almost the same as our previous result, while others, except for two 120h AEL specific subclusters, are merged into a pan-PH group PH4. The majority of PM subclusters are grouped into a pan-PM group, PM1, except for three 120h AEL specific PM subclusters. In summary, we defined a total of 17 subgroups from the normal lymph gland data set (Fig. 2 and Extended Data Fig. 2): 6 PHs, 4 PMs, 2 each for LMs and CCs, one each for PSC, GST, and adipohemocytes, along with three non-hematopoietic cell types (DV, RG, and neurons). We described the analytic procedures in the “Subclustering analysis” section of the Materials and Methods. The newly defined subclusters and their marker genes are updated in the manuscript and Reviewer’s Figure 22.

Figure 22 New subclustering of lymph gland hemocytes (also shown in Figure 2)

As per other reviewers’ suggestions, we now visualize most of the markers for each subcluster: PH1 (Delta, STAT92E^{active}), PH2-PH3 (Tep4, collier^{low}), PH4-PH5 (Tep4, Ance, Nplp2), PM1 (Nplp2, Hml, Pxn), PM2 (NimC1), PM3-PM4 (vir-1), GST-rich (CG18547, CG3397), adipohemocytes (Lsd-2, Sirup) by SABER-FISH or by antibody/reporter/gal4 expression. The results are now shown in Extended Figure Data 2g, 2h, 2i, 3a, 3b, 3d, 4g, 4h, 4i, 4j, and Figure 4 (Extended Data Table 4; Reviewer’s Figure 8 and 10).

I also had a hard time understanding from the Methods how they decided the number of the subclusters, given the fact that by increasing or decreasing the resolution, one can have as many subclusters as they want. They claim that they manually inspected the tSNE plots, if I understood correctly. I would remove this part.

We agree with the reviewer’s concern that the number of clusters can change with different resolution values. In our past analysis, we selected minimum resolution values to separate minor but distinctive populations such as PH1 and PH2. Meanwhile, we noticed that some genes are enriched in scRNA-seq compared to bulk RNA-seq, which are summarized as immune response or metabolic process-regulating genes. We further confirmed that these genes can be induced or reduced by dissociation of lymph glands. To mitigate the effects of these genes and avoid continuum subclusters of hemocytes, we redefined PH and PM subclusters, excluding those highly up- or down-regulated compared to bulk RNA-seq (10-

fold). For optimal subclustering, we developed a new automated optimization method that uses an iterative procedure as follows.

We hypothesized that an optimal resolution value would result in a maximal number of DEGs between subclusters after similar subclusters (Pearson $r \geq 0.95$) were merged to form a super-group. Thus, we tested a range of resolution values from 0.1 to 3.0, increasing by 0.1 to make clusters and transformed the UMI count matrix to pseudo-bulk to calculate the correlation matrix using 2,000 variable genes. Next, we iteratively correlated and excluded the least correlated subcluster (Pearson $r < 0.90$) until a minimum correlation coefficient exceeds 0.9. Then, we again iteratively correlated and aggregated two most correlated subclusters (Pearson $r \geq 0.95$) into a single super-group until the maximum correlation coefficient is lower than 0.95. A description of this method was added to the Materials and Methods.

4) The lists of genes that are presented at different stages of the paper are very tiring and seem useless. They can all be summarized in Tables and removed from the main text.

We now provide Extended Data Table 3 for subcluster-specific genes.

5) The differences that the authors identified between the two different hemocyte lineages (a couple rRNAs and a couple CGs) are very unconvincing. Then, the authors try to compare specific cells types between the two lineages and come up with a couple more differences. Unless they provide stainings of the different populations to show these differences, I do not think that these differences are convincing.

In the analyses, we sought to claim that hemocytes originated from the two lineages are more or less the same except for a few differential genes, including Ubx. We changed the text in the revised manuscript.

Minor points:

1) Why do the authors use these developmental stages (72, 96, and 120 hours AEL)? Is there anything significant in these stages in terms of hemocyte development?

Wild-type lymph glands usually follow a fixed developmental process unless otherwise challenged. At 72h AEL, which commonly corresponds to the late second to early third instar, the lymph gland establishes the prohemocyte-containing medullary zone and initiates differentiation of mature hemocytes. At 96h AEL, both the medullary zone and the cortical zone are organized and the lymph gland forms the structure shown in Figure 1a (lymph gland schematic). At 120h AEL, the cortical zone expands, while the medullary zone shrinks, and the organ is getting ready for the disintegration upon pupariation. The significance of each stage is now included in the introduction.

2) The Introduction is very hard to read for non-experts - it would benefit greatly from a schematic.

Thank you for the valuable suggestion. We now include a schematic in Figure 1a.

3) The authors focus their analysis on cells with many reads/genes per cell. While this is good, they might be selecting for highly transcribing cells. If they relax their criteria to choose specific cells, do they still see the same populations?

We did not particularly select cells sequenced with a large number of genes/UMIs. In our analysis pipeline, we filtered cells with too many UMIs (more than $\langle \text{mean} + 2 \cdot \text{stdev} \rangle$) or too few genes (less than 200 genes). This resulted in populations covering a wide range of UMI or gene counts; for example, in the normal lymph gland dataset, the median of the number of genes (UMI) is 1,473 (6,306) ranging from 403 to 4,045 genes (from 792 to 32,303 UMIs).

To clarify the number of genes or UMI counts for different cell types, we attach relevant figures below (Reviewer's Figure 23).

Figure 23 The number of UMI counts (top) or genes (bottom) of each cell type

4) Figure 1c has a couple typos (e.g. missing comma in 8218 and crystal cells)

We corrected typos in Figure 1 (Reviewer's Figure 24).

Figure 24 New Figure 1c

5) It is unclear what the SCENIC output offers to this paper. I would remove or explain better.

Now we add a detailed description of the SCENIC output.

1. Anderl, I. *et al.* Transdifferentiation and Proliferation in Two Distinct Hemocyte Lineages in *Drosophila melanogaster* Larvae after Wasp Infection. *PLoS Pathog* **12**, e1005746 (2016).
2. Rommelaere, S., Boquete, J.P., Piton, J., Kondo, S. & Lemaitre, B. The Exchangeable Apolipoprotein Nplp2 Sustains Lipid Flow and Heat Acclimation in *Drosophila*. *Cell Rep* **27**, 886-899 e886 (2019).
3. Jang, Y.H., Chae, H.S. & Kim, Y.J. Female-specific myoinhibitory peptide neurons regulate mating receptivity in *Drosophila melanogaster*. *Nat Commun* **8**, 1630 (2017).
4. Ilicic, T. *et al.* Classification of low quality cells from single-cell RNA-seq data. *Genome Biol* **17**, 29 (2016).
5. Krzemien, J. *et al.* Control of blood cell homeostasis in *Drosophila* larvae by the posterior signalling centre. *Nature* **446**, 325-328 (2007).
6. Tattikota, S.G. *et al.* A single-cell survey of *Drosophila* blood. *Elife* **9** (2020).
7. Sinenko, S.A., Shim, J. & Banerjee, U. Oxidative stress in the haematopoietic niche regulates the cellular immune response in *Drosophila*. *EMBO Rep* **13**, 83-89 (2011).
8. Qi, H. *et al.* Processing of the notch ligand delta by the metalloprotease Kuzbanian. *Science* **283**, 91-94 (1999).
9. Miller, M. *et al.* Control of RUNX-induced repression of Notch signaling by MLF and its partner DnaJ-1 during *Drosophila* hematopoiesis. *PLoS Genet* **13**, e1006932 (2017).
10. Morin-Poulard, I. *et al.* Vascular control of the *Drosophila* haematopoietic microenvironment by Slit/Robo signalling. *Nat Commun* **7**, 11634 (2016).
11. Jung, S.H., Evans, C.J., Uemura, C. & Banerjee, U. The *Drosophila* lymph gland as a developmental model of hematopoiesis. *Development* **132**, 2521-2533 (2005).
12. Benmimoun, B., Polesello, C., Haenlin, M. & Waltzer, L. The EBF transcription factor Collier directly promotes *Drosophila* blood cell progenitor maintenance independently of the niche. *Proc Natl Acad Sci U S A* **112**, 9052-9057 (2015).
13. Mukherjee, T., Kim, W.S., Mandal, L. & Banerjee, U. Interaction between Notch and Hif-alpha in development and survival of *Drosophila* blood cells. *Science* **332**, 1210-1213 (2011).
14. Cattenoz, P.B. *et al.* Temporal specificity and heterogeneity of *Drosophila* immune cells. *EMBO J*, e104486 (2020).
15. Krzemien, J., Oyallon, J., Crozatier, M. & Vincent, A. Hematopoietic progenitors and hemocyte lineages in the *Drosophila* lymph gland. *Dev Biol* **346**, 310-319 (2010).

Reviewers' Comments:

Reviewer #2:

Remarks to the Author:

Response to the revision, Reviewer #2

This is an extensive revision of the manuscript from Shim and collaborators. The authors reanalysed the data and added a significant amount of new data to comply with the comments of the reviewers. This has significantly improved and strengthened the manuscript. Several points need still to be addressed to further clarify the interpretations and to simplify the manuscript. The effort is worth, in view of the fact that it represents the first detailed analysis of the lymph gland and will provide important tools.

Related to points 1 and 2:

The goal of this manuscript is to define the transcriptional asset of the lymph gland and will be taken as a reference by the whole community. The authors should make their data meaningful and useful as much as possible. The finding that the dissociation technique affects the inflammatory status of the hemocytes is already an important piece of information. While it is clear that their approach is the best at the present time, some measures should be taken to clarify the significance of the presented sc RNAseq analysis.

Concerning the lamellocytes in normal condition, what markers have been used to define LM1 as lamellocytes? The only markers displayed in Fig 2b are Atilla and Msn which are not specifically enriched in LM1 compared to other PM clusters and the manuscript mentions that only Atilla was used to define the population. How are L4, L5 and PPO3 expressed in LM1 compared to the other clusters? Why were they not used?

Concerning the method to remove the bias due to the dissociation of the lymph gland, were the eliminated genes mostly expressed in lamellocytes? The authors should test by immunohistochemistry or in situ the expression of early lamellocyte markers (LM1) as well as that of a selected number of genes that are not specific to lamellocytes and that were eliminated upon comparison with the bulk data. It could well be that a number of those genes is indeed already expressed, albeit at lower level, (and/or in few cells) in control lymph glands. By totally neglecting these genes without an in vivo validation, the output of the analysis will be biased.

It seems risky to remove 4830 genes (almost a third of the *Drosophila* transcriptome) from the analysis based on the comparison of bulk RNAseq of intact lymph gland and single cell RNAseq of dissociated lymph gland. The difference in expression levels can be due to technical differences in the sequencing techniques independently from the dissociation of the lymph gland. I would suggest the authors to validate this strategy by using identical technology (qPCR or bulk RNAseq) to quantify the impact of the dissociation on the expression levels in lymph glands.

Related to point 3:

The authors merged the sc RNAseq data from the different developmental stages to identify the clusters. However, the data on the number of cells/cluster at the three stages (extended table 2) clearly indicate that 1) the 96h AEL stage is largely over represented 2) many clusters only appear at 120h AEL, 3) in early stages, many clusters are made of very few cells, undermining their identification.

More than 9 000 cells were analysed at 96h AEL, when the primary lobe contains in average less than 1 400 cells, whereas less than 8 000 cells were analysed at 120 h AEL, when the total n of cells reaches 5 000 in average. Because of the over representation of the 96h AEL stage, the data on the PH4 and the PM1 clusters mostly arise from that stage (e.g. PM1: 5 259 cells at 96h AEL vs. 1 340 cells at 120h AEL) and the same is true for the other clusters as well. PH5, PH6, PM2 PM3, PM4, LM1 and the adipohemocytes are basically only present at 120h AEL, reducing significantly the diversity at early stages. Indeed 120 h AEL is the only stage that really shows a significant heterogeneity of hemocytes. Overall, the merge of the data from the three stages to characterize a cluster introduces a strong bias in the interpretation of the sc RNAseq data.

The authors compare the clusters defined by the analysis of all lymph gland stages combined with the clusters defined by the independent analysis of each stage. They do not find any significant

difference between the two approaches and illustrate their conclusions with heatmaps of the main markers of PH and PM clusters at each stage (Extended Data Figure 2c,d). However, the list and order of the markers used for each heatmap change from one stage to the next, which renders the comparison difficult to visualise. Also, the name of the clusters changes at the different stages (PM1 and PM1 mature at 72h AEL; PH2/3 as well as PH3/4 at 120h AEL). In addition, the comparison of the two approaches is illustrated by coloured bars above each Heatmap, which does not allow a clear appreciation of the similarities between the two approaches. For example, several clusters seem to be joined in the split approach and separated in the combined approach (Extended Data Figure 2C). This appears to me as a significant difference between the two approaches. Could the authors illustrate the similarities by displaying dotplots (like in Figure 2b) with the top 5 markers of each clusters in the combined approach and the split approach (one per stage)? If the two clustering strategies are highly similar, the main markers will be conserved and clearly visible with this representation.

Related to points 5 and 6

The PH cells likely provide a pool of renewing precursors, in line with their constant number over time and their position within the lymph gland. The conclusion on the presence on stem-like cells awaits nevertheless *in vivo* validation. PM1 cells are also proliferative and their number also stays constant, but despite the proposed mechanisms, their destiny is not clear. Here and throughout the manuscript, the authors indulge in speculations that at time dilute out the most important message of the manuscript, i.e. the presence of heterogeneous hemocyte populations in the lymph gland hematopoietic tissue and the identification of novel cell types. The revision has already significantly improved the quality of the data, but a further simplification would still benefit to the manuscript, which remains very dense. Perhaps some speculations could make the object of a review article.

Related to point 7:

To confirm the presence of adipohemocytes, the authors co-labelled lymph gland for Sirup or Lsd2 and Hml (Hml Δ > GFP). The data would be more convincing with a positive marker of adipohemocyte such as Srp or eater, which seem strongly expressed in this cluster. Similarly, for the Gst-rich cluster, co-labelling with Nplp2 or Hml should provide a positive co-labelling. Also, since the Nile Red positive cells are not colabeled with an adipohemocyte marker, it is hard to correlate the two labelling directly. As a minimum, the authors should compare the position and the percentage of cells that are Nile Red positive with those of the cells that are positive for an adipohemocyte marker.

Perhaps the authors could accompany the schematic drawing in panel 4H with the name of the major markers used to identify the different clusters. Also, they do show the location of the PSC, perhaps they could also indicate the other zones that constitute the gland (intermediate, cortical, etc).

Related to point 13

In Reviewer Figure 15, the authors compare the Drop-seq data with InDrop-seq data and illustrate the comparison with a dotplot of the main cluster markers. Since Ubx represents the strongest marker of circulating cells, this dotplot has to include Ubx and ideally all the markers shown in Figure 6c to be convincing.

Figure 6a shows a relatively similar profile for the embryonic and larval derived hemocytes. Nevertheless, the profiles shown in reviewer figure 15, which presents the drop-seq data for embryonic and larval derived hemocytes, are quite different. How can this discrepancy be explained?

Related to point 14:

The authors say that "it is infeasible to identify particular genes or gene sets". If I understand properly, the authors say that it is not possible to identify genes expressed at different levels in the lymph gland compared to circulating hemocytes. If this is the case, what is the validity of the analysis presented in Extended Data Fig7?

Related to point 15:

The comment was related to the significance of markers expressed in low percentage of cells of the clusters. I understand the statistical significance of the markers, however, the biological significance is less clear to me. For example, Ubx is expressed in 25% of the circulating cells, significantly enriched in circulating PM and absent from the lymph gland (Figure 6c). Could that mean that Ubx is expressed exclusively in a subcluster of circulating PM? In addition, the immunolabelling of Ubx in circulating cells (Extended Data Fig 7) seem to indicate that all circulating PM express Ubx. Would it be possible to quantify the number of circulating PM expressing Ubx to support the single cell analysis?

It is very likely that the embryonic derived and larval derived hemocytes display differences, however the evidence for this are at the present time rather shallow. Perhaps the depth of the single cell analysis is still inadequate to spot such differences. I would suggest either to delete this part or to expand and provide better evidence.

The introduction mentions the differences and similarities existing between the two populations of hemocytes (embryonic and larval derived), but this is not dealt with in the discussion section. This should be fixed.

Additional points:

The molecular signature of several clusters remains highly unclear or is not visible on Figure 2b, which should represent the strongest markers. The same comment made above for LM1 definition applies to PH1 for which the signature is unclear. The prohemocyte markers Tep4 and Ance are not enriched in this cluster. What defines it as a prohemocyte cluster? PH1 and PH2 only differ by one marker, which is in addition only expressed in 50 % of the cells. Knowing that these clusters are very small, how meaningful are all these subdivisions? Again, PH4 and PH5 present similar profile of markers, and so do PM3 and PM4 (Figure 2B). What distinguish the first ones from the second ones?

Lines 247-252: the mitotic genes are averaged for each cell and for each cluster, only cells presenting a strong average are selected for the representation in Figure 2c. Using average strongly biases the analysis since genes highly expressed will be over represented. It would be more transparent to display the dotplot for each gene of the cell cycle across the clusters. Additionally, it remains unclear to me if the percent expressed in figure 2c (size of the dots) is the proportion of the selected cells or the whole cluster.

The figures are hardly readable, the authors should try and improve them. The font should be larger, the names of the antibodies could perhaps on a white background.

Minor comments

Line 100: add a reference

Line 151: TPM needs to be defined.

Line 183: Extended Data Fig 1e should be Extended Data Fig 1f

Line 283 and following ones contain clear overstatements. For example, Ilp6 is expressed in the PSC but not only, so it cannot be taken as a PSC specific marker.

Line 286: the authors mention Numb as a crystal cell marker. Can the author add it to the list of markers displayed in Fig 2b?

Line 314: can the authors precise the list of mitotic genes highlighting PH1?

Lines 328-329: the link between PH6 and the PM and LM is unclear. Can the authors put the trajectories displayed in Extended Data Fig 4D and Fig 3a,b in the foreground with the position of the clusters?

Line 328: what are the 'late plasmatocytes'?

Lines 364-365: it is misleading to call PPO1 and Atilla mature hemocyte markers. They are exclusive to lamellocytes and crystal cells. In addition, reference 28 do not refer to atilla.

Line 371: Nplp2 is also enriched in CC1.

Line 403: What do the authors mean by "sequentially arrayed"?

Line 435 and following ones: The statement leaves room for ambiguity. Does the ablation of the PSC remove the STAT92 expressing cells or it affects STAT92 expression?

Line 986-987: the authors explain that cluster present in single library were removed. Were the cells removed completely from the analysis or merged with other clusters? In addition, was the size of the library considered? There is a strong heterogeneity in the number of cells per library (from few hundreds to 4000 cells according to Extended data Figure 1a), was this considered? Minor clusters will appear only in library with sufficient cell number.

Line 1395: The link to browse through the data does not seem to work.

Extended data fig 7: presents the data related to the embryonic and larval derived hemocytes. Is the four-colour coding present in panel c really necessary? What is the aim of this further distinction?

Panel a shows the genes in the total hemocyte population, whereas c indicates the genes specific to identified clusters. Perhaps the authors should clarify why not all the genes shown in panel c are indicated in panel a.

Reviewer #3:

Remarks to the Author:

In their revised manuscript Cho et al. provide a significantly improved version of their work, which is now much stronger, more accurate and certainly easier to read for a broad audience. The authors did an excellent job in answering the many comments of the reviewers and I'm fully satisfied with their responses and the modifications they made to the manuscript.

In particular, the reduction of the number of clusters thanks to their new "optimal" subclustering analysis brings a clearer and more appealing picture. They also provide several new pieces of data that confirm their claims and they removed some results which were not fully substantiated or slightly out of focus for the current study.

All together, Cho et al. bring important new insights into the diversity of Drosophila blood cell types. This timely study definitely deserves to be published.

There are still a few mistakes/inconsistencies, but they can easily be corrected by the authors without further revision:

- Page 11/12 (lines 253/255): the description does not really fit with what we see in Fig 2b: Hml expression is not kept "high" in the PM (it decreases a lot from PM1 to PM4).
 - Page 16 (lines 370/371): Nplp2 does not seem lower in CC1 than in PH4/GST-rich/PM1.
 - Page 30 (line 699): reference 82 is the same as 64.
 - Page 32 (line 736): the correct reference here is not Benmimoun et al. PNAS 2015 but Benmimoun et al. Development 2012.
- There is an alteration in Fig. 4b right panel (as seen with the red or blue channel only).
- In Extended data table 3a & 3b: the values in columns C and D are not displayed as % of expressing cells.

Reviewer #4:

Remarks to the Author:

My comments were addressed satisfactorily.

My only new comment is that it would be helpful if the authors discussed in the Introduction and/or Discussion the study that was published recently in eLife and focused on the transcriptomic

progression of hemocyte upon wasp infestation. It is, in some ways, a comparable dataset (at least the part that has to do with was infestation) and would be interesting to see a common conclusion.

REVIEWERS' COMMENTS:

Reviewer #2 (Remarks to the Author):

Response to the revision, Reviewer #2

This is an extensive revision of the manuscript from Shim and collaborators. The authors reanalysed the data and added a significant amount of new data to comply with the comments of the reviewers. This has significantly improved and strengthened the manuscript. Several points need still to be addressed to further clarify the interpretations and to simplify the manuscript. The effort is worth, in view of the fact that it represents the first detailed analysis of the lymph gland and will provide important tools.

Related to points 1 and 2:

The goal of this manuscript is to define the transcriptional asset of the lymph gland and will be taken as a reference by the whole community. The authors should make their data meaningful and useful as much as possible. The finding that the dissociation technique affects the inflammatory status of the hemocytes is already an important piece of information. While it is clear that their approach is the best at the present time, some measures should be taken to clarify the significance of the presented sc RNAseq analysis.

Concerning the lamellocytes in normal condition, what markers have been used to define LM1 as lamellocytes? The only markers displayed in Fig 2b are *Atilla* and *Msn* which are not specifically enriched in LM1 compared to other PM clusters and the manuscript mentions that only *Atilla* was used to define the population. How are *L4*, *L5* and *PPO3* expressed in LM1 compared to the other clusters? Why were they not used?

The lamellocyte cell cluster shown in Figure 1c was defined by using known markers, such as *atilla*. A subsequent subclustering analysis then determined two subclusters within the LM cluster – LM1 and LM2. These two subclusters differed in various known and novel genes, which are listed in Supplementary Data 2.

Concerning the method to remove the bias due to the dissociation of the lymph gland, were the eliminated genes mostly expressed in lamellocytes? The authors should test by immunohistochemistry or in situ the expression of early lamellocyte markers (LM1) as well as that of a selected number of genes that are not specific to lamellocytes and that were eliminated upon comparison with the bulk data. It could well be that a number of those genes is indeed already expressed, albeit at lower level, (and/or in few cells) in control lymph glands. By totally neglecting these genes without an in vivo validation, the output of the analysis will be biased.

It seems risky to remove 4830 genes (almost a third of the *Drosophila* transcriptome) from the analysis based on the comparison of bulk RNAseq of intact lymph gland and single cell RNAseq of dissociated lymph gland. The difference in expression levels can be due to technical differences in the sequencing techniques independently from the dissociation of the lymph gland. I would suggest the authors to validate this strategy by using identical technology (qPCR or bulk RNAseq) to quantify the impact of the dissociation on the expression levels in lymph glands.

Previous studies have already shown that tissue dissociation exaggerates some of the stress responsive genes in the single-cell RNA seq (O'Flanagan et al., *Genome Biology*

2019), and we assume that the same change might take place in our data. We added the concern in the discussion and now the reference is included.

There are slight increases in the mean expression of excluded genes toward LM; however, overall expression levels are uniform across hemocyte subclusters.

Related to point 3:

The authors merged the sc RNAseq data from the different developmental stages to identify the clusters. However, the data on the number of cells/cluster at the three stages (extended table 2) clearly indicate that 1) the 96h AEL stage is largely over represented 2) many clusters only appear at 120h AEL, 3) in early stages, many clusters are made of very few cells, undermining their identification. More than 9 000 cells were analysed at 96h AEL, when the primary lobe contains in average less than 1 400 cells, whereas less than 8 000 cells were analysed at 120 h AEL, when the total n of cells reaches 5 000 in average. Because of the over representation of the 96h AEL stage, the data on the PH4 and the PM1 clusters mostly arise from that stage (e.g. PM1: 5 259 cells at 96h AEL vs. 1 340 cells at 120h AEL) and the same is true for the other clusters as well. PH5, PH6, PM2 PM3, PM4, LM1 and the adipohemocytes are basically only present at 120h AEL, reducing significantly the diversity at early stages. Indeed 120 h AEL is the only stage that really shows a significant heterogeneity of hemocytes. Overall, the merge of the data from the three stages to characterize a cluster introduces a strong bias in the interpretation of the sc RNAseq data.

The authors compare the clusters defined by the analysis of all lymph gland stages combined with the clusters defined by the independent analysis of each stage. They do not find any significant difference between the two approaches and illustrate their conclusions with heatmaps of the main markers of PH and PM clusters at each stage (Extended Data Figure 2c,d). However, the list and order of the markers used for each heatmap change from one stage to the next, which renders the comparison difficult to visualise. Also, the name of the clusters changes at the different stages (PM1 and PM1 mature at 72h AEL; PH2/3 as well as PH3/4 at 120h AEL). In addition, the comparison of the two approaches is illustrated by coloured bars above each Heatmap, which does not allow a clear appreciation of the similarities between the two approaches. For example, several clusters seem to be joined in the split approach and separated in the combined approach (Extended Data Figure 2C).

This appears to me as a significant difference between the two approaches. Could the authors illustrate the similarities by displaying dotplots (like in Figure 2b) with the top 5 markers of each clusters in the combined approach and the split approach (one per stage)? If the two clustering strategies are highly similar, the main markers will be conserved and clearly visible with this representation.

As per the referee's request, timepoint-combined and -split approaches were compared at a subcluster level with the top five highly expressed markers, as shown in the reviewer's Figure (see below). As a result, both PH and PM subclusters represent common markers by two independent approaches, suggesting that the subclusters are determined with similar markers regardless of technical approaches. These data are now included in Supplementary Figure 2c-d.

Related to points 5 and 6

The PH cells likely provide a pool of renewing precursors, in line with their constant number over time and their position within the lymph gland. The conclusion on the presence on stem-like cells awaits nevertheless *in vivo* validation. PM1 cells are also proliferative and their number also stays constant, but despite the proposed mechanisms, their destiny is not clear. Here and throughout the manuscript, the authors indulge in speculations that at time dilute out the most important message of the manuscript, i.e. the presence of heterogeneous hemocyte populations in the lymph gland hematopoietic tissue and the identification of novel cell types. The revision has already significantly improved the quality of the data, but a further simplification would still benefit to the manuscript, which remains very dense. Perhaps some speculations could make the object of a review article.

We agree with the reviewer and will further characterize the PH1 population and write a review article in the future.

Related to point 7:

To confirm the presence of adipohemocytes, the authors co-labelled lymph gland for Sirup or Lsd2 and Hml (Hml delta> GFP). The data would be more convincing with a positive marker of adipohemocyte such as Srp or eater, which seem strongly expressed in this cluster. Similarly, for the Gst-rich cluster, co-labelling with Nplp2 or Hml should provide a positive co-labelling.

Also, since the Nile Red positive cells are not colabeled with an adipohemocyte marker, it is hard to correlate the two labelling directly. As a minimum, the authors should compare the position and the percentage of cells that are Nile Red positive with those of the cells that are positive for an adipohemocyte marker.

Perhaps the authors could accompany the schematic drawing in panel 4H with the name of the major markers used to identify the different clusters. Also, they do show the location of the PSC, perhaps they could also indicate the other zones that constitute the gland (intermediate, cortical, etc).

We included the location of zones and a list of representative markers in Fig 4h (now Fig 5h) for the clear visualization of our analysis.

Related to point 13

In Reviewer Figure 15, the authors compare the Drop-seq data with InDrop-seq data and illustrate the comparison with a dotplot of the main cluster markers. Since Ubx represents the strongest marker of circulating cells, this dotplot has to include Ubx and ideally all the markers shown in Figure 6c to be convincing.

Figure 6a shows a relatively similar profile for the embryonic and larval derived hemocytes. Nevertheless, the profiles shown in reviewer figure 15, which presents the drop-seq data for embryonic and larval derived hemocytes, are quite different. How can this discrepancy be explained?

We also noticed the single-cell platform-specific expression of some genes including Ubx due to unknown reasons. However, the majority of makers were similarly expressed between the two platforms and the Ubx expression was validated in circulating PM by immunohistochemistry.

Related to point 14:

The authors say that “it is infeasible to identify particular genes or gene sets”. If I understand properly, the authors say that it is not possible to identify genes expressed at different levels in the lymph gland compared to circulating hemocytes. If this is the case, what is the validity of the analysis presented in Extended Data Fig7?

Because the label transfer analysis was performed in the latent space, genes or gene sets used in the analysis cannot be extracted. Nevertheless, once cells were annotated using the label transfer, the DEGs between a certain cell-type in the LG and circulation can be calculated based on the reference gene annotation.

Related to point 15:

The comment was related to the significance of markers expressed in low percentage of cells of the clusters. I understand the statistical significance of the markers, however, the biological significance is less clear to me. For example, Ubx is expressed in 25% of the circulating cells, significantly enriched in circulating PM and absent from the lymph gland (Figure 6c). Could that mean that Ubx is expressed exclusively in a subcluster of circulating PM? In addition, the immunolabelling of Ubx in circulating cells (Extended Data Fig 7) seem to indicate that all circulating PM express Ubx. Would it be possible to quantify the number of circulating PM expressing Ubx to support the single cell analysis?

Ubx is detected in almost every hemocytes in InDrop-seq data; however, Drop-seq does not show the Ubx expression. We validated that ~95% of circulating hemocytes display Ubx protein expression, which is consistent with the InDrop-seq result. This discrepancy of Ubx expression in the two different platforms might be due to trivial differences of handlings and preparation procedures yet it is not clear as to why the difference arose.

It is very likely that the embryonic derived and larval derived hemocytes display differences, however the evidence for this are at the present time rather shallow. Perhaps the depth of the single cell analysis is still inadequate to spot such differences. I would suggest either to delete this part or to expand and provide better evidence.

The introduction mentions the differences and similarities existing between the two populations of hemocytes (embryonic and larval derived), but this is not dealt with in the discussion section. This should be fixed.

Due to the word limit, we are not able to include additional discussion in the current manuscript. We hope to expand the discussion in a review article in near future.

Additional points:

The molecular signature of several clusters remains highly unclear or is not visible on Figure 2b, which should represent the strongest markers. The same comment made above for LM1 definition applies to PH1 for which the signature is unclear. The prohemocyte markers *Tep4* and *Ance* are not enriched in this cluster. What defines it as a prohemocyte cluster?

PH1 and PH2 only differ by one marker, which is in addition only expressed in 50 % of the cells. Knowing that these clusters are very small, how meaningful are all these subdivisions?

Clusters and subclusters are defined by a set of genes and we only displayed a part of the top genes or known genes in Figure 2b. We provide the entire list of marker genes in Supplementary Data 2 and attach a dotplot representing additional marker genes below.

Again, PH4 and PH5 present similar profile of markers, and so do PM3 and PM4 (Figure 2B). What distinguish the first ones from the second ones?

We provided a list of marker genes that are differentially expressed or gradually changing between PH4 and PH5. Also, PM3 and PM4 show gradual changes in the expression of a few genes. Despite the variable expressions of marker genes in each subcluster, functions of these genes are unknown and are subject to future studies. These markers are listed in our Supplementary Data 2.

Lines 247-252: the mitotic genes are averaged for each cell and for each cluster, only cells presenting a strong average are selected for the representation in Figure 2c. Using average strongly biases the analysis since genes highly expressed will be over represented. It would be more transparent to display the dotplot for each gene of the cell cycle across the clusters. We indicated the list of cell cycle genes in the Figure legend as well as in the methods section.

Additionally, it remains unclear to me if the percent expressed in figure 2c (size of the dots) is the proportion of the selected cells or the whole cluster.

It is the percent of cells in each cluster/subcluster. i.e. about 25% of PM1 cells express G1, G2, or M phase genes. We edited the figure legend to included genes analyzed.

The figures are hardly readable, the authors should try and improve them. The font should be larger, the names of the antibodies could perhaps on a white background.
 We increased the font size and placed the names of antibodies outside of images.

Minor comments

Line 100: add a reference

We added the reference.

Line 151: TPM needs to be defined.

We defined TPM (transcripts per million) in the text (Supplementary Figure legend).

Line 183: Extended Data Fig 1e should be Extended Data Fig 1f

We changed the figure calling.

Line 283 and following ones contain clear overstatements. For example, Ilp6 is expressed in the PSC but not only, so it cannot be taken as a PSC specific marker.

We agree with the comment and added the point that Ilp6 is also expressed in the other subclusters in the figure legend.

Line 286: the authors mention Numb as a crystal cell marker. Can the author add it to the list of markers displayed in Fig 2b?

Numb is highly expressed and specific to CC2 subcluster in scRNA-seq data and we added Numb in Fig 2b as the reviewer suggested.

Line 314: can the authors precise the list of mitotic genes highlighting PH1?

Cell cycle genes are mentioned in the methods section, "Cell cycling scores" and again highlighted in the figure legend.

Lines 328-329: the link between PH6 and the PM and LM is unclear. Can the authors put the trajectories displayed in Extended Data Fig 4D and Fig 3a,b in the foreground with the position of the clusters?

We displayed the possible position of PH6 in the PM/LM differentiation in Extended Data Fig. 4e (now Supplementary Figure 3e).

Line 328: what are the 'late plasmacytes'?

PM3 and PM4 are the late plasmacytes and we added a description in the line.

Lines 364-365: it is misleading to call PPO1 and Atilla mature hemocyte markers. They are exclusive to lamellocytes and crystal cells. In addition, reference 28 do not refer to atilla.

We modified the sentence to clearly state that PH4, GST-rich, and PM1 are devoid of mature plasmacyte-, crystal cell-, or lamellocyte markers. Reference 28 (now 17) is to refer the presence of the intermediate zone.

Line 371: Nplp2 is also enriched in CC1.

We added CC1 in the sentence.

Line 403: What do the authors mean by "sequentially arrayed"?

We modified the sentence.

Line 435 and following ones: The statement leaves room for ambiguity. Does the ablation of the PSC remove the STAT92 expressing cells or it affects STAT92 expression?

Given that STAT92::edGFP represents the JAK/STAT activity, we assume that loss of the PSC may attenuate the JAK/STAT activity in PH1, consequently leading to the prohemocyte differentiation and the lack of PH1 population. We hypothesize that the same loss can change the Delta expression. Future studies will clarify as to how the PSC ablation shifts the heterogeneity of the prohemocyte pool and the hematopoietic landscape.

Line 986-987: the authors explain that cluster present in single library were removed. Were the cells removed completely from the analysis or merged with other clusters? In addition, was the size of the library considered? There is a strong heterogeneity in the number of cells per library (from few hundreds to 4000 cells according to Extended data Figure 1a), was this considered? Minor clusters will appear only in library with sufficient cell number.

Subclusters enriched only by a single library were excluded in further analysis. The size of the libraries was not considered in this case. Minor subclusters, such as PH1 or PH2, are present in every library.

Line 1395: The link to browse through the data does not seem to work.

We modified the browser and confirmed that the link works well now.

Extended data fig 7: presents the data related to the embryonic and larval derived hemocytes.

Is the four-colour coding present in panel c really necessary? What is the aim of this further distinction?

The color coding follows the cell type colors in Fig. 6b (now Fig. 8b). We now added a color legend in the figure.

Panel a shows the genes in the total hemocyte population, whereas c indicates the genes specific to identified clusters. Perhaps the authors should clarify why not all the genes shown in panel c are indicated in panel a.

Panel a (now Supplementary Fig 5a) compares tissue-level samples which have different compositions of cell-types, and the resulting DEGs would be affected by these factors. For example, PH cells are relatively enriched in the LG and their marker genes, such as *Tep4* or *Ance*, are found in the up-regulated genes in the LG. In the same manner, circulating hemocytes which are consisted of a higher proportion of PM and CC show up-regulated *Hml*, *Pxn*, or *PPO2*. These DEG results thus can be different from panel c which compares individual subcluster.

Reviewer #3 (Remarks to the Author):

In their revised manuscript Cho et al. provide a significantly improved version of their work, which is now much stronger, more accurate and certainly easier to read for a broad audience. The authors did an excellent job in answering the many comments of the reviewers and I'm fully satisfied with their responses and the modifications they made to the manuscript.

In particular, the reduction of the number of clusters thanks to their new "optimal" subclustering analysis brings a clearer and more appealing picture. They also provide several new pieces of data that confirm their claims and they removed some results which were not fully substantiated or slightly out of focus for the current study.

All together, Cho et al. bring important new insights into the diversity of *Drosophila* blood cell types. This timely study definitely deserves to be published.

There are still a few mistakes/inconsistencies, but they can easily be corrected by the authors without further revision:

- Page 11/12 (lines 253/255): the description does not really fit with what we see in Fig 2b: *Hml* expression is not kept "high" in the PM (it decreases a lot from PM1 to PM4).

We modified the sentence.

- Page 16 (lines 370/371): *Nplp2* does not seem lower in CC1 than in PH4/GST-rich/PM1.

We added CC1 in the sentence.

- Page 30 (line 699): reference 82 is the same as 64.

We corrected the reference.

- Page 32 (line 736): the correct reference here is not Benmimoun et al. PNAS 2015 but Benmimoun et al. Development 2012.

We changed the reference.

- There is an alteration in Fig. 4b right panel (as seen with the red or blue channel only).

We modified the figure.

- In Extended data table 3a & 3b: the values in columns C and D are not displayed as % of expressing cells.

We updated the table.

Lucas Waltzer

Reviewer #4 (Remarks to the Author):

My comments were addressed satisfactorily.

My only new comment is that it would be helpful if the authors discussed in the Introduction and/or Discussion the study that was published recently in eLife and focused on the transcriptomic progression of hemocyte upon wasp infestation. It is, in some ways, a comparable dataset (at least the part that has to do with wasp infestation) and would be interesting to see a common conclusion.

We will extend our discussion in a review article in the future.